# A Tale of Two Problems: Multi-Task Bilevel Learning Meets Equality Constrained Multi-Objective Optimization

**Zhiyao Zhang** [1]   **Myeung Suk Oh** [1]   **Zhen Qin** [1]   **Jiaxiang Li** [2]   **Xin Zhang** [2]   **Jia Liu** [1]

## Abstract

In recent years, bilevel optimization (BLO) has attracted significant attention for its broad applications in machine learning. However, most existing works on BLO remain confined to the single-task setting and rely on the lower-level strong convexity assumption, which significantly restricts their applicability to modern machine learning problems of growing complexity. In this paper, we make the first attempt to extend BLO to the multi-task setting under a relaxed lower-level general convexity (LLGC) assumption. To this end, we reformulate the multi-task bilevel learning (MTBL) problem with LLGC into an equality constrained multi-objective optimization (ECMO) problem. However, ECMO itself is a new problem that has not yet been studied in the literature. To address this gap, we first establish a new Karush–Kuhn–Tucker (KKT)-based Pareto stationarity as the convergence criterion for ECMO algorithm design. Based on this foundation, we propose a weighted Chebyshev (WC)-penalty algorithm that achieves a finite-time convergence rate of $\mathcal{O}(ST^{-\frac{1}{2}})$ to KKT-based Pareto stationarity in both deterministic and stochastic settings, where $S$ denotes the number of objectives, and $T$ is the total iterations. Moreover, by varying the preference vector over the $S$-dimensional simplex, our WC-penalty method systematically explores the Pareto front. Finally, solutions to the ECMO problem translate directly into solutions for the original MTBL problem, thereby closing the loop between these two foundational optimization frameworks.

[1]Department of Electrical and Computer Engineering, The Ohio State University, Columbus, OH, USA [2]Meta Platforms, Inc., Menlo Park, CA, USA. Correspondence to: Zhiyao Zhang <zhang.15178@osu.edu>, Jia Liu <liu@ece.osu.edu>.

*Proceedings of the 43rd International Conference on Machine Learning*, Seoul, South Korea. PMLR 306, 2026. Copyright 2026 by the author(s).

## 1. Introduction

**1) Background and Motivation:** As machine learning frameworks have grown increasingly complex in recent years, the demand for addressing learning problems with nested structures has become ever more compelling. Such demands typically arise from two distinct perspectives: 1) multiple, potentially conflicting tasks often need to be considered, and 2) the learning of some tasks often depend on the outcome(s) of other tasks. For instance, when aligning pre-trained large language models (LLMs) with human feedback, one needs to consider various human-aligned criteria on the one hand; on the other hand, many tasks (e.g., policy parameter optimization of LLM alignments and the actor-critic framework in reinforcement learning) often contain a subtask on reward model learning. As a result, recent years have seen growing interests in the Multi-Task Bilevel Learning (MTBL) problems in the following form: (Ye et al., 2021; Gu et al., 2023; Li et al., 2024; Wang et al., 2024; Yang et al., 2024b; Ye et al., 2024; Zhang et al., 2026):

$$\min_{x,y} F(x,y) = \big[f_1(x,y),\ldots,f_S(x,y)\big]^\top$$
$$\text{s.t. } y \in \mathcal{M}(x) := \arg\min_y g(x,y), \qquad \text{(MTBL)}$$

where $S$ denotes the number of tasks, and $x \in \mathbb{R}^p, y \in \mathbb{R}^q$. For example, in the aforementioned LLM alignment, the upper-level (UL) problem corresponds to minimizing the validation loss with respect to multiple human-aligned metrics, such as *helpfulness* and *toxicity*, while the lower-level (LL) problem corresponds to a data weighting task, aiming to curate a high-quality training dataset.

Despite its significance, solving the MTBL problem is highly challenging due to the complex couplings between upper and lower levels and the trade-offs among multiple objectives. So far, most existing works on bilevel optimization (BLO) in the literature rely on the lower-level strong convexity (LLSC) assumption (see, e.g., (Ghadimi & Wang, 2018; Arbel & Mairal, 2021; Ji et al., 2021; Dagréou et al., 2022)). Specifically, this widely adopted LLSC assumption requires that, for any given $x$, $g(x,\cdot)$ is strongly convex with respect to $y$. It is worth noting that the LLSC assumption renders a much simplified and tractable BLO algorithm design and analysis, since the LLSC assumption i) ensures the

existence of a unique solution $y^*(x)$ of the LL problem, and ii) implies a well-defined hyper-gradient $\nabla F(x, y^*(x))$ that requires non-singular Hessian $\nabla_y^2 g(x, y^*(x))$ (Ghadimi & Wang, 2018; Ji et al., 2021). However, the LLSC assumption significantly restricts the applicability of BLO to modern machine learning problems of growing complexity.

While several recent works in the BLO literature have attempted to relax the LLSC assumption to the lower-level general convexity (LLGC) assumption (i.e., the LL function $g(x, \cdot)$ is convex but may not be strongly convex with respect to $y$ for any $x$) (Sabach & Shtern, 2017; Liu et al., 2023a; Cao et al., 2023; Jiang et al., 2023; Yao et al., 2024; Chen et al., 2024b; Lu & Mei, 2024), all existing works remain confined to the single-task setting, while the *multi-task* bilevel optimization problem under the LLGC assumption has yet to be explored. A key challenge in solving MTBL problems under the LLGC assumption stems from the fact that, not only does the hyper-gradients of the MTBL problem become ill-defined due to the lack of LLSC condition, the optimality of the UL subproblem also needs to be re-interpreted in the Pareto equilibrium sense due to the trade-off among multiple tasks. This renders most of the algorithmic techniques developed for single-task LLGC-BLO problems inapplicable. The widening gap between the rapidly growing demand for addressing more general MTBL problems and the inherent limitations of existing BLO techniques motivates us, in this work, to investigate MTBL under the LLGC assumption.

**2) Overview of Our Proposed Approach:** To address the ill-defined hyper-gradient challenge in the MTBL problem under the LLGC assumption, our key idea is to *indirectly* solve the MTBL problem by transforming this problem into an equivalent *single-level constrained* multi-objective optimization that shares the same optimal solutions as the original problem. To this end, we note that solving the lower-level problem in MTBL with $g(x, y)$ being convex for any $x$ is equivalent to solving its first-order stationarity condition $\nabla_y g(x, y) = 0$, which is both necessary and sufficient. This implies that we can reformulate the LLGC-MTBL problem as an equality constrained multi-objective (ECMO) problem as follows (also see Step ① in Fig. 1):

$$\min_{z \in \mathbb{R}^k} F(z) = \big[f_1(z), \dots, f_S(z)\big]^\top$$
$$\text{s.t. } h_i(z) = 0, i = 1, \dots, q, \qquad \text{(ECMO)}$$

where $k := p + q$, $z := [x^\top, y^\top]^\top$ and $h_i(z) := \nabla_{y_i} g(x, y) = 0$. However, even after the ECMO reformulation, we remain far from resolving the MTBL problem, as the ECMO problem itself constitutes a *new* formulation that has not yet been examined in the literature. Specifically, while multi-objective optimization (MOO) problems have been extensively studied (see, e.g., (Sawaragi et al., 1985; Ehrgott, 2005; Désidéri, 2012; Sener & Koltun, 2018;

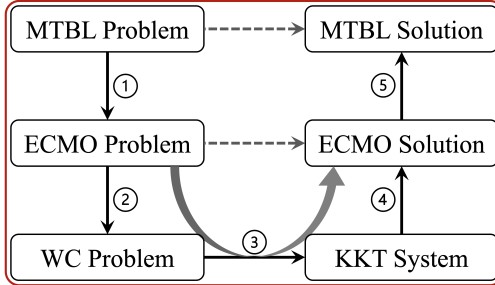

*Figure 1.* Roadmap of our proposed approach for solving the MTBL problem under the LLGC assumption.

Momma et al., 2022; Fernando et al., 2023)), the majority of existing works only considered unconstrained MOO. Meanwhile, constrained MOO problems, including ECMO, are still in their infancy. To date, although several heuristic algorithms have been proposed for ECMO and empirically validated (Qu & Suganthan, 2011; Yang et al., 2019; Cuate et al., 2020; García et al., 2021), none of these existing works offers theoretical performance guarantees in terms of finite-time convergence rate or sample/iteration complexity. To establish the theoretical foundation for solving ECMO (and thus for the LLGC-MTBL problem), there are two main technical challenges: (1) **Lack of Pareto Optimality Condition Characterizations and Appropriate Convergence Metrics:** Unlike unconstrained MOO problems, where the Pareto stationarity can be conveniently employed as the necessary condition of the Pareto optimality for algorithm design, the characterization of Pareto stationarity for ECMO remains unclear. Consequently, the current literature lacks appropriate convergence metrics for solving the ECMO problem; (2) **Algorithm Design and Theoretical Analysis:** Even with the Pareto stationarity characterization and convergence metrics established for ECMO, developing algorithms that can handle the equality constraints in ECMO and enable convergence analysis remains challenging. We address these challenges with the following contributions:

- To rigorously characterize the Pareto stationarity for ECMO problems, we leverage the weighted-Chebyshev (WC) scalarization technique by exploiting the one-to-one correspondence between the Pareto front of the ECMO problem and the set of solutions to the WC-scalarized problem under varying preference weights. This establishes a direct connection between the WC-scalarized problem and the original ECMO problem (cf. Step ② in Figure 1). Subsequently, this one-to-one correspondence allows us to employ the Karush–Kuhn–Tucker (KKT) conditions of the WC-scalarized problem as the necessary and sufficient condition of the Pareto stationarity for the ECMO problem, thereby resolving the challenge of characterizing the Pareto stationarity for ECMO. (cf. Step ③ in Figure 1).

- Based on the KKT-based Pareto stationarity for ECMO, we proposed a WC-Penalty algorithm to solve the

ECMO problem, and establish its finite-time convergence rate guarantee (cf. Step ④ in Figure 1). Specifically, our WC-Penalty method achieves the KKT-based Pareto stationarity at a rate of $\mathcal{O}(ST^{-\frac{1}{2}})$, where $T$ denotes the total number of iteration steps. In addition, by varying the preference vector over the $S$-dimensional simplex, our WC-Penalty method systematically explores the Pareto front.

- Finally, solutions to the ECMO problem translate directly into solutions for the original MTBL problem, thereby closing the loop between these two foundational optimization frameworks (cf. Step ⑤ in Figure 1). In addition, we evaluate our approach on two multi-objective data weighting tasks: reward model training for RLHF, and LLM alignment. Extensive numerical experiments further validate the efficiency of our proposed algorithms across diverse settings.

The remainder of this paper is organized as follows. In Sec. 2, we provide an overview of closely related works. In Sec. 3, we focus on characterizing the Pareto stationarity and establishing convergence metrics for ECMO. In Sec. 4, we will present our WC-Penalty method for ECMO and its convergence rate analysis. In Sec. 5, we will close the loop between ECMO and MTBL by solving two MTBL problems through the lens of ECMO, and Sec. 6 concludes this paper. We used open-source models developed by Meta.

## 2. Related Work

In this section, we provide a brief overview of two lines of research that are closely related to this work, thereby placing our contributions into a comparative perspective.

**1) Multi-Task Bilevel Learning (MTBL):** MTBL problems have received increasing attention in recent years (Ye et al., 2021; Gu et al., 2023; Fernando et al., 2023; Li et al., 2024; Wang et al., 2024; Yang et al., 2024b; Ye et al., 2024). However, in contrast to the more mature bodies of work on MOO and BLO, the theoretical foundations of MTBL remain largely underdeveloped. Among these works, (Yang et al., 2024b; Ye et al., 2021) demonstrated that their proposed algorithms converge asymptotically, but without providing theoretical guarantees of finite-time convergence rate. In contrast, (Fernando et al., 2023; Ye et al., 2024) proposed algorithms with a finite-time convergence rate of $\mathcal{O}(ST^{-\frac{1}{2}})$ and $\mathcal{O}(ST^{-\frac{1}{4}})$, respectively. However, all of these works heavily depend on the LLSC assumption: not only is the algorithmic framework built upon the LLSC assumption, but the optimality criterion also relies on it. Therefore, this significantly limits their applicability to complex real-world scenarios where the LLSC assumption is usually violated.

**2) Equality Constrained Multi-Objective (ECMO):** ECMO problems have found many applications across var-

ious fields, including resource allocation, scheduling optimization, and path planning, just to name a few (Liang et al., 2022; Hao et al., 2024). The most closely related works on ECMO problems are (Cuate et al., 2020; García et al., 2021). Both works proposed algorithmic solutions for ECMO and conduct numerical experiments to validate their methods. However, neither work provided any finite-time convergence guarantees, highlighting that the theoretical foundations for ECMO remain missing. Due to space limitation, we relegate additional detailed comparison and other related work on closely related topics to Appendix B.

## 3. ECMO: Characterizing Pareto Stationarity and Establishing Convergence Metrics

In this section, we will characterize the Pareto stationarity and establish the convergence metrics for the ECMO problem, which lays the foundation for the algorithmic design of ECMO and eventually solving MTBL in later sections.

### 3.1. Pareto Stationarity for ECMO

As in other multi-objective optimization problems, multiple objectives in an ECMO problem could be conflicting with each another. Thus, in general, there does not exist a unique minimizer $z^*$ that simultaneously minimizes all $S$ objectives $f_s(z)$ in ECMO. Hence, an optimal solution to the ECMO problem needs to be interpreted in the Pareto sense.

**Definition 1** (Pareto Optimality). A solution $z$ dominates another solution $z'$ if and only if $f_s(z) \leq f_s(z'), \forall s \in [S]$, and there exists at least one $s \in [S]$ such that the inequality holds strictly. A feasible $\tilde{z}$ is Pareto optimal if and only if no other feasible $\hat{z}$ dominates $\tilde{z}$.

Intuitively, Pareto optimality means that no objective can be improved without sacrificing at least one other objective. A weaker, yet useful, notion is the weak Pareto optimality:

**Definition 2** (Weak Pareto Optimality). A feasible $\tilde{z}$ is called weakly Pareto optimal if and only if no other feasible $\hat{z}$ satisfies: $f_s(\hat{z}) < f_s(\tilde{z}), \forall s \in [S]$.

Clearly, Pareto optimality implies weak Pareto optimality, whereas the converse is not always true. In addition, the set of all (resp. weakly) Pareto optimal points is referred to as the (resp. weak) Pareto set and denoted as $X_{\mathrm{P}}$ (resp. $X_{\mathrm{WP}}$), and the (resp. weak) Pareto front is defined as $\{F(x) : x \in X_{\mathrm{P}}\}$ (resp. $\{F(x) : x \in X_{\mathrm{WP}}\}$). Further, the (weak) Pareto optimality in ECMO is subject to feasibility, i.e., under the constraint $h(z) = \mathbf{0}$ ($h(z) := [h_1(z), \ldots, h_q(z)]^\top$).

However, for nonconvex multi-objective optimization problems, finding (weakly) Pareto optimal solutions is NP-hard in general. Thus, it is often of practical interest to find a Pareto-stationary solution instead, which is the necessary condition of a (weakly) Pareto optimal solution. Intuitively, Pareto stationarity can be interpreted as no common descent

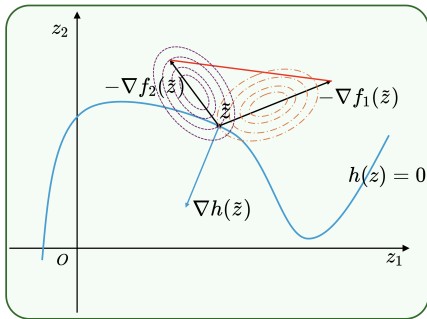

*Figure 2.* $\tilde{z}$ is Pareto stationary but violates Definition 3.

direction exists locally. Note that for unconstrained MOO problems, Pareto stationarity can be defined as follows:

**Definition 3** (Pareto Stationarity for Unconstrained MOO). For the unconstrained MOO problem $\min_z F(z)^\top = (f_1(z), \ldots, f_S(z))$, $\tilde{z}$ is a Pareto stationary point if and only if there does not exist a direction $d \in \mathbb{R}^k$, such that $\nabla f_s(\tilde{z})^\top d < 0, \forall s \in [S]$.

Moreover, the following equivalent Pareto stationarity characterization for unconstrained MOO is often used in practice, which is more amenable for algorithm design: $\tilde{z}$ is a Pareto stationary point if and only if $\exists \alpha \in \Delta_S^+$ ($S$-simplex) such that, $(\nabla f_1(\tilde{z}), \ldots, \nabla f_S(\tilde{z})) \alpha = \mathbf{0}$ (Sener & Koltun, 2018; Lin et al., 2024). As a result, $\|\nabla F(z)\alpha\|_2^2$ can be used as a natural metric for Pareto stationarity in unconstrained MOO problems, i.e., if $\|\nabla F(\tilde{z})\alpha\|_2^2 \le \epsilon$ for some $\epsilon > 0$, then $\tilde{z}$ is called an $\epsilon$-Pareto stationary solution.

Given the multi-objective nature of ECMO, one might be tempted to adopt the same Pareto stationarity definition as in unconstrained MOO. However, as we demonstrate through a counterexample, the Pareto stationarity definition for unconstrained MOO does not hold in the ECMO setting, thereby necessitating a **new** characterization of Pareto stationarity. Consider a two-objective ECMO problem as shown in Figure 2. On the one hand, based on the notion of "common descent direction", the point $\tilde{z}$ is Pareto stationary since any deviation from $\tilde{z}$ to another feasible point $\hat{z}$ on $h(z) = 0$ in the local neighborhood must result in an increase in either $f_1(z)$ or $f_2(z)$. On the other hand, both Definition 3 and its equivalent definition suggest that $\tilde{z}$ is not Pareto stationary, since the vector $-(\alpha \nabla f_1(\tilde{z}) + (1-\alpha) \nabla f_2(\tilde{z}))$ for any $\alpha \in \Delta_2^+$, which can be represented by a point in the red line segment in Figure 2, is a nonzero vector. A key reason that Definition 3 fails under ECMO is primarily due to a lack of feasibility consideration in Definition 3. To address the limitation of Definition 3, we extend the definition of Pareto stationarity to ECMO problems as follows:

**Definition 4** (Pareto Stationarity for ECMO). For an ECMO problem with its *tangent cone* $\mathcal{D}$, a direction $d$ at a solution $z$ is feasible if $z + \epsilon d \in \mathcal{D}$ for small enough $\epsilon > 0$. In ECMO, a feasible point $\tilde{z}$ is Pareto stationary if and only if there does not exist a feasible direction $d \in \mathbb{R}^k$ such that

$\nabla f_s(\tilde{z})^\top d < 0, \forall s \in [S]$.

It is clear that, under ECMO, the Pareto stationary point $\tilde{z}$ in Fig. 2 satisfies Definition 4. To our knowledge, Definition 3 is a **new** result in the literature.

Although Definition 4 is a proper Pareto stationarity definition for ECMO problems, it turns out that deriving an equivalent Pareto stationarity characterization similar to that under unconstrained MOO and more amenable for algorithm design remains nontrivial. An intuitive guess of Pareto stationarity characterization for ECMO is to check if $\nabla F(\tilde{z})\alpha + \nabla h(\tilde{z})v = \mathbf{0}$ and $h(\tilde{z}) = \mathbf{0}$ hold simultaneously for some $\alpha \in \Delta_S^+, v \in \mathbb{R}^q$. However, although this may appear plausible and aligns with the example shown in Fig. 2, it can be invalidated by counterexamples (cf. Appendix C).

This indicates that the characterization of Pareto stationarity for ECMO must be derived through a rigorous and systematic approach rather than relying on intuitive "guesswork". To address this challenge, we first establish a result that paves the way to derive our Pareto stationarity characterization for subsequent algorithmic design for solving ECMO. We say that a solution $\tilde{z}$ is a locally weakly Pareto optimal point for ECMO if there exists some $\delta > 0$, such that $\tilde{z}$ is weakly Pareto optimal within the feasible region $\mathcal{D}(\tilde{z}, \delta) := \mathcal{D} \cap N_\delta(\tilde{z})$, where $\mathcal{D}$ is the tangent cone and $N_\delta(\tilde{z}) = \{z \in \mathbb{R}^k : \|z - \tilde{z}\|_2 \le \delta\}$. With the notion of locally weak Pareto optimality, we have the following theorem that reveals an important insight that Pareto stationarity in ECMO can be implied by the locally weak Pareto optimality (see Appendix C for proofs).

**Theorem 1.** *If $\tilde{z}$ is locally weakly Pareto optimal, then $\tilde{z}$ is also Pareto stationary. Conversely, if $\tilde{z}$ is Pareto stationary and if, for any $s \in [S]$, there exists $\mu_s \in \mathbb{R}^q$ such that 1) $\nabla f_s(\tilde{z}) + \sum_{i=1}^q \mu_{s,i} \nabla h_i(\tilde{z}) = 0$, and 2) $\sum_{i=1}^q \mu_{s,i} \nabla h_i(\tilde{z})^\top d \ne 0$ for any feasible direction $d$, then $\tilde{z}$ is locally weakly Pareto optimal.*

Later, we will leverage Theorem 1 to derive the Pareto stationarity characterization for ECMO via the one-to-one mapping between ECMO and its WC-scalarization (see Step ② in Figure 3).

### 3.2. One-to-One Mapping between ECMO and Its Weighted-Chebyshev Scalarization

Weighted-Chebyshev (WC)-scalarization is a technique to convert a vector-valued MOO problem into a conventional scalar-valued optimization problem. Specifically, WC minimizes a weighted $\ell_\infty$-norm of the vector-valued objective of an MOO problem (i.e., improving the worst-performing objective). Let $\Delta_S^{++}$ denote the strictly positive $S$-dimensional simplex. For ECMO problems, WC can be written as: $\min_z \|\lambda \odot F(z)\|_\infty$, s.t. $h_i(z) = 0, i = 1, \ldots, q$, where $\lambda \in \Delta_S^{++}$ is a given preference vector. Further, to address

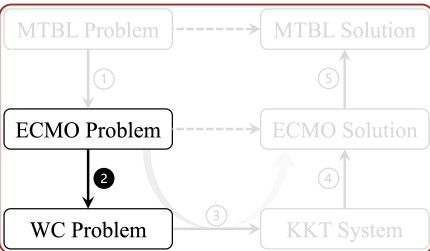

*Figure 3.* One-to-one mapping between ECMO and its WC-scalarized problem.

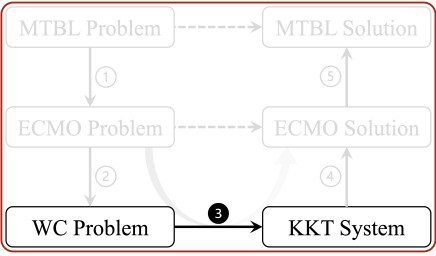

*Figure 4.* Characterize a locally optimal WC-solution by its KKT system.

the non-smoothness of "min-max" operation introduced by the $\ell_\infty$-norm minimization, we can further reformulate the WC-scalarization for the ECMO problem as follows:

$$\min_{\rho, z} \rho,$$
$$\text{s.t. } h_i(z) = 0, i = 1, \ldots, q, \lambda_s f_s(z) \leq \rho, s \in [S]. \quad \text{(WC)}$$

It is well known in the MOO literature that there exists a one-to-one mapping between the solutions of WC-scalarization and the Pareto front of the original MOO problem, which implies that one can systematically explore the entire weak Pareto front $X_{\text{WP}}$ by varying the preference vector over the $S$-dimensional simplex $\Delta_S^{++}$ (Ehrgott, 2005; Lin et al., 2024; Qiu et al., 2024). However, this one-to-one mapping result depends on solving the WC-scalarized problem to optimality, which is challenging due to the potential non-convexity of the MOO problem. Fortunately, in the next theorem, we will show that the locally optimal WC-solution and locally weak Pareto optimal solution (or equivalently, the Pareto stationary solution by Theorem 1) of ECMO are also one-to-one mapped.

**Theorem 2.** *Suppose that $f_s(z) > 0, \forall s \in [S]$[1]. Then, $\tilde{z}$ is a locally weak Pareto optimal solution of ECMO if and only if $(\tilde{\rho}, \tilde{z})$ is a locally optimal WC-solution for some $\tilde{\rho} \in \mathbb{R}$ and $\lambda \in \Delta_S^{++}$.*

With Theorem 2, we are now ready to characterize the Pareto stationarity and derive convergence metrics for ECMO using the Karush-Kuhn-Tucker (KKT) system of the WC-scalarized problem.

### 3.3. Pareto Stationarity Characterization and Convergence Metric Derivations for ECMO

Note that the WC-scalarized problem is a single-level single-objective optimization problem. Following from (Bazaraa et al., 2006), the locally optimal solution of the WC-scalarized problem can be characterized by its KKT sys-

tem (see Fig. 4) and an appropriate constraint qualification as follows (see details in Appendix C.4):

**Lemma 1.** *For the WC-scalarized problem, the following results hold:*

1. *(**Necessity**) Suppose that $(\tilde{\rho}, \tilde{z})$ is a locally optimal WC-solution, and the linearly independent constraint qualification (LICQ)[2] holds at $\tilde{z}$, i.e., $\{\nabla h_i(\tilde{z}), \nabla f_s(\tilde{z}) | i \in [q], s \in \{s \in [S] : \lambda_s f_s(\tilde{z}) = \tilde{\rho}\}\}$ are linearly independent. Then, the KKT condition is satisfied at $(\tilde{\rho}, \tilde{z})$.*

2. *(**Sufficiency**) Suppose the KKT condition and the second-order condition (SOC) hold at $(\tilde{\rho}, \tilde{z})$. Then, $(\tilde{\rho}, \tilde{z})$ is a locally optimal WC-solution.*

Lemma 1 suggests that the KKT condition is both necessary and sufficient for characterizing the locally optimal WC-solutions under some additional conditions. Collectively, Theorems 1 and 2 and Lemma 1 provide a rigorous way to characterize the local optimality of the WC-scalarized problem, which further enables us to characterize the Pareto stationarity of the ECMO problem. Further, by using the KKT condition of the WC-scalarized problem, we define the following KKT system:

**Definition 5** (KKT System). For ECMO with a given $\lambda \in \Delta_S^{++}$, we define the KKT system[3]:

$$\mathcal{K}(\rho, z, \omega, \nu, \lambda) = \begin{pmatrix} \sum_{s=1}^{S} \omega_s - 1 \\ \sum_{s=1}^{S} \omega_s \lambda_s \nabla f_s(z) + \sum_{i=1}^{q} \nu_i \nabla h_i(z) \\ h(z) \\ [\min\{\omega_s, \rho - \lambda_s f_s(z)\}]_{s \in [S]} \end{pmatrix},$$

where $\omega = (\omega_1, \ldots, \omega_S)^\top$, and $\nu = (\nu_1, \ldots, \nu_q)^\top$ are the Lagrange dual multipliers associated with inequality and equality constraints in WC, respectively.

Clearly, the KKT condition holds if and only if $\mathcal{K}(\rho, z, \omega, \nu, \lambda) = 0$. Also, we can measure how far a point deviates from the KKT condition, thereby quantifying its

---

[1]Without loss generality, if the original ECMO problem is non-degenerate, i.e., all $f_s(z)$ are bounded from below, we can add a sufficiently large constant to all $f_s(\cdot)$ to construct $S$ positive-valued functions. The Pareto front of the newly constructed problem has a one-to-one mapping with the Pareto front of the original problem.

[2]It is worth noting that there are multiple constraint qualifications (CQs), and all of them (including LICQ) guarantee the necessity in Lemma 1.

[3]The dual feasibility and complementary slackness are implied by the last term in the KKT system.

distance to optimality for the ECMO problems and providing a rigorous **convergence metric**. Specifically, according to Theorems 1 and 2 and Lemma 1, for any $\epsilon > 0$, we define a point $\tilde{z}$ to be an $\epsilon$-**Pareto stationary solution** of ECMO if and only if there exist some $\rho \in \mathbb{R}, \omega \in \mathbb{R}^S, \nu \in \mathbb{R}^q, \lambda \in \Delta_S^{++}$ such that $\|\mathcal{K}(\rho, z, \omega, \nu, \lambda)\|_2^2 \leq \epsilon$. This indicates that $\|\mathcal{K}(\rho, z, \omega, \nu, \lambda)\|_2^2$ can serve as a convergence metric for our ECMO algorithm design in the next section. More details about the KKT system are in Appendix F.

## 4. Algorithm Design for the ECMO Problem

In this section, we first present our proposed WC-Penalty algorithm for solving the ECMO problem. We then provide its finite-time convergence analysis results and discuss their further insights. Due to space limitation, all proofs for this section are relegated to Appendix D.

### 4.1. The WC-Penalty Algorithm

In Section 3, we have established the equivalence between the ECMO problem and its WC-scalarized problem. We have also characterized the Pareto stationarity of ECMO using the KKT system of the WC-scalarized problem, based on which we further established the KKT-based convergence metric to an $\epsilon$-Pareto stationary solution of the ECMO problem. Note that in the KKT-based Pareto stationarity convergence metric, the term $[\min\{\omega_s, \rho - \lambda_s f_s(z)\}]_{s \in [S]}$ is more difficult to control. This motivates us to reformulate the WC-scalarized problem by adding this term as an equality constraint with slack variables:

$$\min_{\rho, z, \delta} \rho \text{ s.t. } h_i(z) = 0, i = 1, \ldots, q,$$
$$\lambda_s f_s(z) + \delta_s = \rho, s \in [S], \ \delta_s \geq 0, s \in [S],$$

where $\delta := [\delta_1, \ldots, \delta_S]^\top \in \mathbb{R}^S$ contains all slack variables. Let $\mathcal{C} := \mathbb{R} \times \mathbb{R}^k \times \mathbb{R}_+^S$, where $\mathbb{R}_+^S = \{\delta \in \mathbb{R}^S : \delta \geq \mathbf{0}\}$. Then, the reformulated problem above can be viewed as an equality-constrained single-objective problem with a convex feasible region $\mathcal{C}$. To solve this reformulated problem, a natural idea is to incorporate all equality constraints as penalty terms in the objective function, which leads to the following formulation:

$$\min_\theta P(\theta) = \rho + \frac{u}{2} \sum_{i=1}^q h_i(z)^2 + \frac{v}{2} \sum_{s=1}^S (\lambda_s f_s(z) + \delta_s - \rho)^2 \quad (1)$$
$$\text{s.t. } \delta_s \geq 0, s \in [S],$$

where $u, v > 0$ are sufficiently large hyper-parameters to be chosen. For notational convenience, we let $\theta := [\rho^\top, z^\top, \delta^\top]^\top$, so that Eq. (1) can be written as $\min_{\theta \in \mathcal{C}} P(\theta)$.

Thanks to the convex and simple box constraints, one can solve Problem (1) using a projected gradient descent (GD)

---

**Algorithm 1** The WC-Penalty algorithm.

1: **Input:** Iteration rounds $T$, initialization $\theta_0 = (\rho_0, z_0, \delta_0) \in \mathcal{C}$, with $\rho_0 \geq 0$, and step-size $\eta$.
2: **for** $t = 0, 1, \ldots, T-1$ **do**
3:     Compute $\nabla P(\theta_t)$ according to Eq. (2).
4:     Update $\theta_{t+1} = \mathcal{P}_\mathcal{C}(\theta_t - \eta \nabla P(\theta_t))$.
5: **end for**

---

approach as shown in Algorithm 1. Specifically, in each iteration $t$, we compute the gradient $\nabla P(\theta_t)$, take a GD step with some step-size $\eta$, and then project the obtained solution back onto the feasible domain $\mathcal{C}$.

In Algorithm 1, we can compute the gradient $\nabla P(\theta) = [\nabla_\rho P(\theta)^\top, \nabla_z P(\theta)^\top, \nabla_\delta P(\theta)^\top]^\top$ as:

$$\nabla_\rho P(\theta) = 1 - v \sum_{s=1}^S (\lambda_s f_s(z) + \delta_s - \rho),$$
$$\nabla_z P(\theta) = u \sum_{i=1}^q h_i(z) \nabla h_i(z)$$
$$+ v \sum_{s=1}^S (\lambda_s f_s(z) + \delta_s - \rho) \lambda_s \nabla f_s(z), \quad (2)$$
$$\nabla_{\delta_s} P(\theta) = v(\lambda_s f_s(z) + \delta_s - \rho), \quad s \in [S].$$

**Remark 1.** We can generalize Algorithm 1 to stochastic ECMO problems, where the objectives and constraints are in the form of $f_s(z) = \mathbb{E}_\xi[f_s(z; \xi)], \forall s \in [S]$, and $h_i(z) = \mathbb{E}_\zeta[h_i(z; \zeta)], \forall i \in [q]$. The basic idea remains the same, with the key distinction being the use of stochastic gradients. Due to space limitation, we provide the stochastic WC-Penalty algorithm and its analysis in Appendix D.

**Remark 2.** Even though Algorithm 1 does not require the maintenance or updating of the *dual variables*, $\omega$ and $\nu$ do play an significant role in analyzing the convergence rate. As detailed in Appendix D, we select $\omega_{t,s} = v(\lambda_s f_s(z_t) + \delta_{t,s} - \rho_t)$, $\nu_{t,i} = u h_i(z_t)$, where $\omega_t = (\omega_{t,1}, \ldots, \omega_{t,S})^\top$, $\nu_t = (\nu_{t,1}, \ldots, \nu_{t,q})^\top$ at each iteration $t$ to control the KKT system and, in turn, to ensure the finite-time convergence.

### 4.2. Theoretical Convergence Analysis

To analyze the convergence of the proposed WC-Penalty algorithm, we first state several useful assumptions, and then establish the finite-time convergence rate guarantee and iteration complexity results for our WC-Penalty algorithm. Unless noted otherwise, we use $\|\cdot\|$ to denote the $\ell_2$-norm.

**Assumption 1** (Smoothness). There exist some constants $M, L > 0$ such that for any $z_1, z_2 \in \mathbb{R}^k$, and for any $s \in [S], i \in [q]$, we have:

$$|f_s(z_1) - f_s(z_2)| \leq M \|z_1 - z_2\|,$$
$$|h_i(z_1) - h_i(z_2)| \leq M \|z_1 - z_2\|,$$
$$\|\nabla f_s(z_1) - \nabla f_s(z_2)\| \leq L \|z_1 - z_2\|,$$
$$\|\nabla h_i(z_1) - \nabla h_i(z_2)\| \leq L \|z_1 - z_2\|.$$

Assumption 1 is standard and widely adopted in the literature (Ghadimi & Wang, 2018; Ji et al., 2021; Qiu et al., 2023; Lin et al., 2024). We also require that, there exists some $L_P > 0$ such that $\|\nabla P(\theta_1) - \nabla P(\theta_2)\| \leq L_P \|\theta_1 - \theta_2\|$, implying that $P(\theta)$ is $L_P$-smooth. However, $L_P = \Theta(u + v)$ could be large since the chosen penalty coefficients $u, v$ are typically large.

**Assumption 2** (LICQ of the WC Problem). *For any $z \in \mathbb{R}^k$,* we assume that linearly independent constraint qualification (LICQ) holds for the Problem (WC), i.e., the gradients $\{\nabla h_i(z), \nabla f_s(z)\}$ are linearly independent, where $i \in [q]$, and $s \in \{s \in [S] : \lambda_s f_s(z) = \rho\}$.

With LICQ for Problem (WC), $\{\nabla h_1(z), \ldots, \nabla h_q(z)\}$ is linearly independent, implying $\text{rank}(\nabla h(z)) = q$. As a result, $\nabla h(z)^\top \nabla h(z)$ is positive definite. This implies that the minimum *singular* value of $\nabla h(z)$, $\sigma_{\min}(\nabla h(z))$, can be lower bounded by some strictly positive $\sigma > 0$.

Also, without loss of generality, we suppose that $f_s(z) > 0, \forall s \in [S]$ (see the justification in Theorem 2) and that $\{z \in \mathbb{R}^k : h(z) = 0\}$ is nonempty, to ensure the ECMO problem is well-posed. We are now ready to present the main theoretical convergence rate result as follows.

**Theorem 3** (Finite-Time Convergence Rate of Algorithm 1). *Under Assumptions 1 and 2, for any preference $\lambda \in \Delta_S^{++}$, selecting $\eta = \Theta(T^{-\frac{1}{4}})$ and $u = v = \Theta(T^{\frac{1}{4}})$, Algorithm 1 achieves the following convergence result:* $\frac{1}{T} \sum_{t=0}^{T-1} \|\mathcal{K}(\rho_t, z_t, \omega_t, \nu_t, \lambda)\|^2 = \mathcal{O}(S/T^{\frac{1}{2}})$.

In addition to the finite-time convergence rate, *iteration complexity* also serves as another key metric for evaluating the efficiency of algorithms. Specifically, ensuring $\frac{1}{T} \sum_{t=0}^{T-1} \|\mathcal{K}(\rho_t, z_t, \omega_t, \nu_t, \lambda)\|^2 < \epsilon$ for any $\epsilon > 0$ indicates that the obtained sequence achieves an $\epsilon$-Pareto stationary solution for the ECMO problem. The iteration complexity result below immediately follows from Theorem 3:

**Corollary 1.** *To achieve an $\epsilon$-Pareto stationary solution for any $\epsilon > 0$, Algorithm 1 requires $\mathcal{O}(S^2 \epsilon^{-2})$ evaluations of $\nabla f_s(z)$ for each $s \in [S]$, and $\mathcal{O}(S^2 \epsilon^{-2})$ evaluations of $\nabla h_i(z)$ for $i \in [q]$.*

**Remark 3.** Theorem 3 is established in two key steps. In the first step, we consider the dynamics of $P(\theta_t)$ generated by Algorithm 1, which is designed to solve $\min_{\theta \in \mathcal{C}} P(\theta)$, and hence solving the WC-scalarized problem. In the second step, we judiciously select the parameters, which, according to Theorem 2 and Lemma 1, allow us to control each term in the KKT system defined in Definition 5. Collectively, these two key steps establish the theoretical guarantees for solving ECMO as shown in Figure 5. Due to space limitation, the proof of Theorem 3 is relegated to Appendix D.

**Remark 4.** To our knowledge, Theorem 3 establishes the first finite-time convergence guarantee in the literature of ECMO. This result ensures that Algorithm 1 can achieve

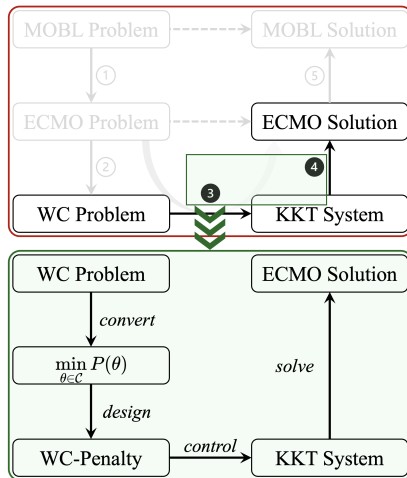

*Figure 5.* Steps to prove Theorem 3.

Pareto stationarity for any given preference weight vector $\lambda$. Moreover, according to the previous discussions on WC-scalarization, by varying $\lambda$ over $\Delta_S^{++}$, Algorithm 1 can systematically explore the entire Pareto stationary front.

## 5. Returning to MTBL Problems through the Lens of ECMO

Finally, we can easily solve the MTBL problem as a special case of the ECMO problem: we first specialize Eq. (1) in this scenario by splitting the variable $z$ explicitly into $x, y$:

$$\min_{\rho, x, y, \delta} P(\rho, x, y, \delta) = \rho + \frac{u}{2} \sum_{i=1}^q (\nabla_y g(x, y))_i^2$$
$$+ \frac{v}{2} \sum_{s=1}^S (\lambda_s f_s(x, y) + \delta_s - \rho)^2$$
$$\text{s.t. } \delta_s \geq 0, s \in [S].$$

For convenience, we still denote 1) the combined variable as $\theta = (\rho^\top, x^\top, y^\top, \delta^\top)^\top$, and 2) the feasible region as $\mathcal{C} = \mathbb{R} \times \mathbb{R}^p \times \mathbb{R}^q \times \mathbb{R}_+^S$. Under this change of variables, we can follow Algorithm 1 exactly to solve the MTBL problem, and the theoretical results in Section 4 naturally translate to the MTBL setting. Next, to validate the effectiveness of our proposed algorithm, we apply it to two MTBL tasks and present the corresponding numerical results.

### 5.1. Data Weighting for Multi-Objective RLHF Reward Model Training

**1) Experimental Setup:** The multi-objective data weighting task aims to determine optimal proportion in mixing training datasets for training a reward model to maximize multiple validation metrics in the Pareto sense. This task is important in reinforcement learning with human feedback (RLHF), where: 1) large-scale training data often has unknown origins, varied tendencies, and mixed qualities, and 2) human preferences (e.g., *helpfulness*, *verbosity*) may conflict with each other. Here, we train the reward model

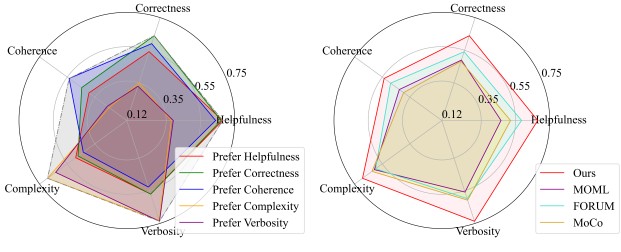

*(a)* Pareto exploration.      *(b)* Baseline comparison.

*Figure 6.* Data weighting for RLHF reward model training.

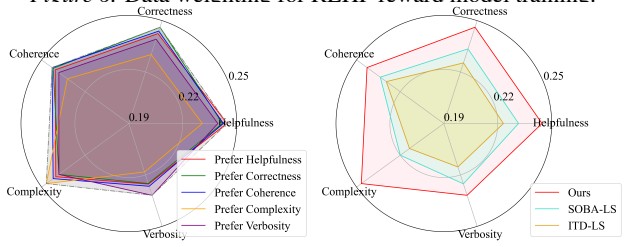

*(a)* Pareto exploration.      *(b)* Baseline comparison.

*Figure 7.* Data weighting task in LLM alignment.

for RLHF on the HelpSteer dataset (Wang et al., 2023), and consider all of the 5 provided criteria, *helpfulness*, *correctness*, *coherence*, *complexity*, and *verbosity*, as validation metrics. We evaluate three MTBL algorithms, MOML (Ye et al., 2021), MoCo (Fernando et al., 2023), FORUM (Ye et al., 2024), as our baselines.

**2) Experimental Results:** In Fig. 6, we set the preference vector $\lambda$ as $\lambda_s = 0.96$ for some $s \in [S]$ and $\lambda_{s'} = 0.01$, $\forall s' \neq s$, using $1/\text{loss}$ as our metric for each objective. As shown in Fig. 6a, by varying the preference vectors, Algorithm 1 can efficiently explore a diverse set of Pareto stationary solutions, enabling our algorithm to recover a large portion of the Pareto front. Moreover, Fig. 6b further demonstrates that our proposed algorithm outperforms existing methods in recovering the Pareto front, highlighting its effectiveness in Pareto front exploration. Due to space limitation, additional numerical results on convergence performances and comparisons with several bilevel algorithms using linear scalarization that demonstrate the strengths of our algorithm are relegated to Appendix E.1.

### 5.2. Data Weighting in Multi-Objective LLM Alignment

**1) Experimental Setup:** We consider the data weighting task on multi-objective LLM alignment, where the goal is to determine the proportion weights of dataset to minimize multiple human-aligned losses in validation. The dataset used here is still HelpSteer (Wang et al., 2023), which contains 5 potentially conflicting criteria, and the base LLM model is Llama-3.2-1B-Instruct (Meta, 2024). More setup details can be found in Appendix E.2.

**2) Experimental Results:** As shown in Fig. 7, we set the preference vector $\lambda$ as $\lambda_s = 0.96$ for some $s \in [S]$ and

*Table 1.* Hypervolume results in LLM alignment.

| Alg. | **Ours** | MOML | MoCo |
|------|----------|------|------|
| HV ($\uparrow$) | **2.47e-2** | 7.02e-7 | 1.66e-5 |

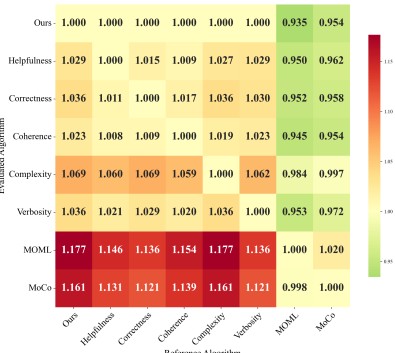

*Figure 8.* $\epsilon$-metric results in LLM alignment.

$\lambda_{s'} = 0.01$, $\forall s' \neq s$, using $1/\text{loss}$ as our metric for each objective. Fig. 7a shows that Alg. 1 is able to achieve Pareto stationary points with better performance on specific objectives when larger weights are assigned to them, again verifying the Pareto exploration capability of our algorithm. Moreover, Fig. 7b indicates that our algorithm outperforms two bilevel baselines adapted from (Ji et al., 2021; Dagréou et al., 2022), where we extend them with linear scalarization technique for solving MTBL problems.

In addition, Table 1 demonstrates that the hypervolume obtained by our method is significantly larger than that of the baselines (Ye et al., 2021; Fernando et al., 2023). Moreover, Fig. 8 further confirms that, in terms of $\epsilon$-metric: 1) our method consistently outperforms the baselines, and 2) with varying preference vectors, our method converges to the desired solutions. Due to space limitation, we relegate additional results to Appendix E.2. Additionally, we also conduct a multi-task meta-learning experiment in Appendix E.3 to further validate the efficacy of our algorithm.

## 6. Concluding Remarks

In this paper, we studied the LLGC-MTBL problems through the lens of ECMO. We first extended the notion of Pareto stationarity to ECMO and proposed a KKT-based Pareto stationarity convergence metric, based on which we developed a WC-Penalty algorithm for ECMO. Next, we established the finite-time convergence rate of our WC-Penalty algorithm. To our knowledge, this convergence result is the first of its kind in the literature. Lastly, we showed that our WC-Penalty algorithm can be used to solve the LLGC-MTBL problems not only with theoretical convergence guarantee but also effectively in practice as evidenced by our extensive numerical results.

## Acknowledgement

This work is supported in part by a Meta Research Award INB3267726, NSF grants CAREER CNS-2110259, CNS-2112471, ECCS-233104, DARPA YFA D24AP00265, DARPA HR0011-25-2-0019, and ONR grant N00014-24-1-2729.

## Impact Statement

This paper presents work whose goal is to advance the field of machine learning. There are many potential societal consequences of our work, none of which we feel must be specifically highlighted here.

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

# Appendix

## A. Notations

We summarize the notations throughout this paper in Table 2.

| Notations | Definitions |
|---|---|
| $P_{\mathcal{E}}(\cdot)$ | Projection the point onto a convex set $\mathcal{E}$ |
| $\mathcal{D}$ | Feasible region $\{z \in \mathbb{R}^k : h(z) = \mathbf{0}\}$ |
| $N_\delta(\tilde{z})$ | $\delta$-neighborhood of $\tilde{z}$, i.e., $\{z \in \mathcal{R}^k : \|z - \tilde{z}\|_2 \leq \delta\}$ |
| $\Delta_S^+$ | Simplex in $S$ dimension |
| $\Delta_S^{++}$ | Strictly positive simplex in $S$ dimension |
| $X_{\mathrm{P}}$ | The set of Pareto optimal points |
| $X_{\mathrm{WP}}$ | The set of weakly Pareto optimal points |
| $\{F(x) : x \in X_{\mathrm{P}}\}$ | Pareto Front |
| $\{F(x) : x \in X_{\mathrm{WP}}\}$ | Weak Pareto front |
| $\rho$ | Additional variable for WC problem |
| $\omega, \nu$ | Dual variables for KKT system |
| $\lambda$ | Preference vector |

*Table 2.* Summarized notation table in the paper.

## B. Additional Related Work on Closely Related Topics

In this section, we review existing literature in the areas of Multi-Objective Optimization (MOO), Bilevel Optimization (BLO), Multi-Task Bilevel Learning (MTBL), and Equality Constrained Multi-Objective (ECMO) problems. Notably, to put our work in comparative perspectives, we also provide the comparison of our approach with the existing MTBL methods in Table 3.

**Multi-Objective Optimization (MOO).** Research on MOO dates back to (Sawaragi et al., 1985), and continues to attract significant attention in recent years (Ehrgott, 2005; Chankong & Haimes, 2008; Hwang & Masud, 2012; Gunantara, 2018). Methods for unconstrained MOO can be broadly categorized into scalarization approaches and adaptive gradient methods. Scalarization approaches transform the MOO problems to single-objective problems. Among them, the most widely used are linear scalarization (Ehrgott, 2005; Lin et al., 2024; Qiu et al., 2024) and Weighted-Chebyshev method (Momma et al., 2022; Lin et al., 2024; Qiu et al., 2024). Adaptive gradient methods, on the other hand, aim to find Pareto optimal solutions through iterative updates and gradient descent schemes, and have been explored in works such as (Miettinen & Mäkelä, 1995; Désidéri, 2012; Mercier et al., 2018; Fernando et al., 2023; Chen et al., 2024a). The applications of MOO span a variety of domains, including but not limited to multi-task learning (Sener & Koltun, 2018; Lin et al., 2019; Momma et al., 2022), multi-objective training and clustering (Mossalam et al., 2016; Alok et al., 2015; González-Almagro et al., 2020), architecture search (Jin et al., 2007). Although these works extensively studied MOO literature, most results and techniques rely heavily on the absence of constraints. This leaves the foundation of ECMO problems still in its infancy.

**Bilevel Optimization (BLO).** BLO also has a long-standing history, with early foundational work such as (Bracken & McGill, 1973). In recent years, its importance has surged in machine learning, particularly in applications involving large-scale models and (LLMs), where variable coupling across different optimization levels demands sophisticated BLO frameworks (Chakraborty et al., 2023; Shen et al., 2024a;b). Over the past decade, significant progress has been made in the development of BLO methods (Zhang et al., 2024). Works like (Ghadimi & Wang, 2018; Arbel & Mairal, 2021; Ji et al., 2021; Dagréou et al., 2022) provided a wide range of techniques and paradigms. Moreover, (Tarzanagh et al., 2022; Huang et al., 2023; Qiu et al., 2023; Liu et al., 2023b) also extended BLO to federated learning, decentralized learning, etc. While the lower-level strongly convex (LLSC) assumption is quite restrictive, it is widely adopted in the aforementioned works. Although several recent efforts have been made to relax it by considering only convexity (LLGC) (Sabach & Shtern, 2017; Liu et al., 2023a; Cao et al., 2023; Jiang et al., 2023; Yao et al., 2024; Chen et al., 2024b; Lu & Mei, 2024; Qin et al., 2025; Jiang et al., 2025), such results remain confined to the basic BLO scenario only. Their applicability to general scenarios, such as federated BLO, decentralized BLO, and MTBL, remains largely unexplored, as the different setups and optimality

*Table 3.* Comparison of Different MTBL Algorithms.

| Algorithm | Scenario | Convergence | Exploration |
|---|---|---|---|
| gMOBA (Yang et al., 2024b) | Deterministic | Asymptotic | ✗ |
| MOML (Ye et al., 2021) | Deterministic | Asymptotic† | ✗ |
| FORUM (Ye et al., 2024) | Deterministic | $\mathcal{O}(ST^{-\frac{1}{4}})$‡ | ✗ |
| MoCo (Fernando et al., 2023) | Stochastic | $\mathcal{O}(ST^{-\frac{1}{2}})$ | ✗ |
| **WC-Penalty (This Work)** | Deterministic | $\mathcal{O}(ST^{-\frac{1}{2}})$ | ✓ |
| **WC-Penalty (This Work)** | Stochastic | $\mathcal{O}(ST^{-\frac{1}{2}})$ | ✓ |

†: A journal version provides a finite-time rate under a different metric. ‡: Note that even though the number of objectives $S$ is not explicitly stated in their main theorem, a closer examination of the proof reveals that the $S$ is hidden in the $\mathcal{O}(\cdot)$ notation implicitly.

evaluation metrics lead to distinct challenges and require specific methodologies.

**Multi-Task Bilevel Learning (MTBL).** MTBL problem has gained increasing attention in recent years (Ye et al., 2021; Gu et al., 2023; Fernando et al., 2023; Li et al., 2024; Wang et al., 2024; Yang et al., 2024b; Ye et al., 2024). It has received significant attention due to the rise of its wide range of applications, such as meta-learning, LLM alignment, and multi-domain training (Ye et al., 2021; Zhang et al., 2026; Kong et al., 2025; Dong et al., 2024; Wu et al., 2024; Liang et al., 2025). Compared to more mature literature on MOO and BLO, existing theoretical results for MTBL remain quite limiting. Among these works, (Yang et al., 2024b; Ye et al., 2021) demonstrate that their proposed algorithms converge asymptotically, without providing any finite-time convergence guarantees. In contrast, (Fernando et al., 2023; Ye et al., 2024) provide algorithms with a convergence rate of $\mathcal{O}(ST^{-\frac{1}{2}})$ and $\mathcal{O}(ST^{-\frac{1}{4}})$, respectively. Recently, (Zhang et al., 2026), for the first time in the literature, investigates the Pareto front exploration, yet their approach requires the restrictive LLSC condition. However, all of these works heavily depend on the LLSC condition: not only is the algorithmic framework built upon the LLSC condition, but the optimality criterion also relies on it. Therefore, this strong assumption significantly limits their applicability to complex real-world scenarios where this assumption is usually violated.

**Equality Constrained Multi-Objective (ECMO).** ECMO problems have found wide applications across various fields, including resource allocation, scheduling optimization, and path planning, just to name a few (Liang et al., 2022; Hao et al., 2024). The most closely related works on ECMO problems are (Cuate et al., 2020; García et al., 2021). Both studies propose algorithmic solutions for ECMO and conduct numerical experiments to validate their methods. However, neither provides any finite-time convergence guarantees, highlighting that the theoretical foundations for ECMO remain in their infancy. A closely related and important extension of ECMO is the Inequality Constrained Multi-Objective (ICMO) problem, where inequality constraints are also incorporated (Fan et al., 2017; Afshari et al., 2019; Liang et al., 2022; Hao et al., 2024). As with ECMO, the theoretical understanding of ICMO remains limited. Numerous heuristic algorithms have been proposed in the literature (Jimenez et al., 2002; Tanabe & Oyama, 2017; Fan et al., 2019; Yang et al., 2019; Ming et al., 2022; Belaiche et al., 2023; Li et al., 2023; Long et al., 2023; Yang et al., 2024a; Song et al., 2024; Qiao et al., 2024), offering a variety of algorithmic frameworks accompanied by experimental evaluations. However, these works do not establish convergence guarantees, underscoring the lack of rigorous theoretical foundations for ICMO (and ECMO) problems.

## C. Discussions and Proofs of Section 3

In this section, we first demonstrate that naively extending the definitions in unconstrained MOO is insufficient for characterizing Pareto stationarity in ECMO via two judicious constructed counterexamples. We then provide the proofs of Theorems 1 and 2 stated in Section 3.

### C.1. Discussion about Pareto Stationarity in ECMO

In unconstrained MOO problems, $\tilde{z}$ is a Pareto stationary point if and only if $\exists \alpha \in \Delta_S^+$ ($S$-simplex) such that, $(\nabla f_1(\tilde{z}), \dots, \nabla f_S(\tilde{z})) \alpha = \mathbf{0}$ (Sener & Koltun, 2018; Lin et al., 2024). Accordingly, for any $\epsilon > 0$, an $\epsilon$-Pareto stationary solution $\tilde{z}$ can be defined as $\|\nabla F(\tilde{z})\alpha\| \le \epsilon$ for some $\alpha \in \Delta_S^+$. Nevertheless, we now demonstrate that, this ($\epsilon$-)Pareto stationarity definition becomes irrational in ECMO problems.

By considering the KKT condition for each objective $f_s(z), s \in [S]$, we can construct Lagrangian as: $\mathbb{L}_s(z, v) = f_s(z) + v^\top h(z)$. Then, for each $s \in [S]$, the KKT condition can be written as: $\nabla f_s(z) + \nabla h(z)v = 0$ and $h(z) = 0$. Therefore, similar to the equivalent definition of Definition 3 shown in Section 3, we can define the following system to

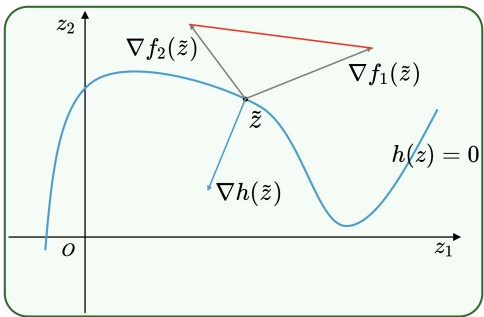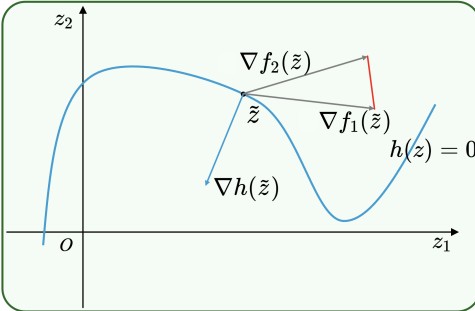

*Figure 9.* Pareto stationary and nonstationary examples in ECMO problems.

consider Pareto stationarity:

$$\mathrm{PS}(z, v, \alpha) = \left( \begin{pmatrix} \nabla \mathbb{L}_1(z, v) \\ h(z) \end{pmatrix}, \ldots, \begin{pmatrix} \nabla \mathbb{L}_S(z, v) \\ h(z) \end{pmatrix} \right) \alpha = \begin{pmatrix} \nabla F(z)\alpha + \nabla h(z)v \\ h(z) \end{pmatrix} = \mathbf{0},$$

where $\alpha \in \Delta_S^+$, and $v \in \mathbb{R}^q$. $\mathrm{PS}$ not only takes the feasible direction into account, but also enforces the feasibility directly. It precisely captures both the Pareto stationary and nonstationary scenarios depicted in Figure 9. To see this, $\alpha_1 \nabla f_1(\tilde{z}) + \alpha_2 \nabla f_2(\tilde{z})$ is represented by the red line. For the left example, where $\tilde{z}$ is Pareto stationary, we can select $\alpha$ such that $\nabla F(\tilde{z})\alpha$ is collinear with $\nabla h(\tilde{z})$, allowing the existence of $v \in \mathbb{R}$ to achieve $\mathrm{PS}(\tilde{z}, v, \alpha) = 0$. In contrast, for the right example, $v\nabla h(\tilde{z})$ does not lie in the convex hull of $\{\nabla f_1(\tilde{z}), \nabla f_2(\tilde{z})\}$ for any $v \in \mathbb{R}$, indicating the first term in $\mathrm{PS}$ can never achieve $\mathbf{0}$, correctly aligning with the Pareto nonstationarity of $\tilde{z}$.

However, we can construct some scenarios where 1) $\tilde{z}$ is Pareto stationary, i.e., no feasible movement can simultaneously improve, or at least not hurt, all objectives, but 2) $\mathrm{PS}(\tilde{z}, v, \alpha) \neq 0$ for any $\alpha \in \Delta_2^+$, and $v \in \mathbb{R}$. The following two concrete examples illustrate such cases, highlighting limitations of the $\mathrm{PS}$ formulation in fully capturing Pareto stationarity for ECMO problems.

*Example 1.* Consider a 1-dimensional bi-objective problem with 1 constraint as follows:

$$\min_z F(z)^\top = (-\frac{1}{2}z^2, -z) \qquad \text{s.t. } h(z) = \begin{cases} 0 & \text{if } -1 \leq z \leq 1 \\ (|z| - 1)^2 = 0 & \text{otherwise,} \end{cases}$$

Obviously, $\tilde{z} = 1$ is a Pareto stationary point. However, since $\nabla F(\tilde{z})\alpha = -1$ for any $\alpha \in \Delta_2^+$, and $\nabla h(\tilde{z}) = 0$, we know that $\mathrm{PS}(\tilde{z}, v, \alpha) = -1 \neq 0$.

*Example 2.* Although the previous example is carefully constructed, one might wonder whether the failure arises from the lack of second-order differentiability of $h(z)$? To address that, this example employs more general functions to refute this hypothesis, thereby indicating the intrinsic irrationality of $\mathrm{PS}$ system itself. To this end, we consider a 3-dimensional bi-objective problem with 2 constraints as follows:

$$\min_z F(z)^\top = (z_1 + z_2, z_1 - z_2) \qquad \text{s.t. } h(z) = \begin{cases} z_1^2 + z_3^2 - 1 = 0, \\ z_3 - 1 = 0. \end{cases}$$

In this example, the feasible region is given by $\mathcal{D} = \{0\} \times \mathbb{R} \times \{1\}$. We consider $\tilde{z} = (0, 0, 1)^\top \in \mathcal{D}$. Obviously, $\tilde{z}$ is Pareto stationary. However, $(\nabla F(\tilde{z})\alpha)_1 = 1, \forall \alpha \in \Delta_2^+$ and $(\nabla h(\tilde{z})v)_1 = 0, \forall v \in \mathbb{R}^2$ implies that $\mathrm{PS}(\tilde{z}, v, \alpha) \neq 0$ for any $\alpha \in \Delta_2^+$, and $v \in \mathbb{R}^2$. This, again, contradicts the idea that $\mathrm{PS}$ characterizes Pareto stationarity.

These counterexamples motivate us to consider adopting constraint qualification conditions in Section 4, which are not only important for characterizing the Pareto stationarity, but also critical for avoiding corner cases caused by a degenerate Jacobian matrix.

## C.2. Proof of Theorem 1

*Proof.* **1)** We first assume that $\tilde{z}$ is locally weakly Pareto optimal in $\mathcal{D}(\tilde{z}, \delta)$ for some positive $\delta$. Suppose it's not Pareto stationary, then, there exists some feasible direction $d \in \mathbb{R}^k$, such that $\nabla f_s(\tilde{z})^\top d < 0, \forall s \in [S]$. We can select a positive

and sufficiently small $\epsilon$, such that $\hat{z} = \tilde{z} + \epsilon d \in \mathcal{D}(\tilde{z}, \delta)$, where $\epsilon > 0$. Then, we have:

$$f_s(\hat{z}) = f_s(\tilde{z}) + \epsilon \nabla f_s(\tilde{z})^\top d + o(\epsilon), \forall s \in [S],$$

$$\implies \frac{f_s(\hat{z}) - f_s(\tilde{z})}{\epsilon} = \nabla f_s(\tilde{z})^\top d + \frac{o(\epsilon)}{\epsilon} < 0, \forall s \in [S],$$

where $o(\cdot)$ denotes the higher-order terms. Thus, $f_s(\hat{z}) < f_s(\tilde{z}), \forall s \in [S]$. This is contradicted with the definition of locally weak Pareto optimality. Hence, $\tilde{z}$ is Pareto stationary.

**2)** On the other hand, we assume $\tilde{z}$ is Pareto stationary. Suppose it's not locally weakly Pareto optimal in $\mathcal{D}(\tilde{z}, \delta)$ for all positive $\delta$. Then, for any $\delta > 0$, there exists some $\hat{z} = \tilde{z} + \epsilon d \in \mathcal{D}(\tilde{z}, \delta)$, where $\epsilon > 0, d \neq \mathbf{0}$ is a feasible direction, such that $f_s(\hat{z}) < f_s(\tilde{z}), \forall s \in [S]$. Then, we have:

$$f_s(\hat{z}) = f_s(\tilde{z}) + \epsilon \nabla f_s(\tilde{z})^\top d + o(\epsilon) < f_s(\tilde{z}), \forall s \in [S] \implies \nabla f_s(\tilde{z})^\top d \leq 0, \forall s \in [S].$$

Since $\tilde{z}$ satisfies that, for any $s \in [S]$, there exists $\mu_s$ such that 1) $\nabla f_s(\tilde{z}) + \sum_{i=1}^q \mu_{s,i} \nabla h_i(\tilde{z}) = 0$, and 2) $\sum_{i=1}^q \mu_{s,i} \nabla h_i(\tilde{z})^\top d \neq 0$, we have:

$$\nabla f_s(\tilde{z})^\top d = -\sum_{i=1}^q \mu_{s,i} \nabla h_i(\tilde{z})^\top d \neq 0,$$

for any $s \in [S]$. Therefore, $\nabla f_s(\tilde{z})^\top d < 0, \forall s \in [S]$. This contradicts with the Pareto stationarity of $\tilde{z}$. Hence, $\tilde{z}$ is locally weakly Pareto optimal. $\square$

### C.3. Proof of Theorem 2

*Proof.* **1)** We assume that $(\rho, \tilde{z})$ is a local solution of WC for some $\lambda \in \Delta_S^{++}$, then there exists some $\delta > 0$, such that $(\rho, \tilde{z})$ is the minimizer of WC in $\mathcal{D}(\tilde{z}, \delta)$. Therefore, according to the definition of $\ell_\infty$-norm operation in WC, we have:

$$\max_s \lambda_s f_s(\tilde{z}) \leq \max_s \lambda_s f_s(z), \forall z \in \mathcal{D}(\tilde{z}, \delta).$$

Suppose $\tilde{z}$ is not a locally weak Pareto optimal solution of ECMO. Then, there exists some $\hat{z} \in \mathcal{D}(\tilde{z}, \delta)$, such that $f_s(\hat{z}) < f_s(\tilde{z}), \forall s \in [S]$. Therefore, we have $\lambda_s f_s(\hat{z}) < \lambda_s f_s(\tilde{z}), \forall s \in [S]$, thus:

$$\max_s \lambda_s f_s(\hat{z}) < \max_s \lambda_s f_s(\tilde{z}),$$

which leads to contradiction. Thus, $\tilde{z}$ is a locally weak Pareto optimal solution of ECMO.

**2)** Conversely, we assume that $\tilde{z}$ is a locally weak Pareto optimal solution of ECMO, then there exists some $\delta > 0$, such that $f_s(\tilde{z}) < f_s(z), \forall s \in [S], z \in \mathcal{D}(\tilde{z}, \delta)$. We set $\lambda$ as follows:

$$\lambda_s = \frac{(f_s(\tilde{z}))^{-1}}{\sum_{s'} (f_{s'}(\tilde{z}))^{-1}},$$

which implies $\lambda \in \Delta_S^{++}$ and:

$$\|\lambda \odot F(\tilde{z})\|_\infty = \frac{1}{\sum_{s'} (f_{s'}(\tilde{z}))^{-1}} = \lambda_s f_s(\tilde{z}), \forall s \in [S].$$

Suppose $(\rho, \tilde{z})$ is not a local solution of WC for any $\rho \in \mathbb{R}$. Then, there exists some $\hat{z} \in \mathcal{D}(\tilde{z}, \delta)$, satisfying:

$$\max_s \lambda_s f_s(\hat{z}) < \max_s \lambda_s f_s(\tilde{z}) = \frac{1}{\sum_{s'} (f_{s'}(\tilde{z}))^{-1}}.$$

Therefore, $\lambda_s f_s(\hat{z}) < \lambda_s f_s(\tilde{z}), \forall s \in [S]$. Since $\lambda$ is positive, we know $f_s(\hat{z}) < f_s(\tilde{z}), \forall s \in [S]$, which contradicts with the assumption. Thus, $\tilde{z}$ is a local solution of WC. $\square$

### C.4. KKT Condition of WC

For completeness, we state the KKT condition of WC in this section. To begin with, the Lagrangian of WC is:

$$\mathbb{L}(\rho, z, \omega, \nu, \lambda) = \rho + \sum_{s=1}^{S} \omega_s(\lambda_s f_s(z) - \rho) + \sum_{i=1}^{q} \nu_i h_i(z),$$

where $\omega = (\omega_1, \ldots, \omega_S)^\top$, and $\nu = (\nu_1, \ldots, \nu_q)^\top$ are the multipliers associated with inequality and equality constraints in WC, respectively. For a point $(\rho, z)$, its KKT condition contains four parts: stationarity, primal feasibility, dual feasibility, and complementary slackness. The stationary condition requires the gradients of $\rho$ and $z$ vanish:

$$1 - \sum_{s=1}^{S} \omega_s = 0, \qquad \sum_{s=1}^{S} \omega_s \lambda_s \nabla f_s(z) + \sum_{i=1}^{q} \nu_i \nabla h_i(z) = 0.$$

The primal and dual feasibility requires 1) the constraints are satisfied, and 2) the multiplier associated with inequality is positive:

$$h_i(z) = 0, i = 1, \ldots, m, \quad \lambda_s f_s(z) - \rho \leq 0, s = 1, \ldots, S, \quad \omega_s \geq 0, s = 1, \ldots, S.$$

In the end, the complementary slackness condition is:

$$\omega_s(\lambda_s f_s(z) - \rho) = 0, s = 1, \ldots, S.$$

## D. Additional Results and Proofs of Section 4

In this section, we first propose and analyze a linear scalarization algorithm for convex ECMO problems. After that, we prove Theorem 3 stated in Section 4. Finally, we provide the stochastic version of WC-Penalty algorithm and the theoretical analysis on its finite-time convergence rate.

### D.1. Linear Scalarization Method for Convex ECMO Problems

For the special case of convex ECMO problems, we also propose a linear scalarization (LS)-based algorithm along with its finite-time convergence guarantee. In this subsection, we first introduce Linear Scalarization method, and propose a simple algorithm for convex ECMO problems along with its performance guarantee. We then prove this theoretical result, and clarify how it relates to the KKT system and ECMO problems.

**Linear Scalarization (LS)**, or weighted sum method, is one of the most straightforward MOO methods. Intuitively, we assign a weight to each of the objective function, and minimize their weighted sum, i.e., solve a corresponding single-objective problem. For ECMO problems, LS can be represented as:

$$\min_{z \in \mathbb{R}^k} \sum_{s=1}^{S} \lambda_s f_s(z)$$
$$\text{s.t. } h_i(z) = 0, i = 1, \ldots, q,$$

where $\lambda \in \Delta_S^+$ is the given preference vector. While LS is extremely simple, it cannot, in general, recover the entire weak Pareto front unless all objective functions are convex and the feasible region is a convex set (Ehrgott, 2005). This suggests that, LS is not sufficient to generally solve the ECMO problems, and alternative techniques are needed to handle such general (nonconvex) cases.

Although LS method can only recover the entire weak Pareto front in some special cases, we can still apply this simple method to solve the convex ECMO problems. Specifically, in this subsection, we assume that upper level objective functions $f_1(z), \ldots, f_S(z)$ are convex functions, and lower level constraints $h_1(z), \ldots, h_q(z)$ are affine functions. Then, LS method transforms ECMO into a corresponding single-objective convex problem as follows:

$$\min_{z \in \mathbb{R}^k} \mathcal{L}_\lambda(z) = \sum_{s=1}^{S} \lambda_s f_s(z) \tag{3}$$
$$\text{s.t. } Az = b,$$

---

**Algorithm 2** LS Algorithm

---

1: **Input:** Iteration rounds $T$, initialization $z_0 \in \mathcal{D}$, and step-size $\eta$.
2: **for** $t = 0, 1, \ldots, T - 1$ **do**
3:    Compute $z_t^+ = z_t - \eta \nabla \mathcal{L}_\lambda(z_t)$.
4:    Update $z_{t+1} = \mathcal{P}_\mathcal{D}(z_t^+)$.
5: **end for**

---

where $\lambda \in \Delta_S^+$, $A \in \mathbb{R}^{q \times k}$, $b \in \mathbb{R}^q$ and $\text{rank}(A) = m$. We denote the feasible set as $\mathcal{D} := \{z \in \mathbb{R}^k : Az = b\}$, which is a closed and convex set since it's the intersection of $2m$ half-spaces.

Generally speaking, Algorithm 2 follows a simple projected gradient descent paradigm, where the convexity of the feasible region allows well-defined projection operation. Specifically, we denote the projection as $\mathcal{P}_\mathcal{E}(z) = \arg \min_{z' \in \mathcal{E}} \|z - z'\|_2^2$, where $\mathcal{E}$ can be any convex set. In each step $t$, we compute the gradient of $\mathcal{L}_\lambda(z_t)$, update $z_t^+$ according to gradient descent method, and project the obtained $z_t^+$ back to the feasible region $\mathcal{D}$ to get $z_{t+1}$. As shown later, this extremely simple method is effective in addressing the convex ECMO problem.

Now, we are ready to state the theoretical results for Algorithm 2.

**Theorem 4** (Finite-Time Convergence Rate of Algorithm 2). *Suppose that $f_1(z), \ldots, f_S(z)$ are convex functions, and $h_1(z), \ldots, h_q(z)$ are affine functions. Under Assumption 1, for any preference $\lambda \in \Delta_S^+$, selecting $\eta = \frac{1}{L}$, Algorithm 2 has the following convergence result:*

$$\mathcal{L}_\lambda(z_T) - \mathcal{L}_\lambda(z^*) \leq \frac{L}{2T} \|z_0 - z^*\|_2^2,$$

*where $z^*$ is the solution (global minimizer) of Equation (3).*

*Proof.* On the one hand, since $f_1(z), \ldots, f_S(z)$ are convex and $L$-smooth, so is $\mathcal{L}_\lambda$ for any $\lambda \in \Delta_S^+$. According to the descent lemma, we have:

$$\mathcal{L}_\lambda(z_{t+1}) \leq \mathcal{L}_\lambda(z_t) + \nabla \mathcal{L}_\lambda(z_t)^\top (z_{t+1} - z_t) + \frac{L}{2} \|z_{t+1} - z_t\|_2^2, \tag{4}$$

where $t = 0, 1, \ldots, T - 1$.

On the other hand, due to the convexity of $\mathcal{D}$, we have the following result:

$$
\begin{aligned}
&\|z_{t+1} - z^*\|_2^2 \\
\leq & \|z_t^+ - z^*\|_2^2 - \|z_{t+1} - z_t^+\|_2^2 \\
= & \|z_t - \eta \nabla \mathcal{L}_\lambda(z_t) - z^*\|_2^2 - \|z_{t+1} - z_t + \eta \nabla \mathcal{L}_\lambda(z_t)\|_2^2 \\
= & \|z_t - z^*\|_2^2 - 2\eta \nabla \mathcal{L}_\lambda(z_t)^\top (z_t - z^*) - \|z_{t+1} - z_t\|_2^2 - 2\eta \nabla \mathcal{L}_\lambda(z_t)^\top (z_{t+1} - z_t) \\
= & \|z_t - z^*\|_2^2 - \frac{2}{L} \nabla \mathcal{L}_\lambda(z_t)^\top (z_t - z^*) - \|z_{t+1} - z_t\|_2^2 - \frac{2}{L} \nabla \mathcal{L}_\lambda(z_t)^\top (z_{t+1} - z_t),
\end{aligned}
\tag{5}
$$

where the first inequality is derived from the convexity of $\mathcal{D}$:

$$
\begin{aligned}
\|z_t^+ - z^*\|_2^2 &= \|z_t^+ - z_{t+1}\|_2^2 + \|z_{t+1} - z^*\|_2^2 + 2\langle z_t^+ - z_{t+1}, z_{t+1} - z^* \rangle \\
&\geq \|z_t^+ - z_{t+1}\|_2^2 + \|z_{t+1} - z^*\|_2^2.
\end{aligned}
$$

Combining Equations (4) and (5), we have the following result given the convexity of $\mathcal{L}_\lambda$:

$$
\begin{aligned}
\mathcal{L}_\lambda(z_{t+1}) &\leq \mathcal{L}_\lambda(z_t) + \frac{L}{2} \left( \|z_t - z^*\|_2^2 - \|z_{t+1} - z^*\|_2^2 \right) - \nabla \mathcal{L}_\lambda(z_t)^\top (z_t - z^*) \\
&\leq \mathcal{L}_\lambda(z^*) + \frac{L}{2} \left( \|z_t - z^*\|_2^2 - \|z_{t+1} - z^*\|_2^2 \right),
\end{aligned}
$$

which implies

$$\mathcal{L}_\lambda(z_T) - \mathcal{L}_\lambda(z^*) \leq \frac{1}{T} \sum_{t=0}^{T-1} (\mathcal{L}_\lambda(z_{t+1}) - \mathcal{L}_\lambda(z^*))$$
$$\leq \frac{L}{2T} \left( \|z_0 - z^*\|_2^2 - \|z_T - z^*\|_2^2 \right)$$
$$\leq \frac{L}{2T} \|z_0 - z^*\|_2^2,$$

where the first inequality is due to the decreasing property, i.e., for $t = 0, 1, \ldots, T-1$, we have:

$$\mathcal{L}_\lambda(z_{t+1}) \leq \mathcal{L}_\lambda(z_t) + \nabla \mathcal{L}_\lambda(z_t)^\top (z_{t+1} - z_t) + \frac{L}{2} \|z_{t+1} - z_t\|_2^2$$
$$= \mathcal{L}_\lambda(z_t) - \frac{1}{L} \|\nabla \mathcal{L}_\lambda(z_t)\|_2^2 + \frac{1}{2L} \|\nabla \mathcal{L}_\lambda(z_t)\|_2^2$$
$$\leq \mathcal{L}_\lambda(z_t).$$

$\square$

Theorem 4 illustrates that Algorithm 2 has a convergence rate of $\mathcal{O}(T^{-1})$ to global optima for Equation (3). This reveals that we can recover the whole Pareto front of the original ECMO problem by traversing $\lambda$ over the simplex $\Delta_S^+$. Next, we show that this result can also be interpreted through the lens of the KKT system, providing consistency with the analysis of Algorithm 1.

Theorem 4 demonstrates that the sequence generated by Algorithm 2 converges to the global optimum of Equation (3). Since it's a convex problem without inequality constraints, we know that its KKT condition holds at some feasible point $\tilde{z}$ if and only if $\tilde{z}$ is the solution of Equation (3) (and the solution is the global minimizer due to the convexity of the problem). Therefore, we can establish the KKT system $\mathcal{K}(z, v, \lambda)$ as follows:

$$\mathcal{K}(z, v, \lambda) = \begin{pmatrix} \nabla F(z)\lambda + \nabla h(z)v \\ h(z) \end{pmatrix}_{(k+m) \times 1} = \begin{pmatrix} \nabla F(z)\lambda + A^\top v \\ Az - b \end{pmatrix}_{(k+m) \times 1},$$

where $\lambda \in \Delta_S^+$, and $v \in \mathbb{R}^q$. While this KKT system for Equation (3) is quite different from the one defined in Definition 5, it completely characterizes the optimality of any point $z$. In other words, given some $\lambda \in \Delta_S^+$, $\tilde{z} \in \mathcal{D}$ is the optimal point for Equation (3) if and only if $\mathcal{K}(\tilde{z}, v, \lambda) = 0$ for some $v \in \mathbb{R}^q$.

The following lemma ensures the optimality characterized by the KKT system can be adopted for evaluating the optimality of ECMO problems:

**Lemma 2** ((Ehrgott, 2005)). *Suppose $f_s(z), \forall s \in [S]$ are convex, and nonempty set $\mathcal{D}$ is convex and closed. Then, $\{z \in \mathcal{D} : z = \arg\min_{z'} \mathcal{L}_\lambda(z'), \forall \lambda \in \Delta_S^+\}$ is exactly the weak Pareto front of ECMO.*

Now, we are ready to bridge Theorem 4 with ECMO problems. According to Algorithm 2, $z_t$ is feasible for any $t$. Hence, for any $\lambda \in \Delta_S^+$, we have:

$$\min_v \|\mathcal{K}(z_T, v, \lambda)\|_2^2 \leq \|\mathcal{K}(z_T, 0, \lambda)\|_2^2 = \|\mathcal{L}_\lambda(z_T)\|_2^2.$$

By Theorem 4, we have:

$$\|\mathcal{L}_\lambda(z_T)\|_2^2 \leq 2L(\mathcal{L}_\lambda(z_T) - \mathcal{L}_\lambda(z^*)) \leq \frac{L^2}{T} \|z_0 - z^*\|_2^2.$$

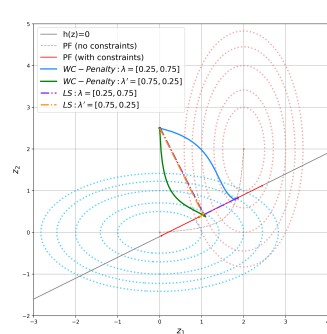

Therefore, we obtain $\min_v \|\mathcal{K}(z_T, v, \lambda)\|_2^2 \leq \frac{L^2}{T} \|z_0 - z^*\|_2^2$. According to the argument about KKT system and Lemma 2, we know that Algorithm 2 converges to weakly Pareto optimal solutions at a rate of $\mathcal{O}(T^{-1})$. In addition, we can also traverse $\lambda$ over $\Delta_S^+$ to let Algorithm 2 reconstruct the entire weak Pareto front.

To give a more concrete example, we provide a concrete example to show the performance of our proposed algorithms. We consider a bi-objective problem, with objective functions $f_1(z) = z_1^2 + 4z_2^2$ and $f_2(z) = 4(z_1 - 2)^2 + (z_2 - 2)^2$, and constraint $h(z) = 0.5z_1 - z_2 - 0.1 = 0$. Besides, we consider two preference vectors $\lambda = (0.25, 0.75)^\top$ and

*Figure 10.* A toy example.

$\lambda' = (0.75, 0.25)^\top$. Figure 10 demonstrates the convergence performances of Algorithms 1 and 2. Specifically, the *deep-sky-blue* and *light-coral* dashed curves are contour plots of two objective functions $f_1(z)$ and $f_2(z)$. The *gray* line is the equality constraint $h(z) = 0$, and the *red* part on it is the Pareto front. The *rosy-brown* curve is the Pareto front when no constraints are included. The *black* point is the initial point for all sequences. Four curves in *blue*, *green*, *purple*, and *orange* are the sequences of proposed algorithms under different preference vectors.

We can clearly observe in Figure 10 that all of the four sequences converge to the Pareto front. Moreover, under the guidance of different preference vectors, the convergence points show distinct directional tendencies along the front.

### D.2. Proof of Theorem 3

*Proof.* We prove Theorem 3 in three steps as follows. To begin with, we first denote the update direction $d_t = (d_{t,\rho}^\top, d_{t,z}^\top, d_{t,\delta}^\top)^\top$ by $\theta_{t+1} = \theta_t - \eta d_t, \forall t = 0, \dots, T-1$.

**Step A: General Control.**

According to Assumption 1 and the descent lemma, for $t = 0, \dots, T-1$, we have:

$$P(\theta_{t+1}) \le P(\theta_t) + \langle \nabla P(\theta_t), \theta_{t+1} - \theta_t \rangle + \frac{L_P}{2} \|\theta_{t+1} - \theta_t\|^2$$

$$= P(\theta_t) - \eta \langle \nabla P(\theta_t), d_t \rangle + \frac{L_P \eta^2}{2} \|d_t\|^2.$$

The property of projection $\langle \mathcal{P}_\mathcal{C}(\theta_1) - \mathcal{P}_\mathcal{C}(\theta_2), \theta_1 - \theta_2 \rangle \ge \|\mathcal{P}_\mathcal{C}(\theta_1) - \mathcal{P}_\mathcal{C}(\theta_2)\|^2$ implies:

$$\langle \theta_{t+1} - \theta_t, (\theta_t - \eta \nabla P(\theta_t)) - \theta_t \rangle \ge \|\theta_{t+1} - \theta_t\|^2,$$
$$\implies \langle -\eta d_t, -\eta \nabla P(\theta_t) \rangle \ge \eta^2 \|d_t\|^2,$$
$$\implies \|d_t\|^2 \le \langle \nabla P(\theta_t), d_t \rangle.$$

Therefore, we have:

$$P(\theta_{t+1}) \le P(\theta_t) + \langle \nabla P(\theta_t), \theta_{t+1} - \theta_t \rangle + \frac{L_P}{2} \|\theta_{t+1} - \theta_t\|^2$$

$$= P(\theta_t) - \eta \|d_t\|^2 + \frac{L_P \eta^2}{2} \|d_t\|^2,$$

which implies:

$$\eta \left(1 - \frac{L_P \eta}{2}\right) \|d_t\|^2 \le P(\theta_t) - P(\theta_{t+1}).$$

Telescoping from $t = 0$ to $T-1$, we obtain:

$$\eta \left(1 - \frac{L_P \eta}{2}\right) \cdot \frac{1}{T} \sum_{t=0}^{T-1} \|d_t\|^2 \le \frac{1}{T} \left(P(\theta_0) - P(\theta_T)\right). \tag{6}$$

**Step B: KKT system.**

In this step, we control each term in the KKT system defined in Definition 5.

**Step B.1: Stationarity Terms.** According to Equation (6), we know that all of $d_{t,\rho}$, $d_{t,z}$, and $d_{t,\delta}$ can be well controlled, since $\|d_t\|^2 = \|d_{t,\rho}\|^2 + \|d_{t,z}\|^2 + \|d_{t,\delta}\|^2$. We first note the expression of $d_{t,z}$ as follows:

$$d_{t,z} = u \sum_{i=1}^{q} h_i(z_t) \nabla h_i(z_t) + v \sum_{s=1}^{S} (\lambda_s f_s(z_t) + \delta_{t,s} - \rho_t) \lambda_s \nabla f_s(z_t),$$

where $\delta_{t,s}$ it the $s$-th element of $\delta_t$. Then, to select $\omega$ and $\nu$ defined in Definition 5, we set $\omega_{t,s} = v(\lambda_s f_s(z_t) + \delta_{t,s} - \rho_t)$, $\nu_{t,i} = u h_i(z_t)$ to be the dual variables at iteration $t$, where $\omega_t = (\omega_{t,1}, \dots, \omega_{t,S})^\top$, $\nu_t = (\nu_{t,1}, \dots, \nu_{t,q})^\top$. Therefore, since

$\|d_{t,z}\|^2$ is controlled by $\|d_t\|^2$, the stationarity term for $z_t$ can also be well characterized. In addition, by the selection of $\omega_t$, we know:

$$1 - \sum_{s=1}^S \omega_{t,s} = 1 - \sum_{s=1}^S v(\lambda_s f_s(z_t) + \delta_{t,s} - \rho_t) = d_{t,\rho},$$

implying that the stationarity term for $\rho_t$ in Definition 5 is also controlled.

**Step B.2: Primal Feasibility Term.** We next consider the primal feasibility term, i.e., $h(z_t)$. We first introduce the following notation for convenience: $c_{t,s} = \lambda_s f_s(z_t) + \delta_{t,s} - \rho_t$. Then, according to the update direction for $z$, we have:

$$d_{t,z} = u\nabla h(z_t)h(z_t) + v\sum_{s=1}^S c_{t,s}\lambda_s \nabla f_s(z_t),$$

which, along with Assumption 2, implies:

$$u\sigma\|h(z_t)\| \le \|u\nabla h(z_t)h(z_t)\| = \|d_{t,z} - v\sum_{s=1}^S c_{t,s}\lambda_s \nabla f_s(z_t)\| \le \|d_{t,z}\| + vM\sum_{s=1}^S |c_{t,s}|,$$

where the last term can be derived as follows:

$$v\|\sum_{s=1}^S c_{t,s}\lambda_s \nabla f_s(z_t)\| \le v\sum_{s=1}^S |c_{t,s}| \cdot \|\lambda_s \nabla f_s(z_t)\| \le vM\sum_{s=1}^S |c_{t,s}|.$$

To further control $\sum_{s=1}^S |c_{t,s}|$, we consider the following two index sets:

$$\mathcal{I}_t = \{s \in [S] : vc_{t,s} \le \frac{\delta_{t,s}}{\eta}\}, \quad \mathcal{J}_t = \{s \in [S] : vc_{t,s} > \frac{\delta_{t,s}}{\eta}\}.$$

If $s \in \mathcal{I}_t$, then the corresponding component $\delta_{t,s}$ is not projected in step $t$, indicating $(d_{t,\delta})_s = \nabla_{\delta_s} P(\theta_t)$. By Cauchy–Schwarz inequality, we get:

$$\sum_{s\in\mathcal{I}_t} |c_{t,s}| \le \sqrt{|\mathcal{I}_t|}\frac{\|d_{t,\delta}\|}{v}.$$

If $s \in \mathcal{J}_t$, then 1) the corresponding component is projected in step $t$, 2) $c_{t,s}$ is nonnegative (since $\delta_{t,s} \ge 0$). Then, we have:

$$\begin{aligned}
\sum_{s\in\mathcal{J}_t} |c_{t,s}| &= \sum_{s\in\mathcal{J}_t} c_{t,s} \\
&= \sum_{s\in[S]} c_{t,s} - \sum_{s\in\mathcal{I}_t} c_{t,s} \\
&\le |\sum_{s\in[S]} c_{t,s}| + \sum_{s\in\mathcal{I}_t} |c_{t,s}| \\
&\le \frac{|d_{t,\rho}| + 1}{v} + \sqrt{|\mathcal{I}_t|}\frac{\|d_{t,\delta}\|}{v} \\
&\le \frac{2\sqrt{S}\|d_t\| + 1}{v},
\end{aligned}$$

where the second last inequality is due to the definition of $d_{t,\rho}$. Combining the aforementioned results together, we can obtain:

$$u\sigma\|h(z_t)\| \le \|d_{t,z}\| + vM\frac{3\sqrt{S}\|d_t\| + 1}{v} \le (3\sqrt{S}M + 1)\|d_t\| + M,$$

$$\implies \|h(z_t)\| \le \frac{3\sqrt{S}M + 1}{u\sigma}\|d_t\| + \frac{M}{u\sigma},$$

$$\implies \frac{1}{T}\sum_{t=0}^{T-1} \|h(z_t)\|^2 \le \frac{2(3\sqrt{S}M + 1)^2}{u^2\sigma^2} \cdot \frac{1}{T}\sum_{t=0}^{T-1} \|d_t\|^2 + \frac{2M^2}{u^2\sigma^2}.$$

We can select $u$ to be sufficiently large such that the coefficient of the first term $\frac{1}{T}\sum_{t=0}^{T-1}\|d_t\|^2$ is smaller than $\frac{1}{2}$.

**Step B.3: Dual Feasibility and Complementary Slackness Term.** Now, we consider the last term in the KKT system: $r_{t,s} = \min\{\omega_{t,s}, \rho_t - \lambda_s f_s(z_t)\}, \forall s \in [S]$. We denote $a_{t,s} = \rho_t - \lambda_s f_s(z_t)$, $b_{t,s} = \omega_{t,s} = v(\lambda_s f_s(z_t) + \delta_{t,s} - \rho_t)$ for convenience. We note that $b_{t,s} = v(\delta_{t,s} - a_{t,s})$. Besides, we also note the following fact:

$$
\begin{aligned}
(d_{t,\delta})_s &= \frac{1}{\eta}(\delta_{t,s} - \delta_{t+1,s}) \\
&= \frac{1}{\eta}(\delta_{t,s} - \max\{\delta_{t,s} - \eta\nabla_{\delta_s}P(\theta_t), 0\}) \\
&= \min\{b_{t,s}, \frac{\delta_{t,s}}{\eta}\},
\end{aligned}
$$

where the second equality is due to the projection operation. Now, we discuss $r_{t,s}^2$ in two cases.

In the first case, we suppose that $s \in \mathcal{I}_t$, i.e., $b_{t,s} \le \frac{\delta_{t,s}}{\eta}$. If $b_{t,s} \le a_{t,s}$, then $r_{t,s}^2 = b_{t,s}^2$. If $b_{t,s} > a_{t,s}$, then, combining with $b_{t,s} = v(\delta_{t,s} - a_{t,s}) \ge -va_{t,s}$, we know $r_{t,s}^2 = a_{t,s}^2 \le b_{t,s}^2$. Therefore, $r_{t,s}^2$ can be controlled by $b_{t,s}^2 = |(d_{t,\delta})_s|^2$.

In the second case, we suppose that $s \in \mathcal{J}_t$, i.e., $b_{t,s} > \frac{\delta_{t,s}}{\eta}$. There are two subclasses: 1) If $a_{t,s} \ge 0$, we have:

$$
r_{t,s}^2 \le \left(\frac{va_{t,s} + b_{t,s}}{v+1}\right)^2 = \left(\frac{v}{v+1}\right)^2 \delta_{t,s}^2 \le \eta^2\left(\frac{\delta_{t,s}}{\eta}\right)^2 \le |(d_{t,\delta})_s|^2.
$$

2) If $a_{t,s} < 0 \le b_{t,s}$, then $r_{t,s}^2 = a_{t,s}^2 \le b_{t,s}^2/v^2 = c_{t,s}^2$, and we have $s \in \mathcal{J}_t$. We can follow Step B.2 to obtain:

$$
\sum_{s\in\mathcal{J}_t} c_{t,s}^2 \le \left(\sum_{s\in\mathcal{J}_t} c_{t,s}\right)^2 \le \left(\frac{2\sqrt{S}\|d_t\| + 1}{v}\right)^2.
$$

Here, we note that for each $s \in [S]$, only one of the cases holds. Therefore, we combine these results to get:

$$
\sum_{s=1}^{S} r_{t,s}^2 \le \|d_{t,\delta}\|^2 + \frac{4S\|d_t\|^2 + 2}{v^2},
$$

which implies:

$$
\frac{1}{T}\sum_{t=0}^{T-1}\sum_{s=1}^{S} r_{t,s}^2 \le \frac{1}{T}\sum_{t=0}^{T-1}\|d_{t,\delta}\|^2 + \frac{2}{v^2} + \frac{4S}{v^2}\cdot\frac{1}{T}\sum_{t=0}^{T-1}\|d_t\|^2.
$$

We can select $v$ to be sufficiently large such that the coefficient of the last term $\frac{1}{T}\sum_{t=0}^{T-1}\|d_t\|^2$ is smaller than $\frac{1}{2}$.

**Step C: Combination and Parameter Selection.**

Finally, we can combine the all the results we obtained from Steps A and B to get the following convergence performance guarantee:

$$
\begin{aligned}
&\|\mathcal{K}(\rho_t, z_t, \omega_t, \nu_t, \lambda)\|^2 \\
=&(1 - \sum_{s=1}^{S}\omega_{t,s})^2 + \|\sum_{s=1}^{S}\omega_{t,s}\lambda_s\nabla f_s(z_t) + \sum_{i=1}^{q}\nu_{t,i}\nabla h_i(z_t)\|^2 \\
&+ \|h(z_t)\|^2 + \sum_{s=1}^{S}[\min\{\omega_{t,s}, \rho_t - \lambda_s f_s(z_t)\}]^2 \\
=&\|d_{t,\rho}\|^2 + \|d_{t,z}\|^2 + \|h(z_t)\|^2 + \sum_{s=1}^{S}r_{t,s}^2,
\end{aligned}
$$

which implies:

$$
\begin{aligned}
&\frac{1}{T}\sum_{t=0}^{T-1}\|\mathcal{K}(\rho_t, z_t, \omega_t, \nu_t, \lambda)\|^2 \\
=&\frac{1}{T}\sum_{t=0}^{T-1}\left(\|d_{t,\rho}\|^2 + \|d_{t,z}\|^2 + \|h(z_t)\|^2 + \sum_{s=1}^{S} r_{t,s}^2\right) \\
\leq&\frac{1}{T}\sum_{t=0}^{T-1}\left(\|d_{t,\rho}\|^2 + \|d_{t,z}\|^2 + \|d_{t,\delta}\|^2\right) \\
&+ \frac{2(3\sqrt{S}M+1)^2}{u^2\sigma^2}\cdot\frac{1}{T}\sum_{t=0}^{T-1}\|d_t\|^2 + \frac{2M^2}{u^2\sigma^2} + \frac{2}{v^2} + \frac{4S}{v^2}\cdot\frac{1}{T}\sum_{t=0}^{T-1}\|d_t\|^2 \\
\leq&\frac{2}{T}\sum_{t=0}^{T-1}\|d_t\|^2 + \frac{2M^2}{u^2\sigma^2} + \frac{2}{v^2} \\
\leq&\frac{2(P(\theta_0) - P(\theta_T))}{\eta T(2 - L_P\eta)} + \frac{2M^2}{u^2\sigma^2} + \frac{2}{v^2} \\
\leq&\frac{2\left(\rho_0 - \rho_T + \frac{u}{2}(\|h(z_0)\|^2 - \|h(z_T)\|^2) + \frac{v}{2}\sum_{s=1}^{S}(c_{0,s}^2 - c_{T,s}^2)\right)}{\eta T(2 - L_P\eta)} + \frac{2M^2}{u^2\sigma^2} + \frac{2}{v^2} \\
\leq&\frac{2\rho_0 + 4 + u\|h(z_0)\|^2 + v\sum_{s=1}^{S}(\lambda_s f_s(z_0) + \delta_{0,s} - \rho_0)^2}{\eta T(2 - L_P\eta)} + \frac{2M^2}{u^2\sigma^2} + \frac{2}{v^2},
\end{aligned}
\tag{7}
$$

where the last inequality is derived from the fact that $\rho_t$ has a trivial lower bound, which can be argued as follows. On the one hand, by letting $\eta \leq \frac{1}{L_P}$, we can get $P(\theta_{t+1}) \leq P(\theta_t), \forall t = 0, \ldots, T-1$. On the other hand, since $f_s(z) > 0, \forall s \in [S]$, we have $\nabla_\rho P(\theta_t) \leq 0$ when:

$$
\rho_t \leq -\frac{1}{vS} \leq \frac{1}{S}\left(\sum_{s=1}^{S}(\lambda_s f_s(z_t) + \delta_{t,s}) - \frac{1}{v}\right),
$$

which implies $\rho_{t+1} \geq \rho_t, \forall t = 0, \ldots, T-1$ if $\rho_t \leq -\frac{1}{vS}$. Therefore, we can further obtain:

$$
\begin{aligned}
&\rho_t \geq -\frac{1}{vS} - \eta\nabla_\rho P(\theta_t), \\
\Longrightarrow&\rho_t \geq -\frac{1}{vS} - \eta + \eta v\left(\sum_{s=1}^{S}\lambda_s f_s(z_t) + \delta_{t,s} - \rho_t\right), \\
\Longrightarrow&\rho_t \geq -\frac{1}{vS} - \eta - \eta v S\rho_t, \\
\Longrightarrow&\rho_t \geq -2,
\end{aligned}
$$

To finish the analysis, we select parameters such that Equation (7) converges. Let $u = \Theta(T^\xi)$, $v = \Theta(T^\xi)$, and $\eta = \Theta(T^{-\gamma})$. It is also worth noting that $L_P$ has the same order with $u$ and $v$. Thus, we maximize an order $o$, such that:

$$
\begin{aligned}
1 - \gamma - \xi &\geq o, \\
2\xi &\geq o, \\
\gamma &\geq \xi.
\end{aligned}
$$

Thus, we select $\gamma = \xi = \frac{1}{4}$, then $o = \frac{1}{2}$, i.e., the convergence rate is $\mathcal{O}(S/T^{\frac{1}{2}})$. $\qquad\square$

**Remark 5.** For hyperparameter selection, we also provide the following practical guidance, based on the insights from our theoretical analysis. (1) In practice, the total number of iterations $T$ is usually known beforehand or can be set. Therefore, the $\Theta(\cdot)$-scaling results indicate that we only need to choose the $u$, $v$, and $\eta$ parameters following the correct scaling order in terms of $T$. (2) From the analysis, we find that the hidden constants in these $\Theta(\cdot)$-scaling results only depend on Lipschitz continuity coefficient $M$ and the minimum singular value $\sigma$, both of which are relatively easy to estimate historically from

the dataset or online through the warm-up stage in training. (3) In additional to the above quantitative characterizations for choosing $u$, $v$, and $\eta$, one can additionally pick these parameters following some practical rules of thumb. Since $u$ and $v$ are the coefficients of the penalty terms, one can pick larger $u$- and $v$-values if ensuring small constraint violations is more preferred. On the other hand, if minimizing the objective is more preferred, one can choose relatively small $u$- and $v$-values.

### D.3. Stochastic WC-Penalty Algorithm

Now, we consider the ECMO problem in its stochastic form as follows:

$$\min_{z\in\mathbb{R}^k} F(z)^\top = (f_1(z),\ldots,f_S(z))$$
$$\text{s.t. } h_i(z) = 0, i = 1,\ldots,q,$$

where $f_s(z) = \mathbb{E}_\xi[f_s(z;\xi)], \forall s \in [S]$, and $h_i(z) = \mathbb{E}_\zeta[h_i(z;\zeta)], \forall i \in [q]$. Since this problem shares exactly the same form as ECMO, differing only in the specific $f_s(z)$ and $h_i(z)$, the KKT system defined in Definition 5 remains applicable. Consequently, it is still irrational to apply the penalty method, specifically, Equation (1), to address this problem.

Hence, we adopt a similar algorithmic framework, i.e., Algorithm 3, to deal with this stochastic ECMO problem. Note that we keep the aforementioned notation $\mathcal{C} = \mathbb{R} \times \mathbb{R}^k \times \mathbb{R}_+^S$ as the feasible region, and $\theta = (\rho^\top, z^\top, \delta^\top)^\top$ for simplicity. The stochastic gradients can be computed as follows:

$$\hat{\nabla}_\rho P(\theta_t) = 1 - v\sum_{s=1}^{S}(\lambda_s f_s(z_t;\mathcal{B}_t^s) + \delta_{t,s} - \rho_t),$$
$$\hat{\nabla}_z P(\theta_t) = u\sum_{i=1}^{q}h_i(z_t;\mathcal{T}_t^i)\nabla h_i(z_t;\mathcal{T}_t^i) + v\sum_{s=1}^{S}(\lambda_s f_s(z_t;\mathcal{B}_t^s) + \delta_{t,s} - \rho_t)\lambda_s\nabla f_s(z_t;\mathcal{B}_t^s), \quad (8)$$
$$\hat{\nabla}_{\delta_s} P(\theta_t) = v(\lambda_s f_s(z_t;\mathcal{B}_t^s) + \delta_{t,s} - \rho_t),$$

where $\mathcal{B}_t^s$ and $\mathcal{T}_t^i$ denote the mini-batches of sampled data at iteration with batch-sizes $\mathcal{B}(t)$ and $\mathcal{T}(t)$, respectively, for each $t, s, i$. Before giving the theoretical results and the analysis, we need an additional assumption stated as follows:

**Assumption 3** (Bounded Variance). There exist some constants $\sigma_f$ and $\sigma_h$, such that $\mathbb{E}(f_s(z;\xi) - f_s(z))^2 \leq \sigma_f^2, \forall z \in \mathbb{R}^k, s \in [S]$, and $\mathbb{E}(h_i(z;\zeta) - h_i(z))^2 \leq \sigma_h^2, \forall z \in \mathbb{R}^k, i \in [q]$.

Now, we are ready to present the theoretical results for Algorithm 3.

**Theorem 5** (Finite-Time Convergence Rate of Algorithm 3). *Under Assumptions 1, 2 and 3, for any $\kappa \in (0,1)$, preference $\lambda \in \Delta_S^{++}$, selecting $\eta = \Theta(T^{-\frac{1}{4}})$, $u = v = \Theta(T^{\frac{1}{4}})$, and $\mathcal{B}(t) = \mathcal{T}(t) = \Theta(T^{\frac{5}{4}}), \forall t$, Algorithm 3 has the following convergence result with probability at least $1 - \kappa$:*

$$\frac{1}{T}\sum_{t=0}^{T-1}\|\mathcal{K}(\rho_t, z_t, \omega_t, \nu_t, \lambda)\|^2 = \mathcal{O}\left(\frac{S}{T^{\frac{1}{2}}}\right).$$

*Proof.* To begin with, we define $\theta_{t+1} = \theta_t - \eta d_t$, which is different from the previous definition in deterministic scenario due to the stochastic nature of gradients. We split the analysis into three main steps here.

**Step A: General Control.**

---

**Algorithm 3** Stochastic WC-Penalty Algorithm

1: **Input:** Iteration rounds $T$, initialization $\theta_0 \in \mathcal{C}$, where $\rho_0 \geq 0$, and step-size $\eta$.
2: **for** $t = 0, 1, \ldots, T-1$ **do**
3:     Draw sample batches $\mathcal{B}_t^1, \ldots, \mathcal{B}_t^S$ and $\mathcal{T}_t^1, \ldots, \mathcal{T}_t^q$.
4:     Compute stochastic gradients: $\hat{\nabla}P(\theta_t)$ by Equation (8).
5:     Update $\theta_{t+1} = \mathcal{P}_\mathcal{C}(\theta_t - \eta\hat{\nabla}P(\theta_t))$.
6: **end for**

---

**Step A.1: Applying Descent Lemma.** According to the descent lemma, we have:

$$P(\theta_{t+1}) \leq P(\theta_t) + \langle \nabla P(\theta_t), \theta_{t+1} - \theta_t \rangle + \frac{L_P}{2} \|\theta_{t+1} - \theta_t\|^2$$

$$= P(\theta_t) - \eta \langle \nabla P(\theta_t), d_t \rangle + \frac{L_P \eta^2}{2} \|d_t\|^2$$

$$= P(\theta_t) - \eta \langle \nabla P(\theta_t) - \hat{\nabla} P(\theta_t) + \hat{\nabla} P(\theta_t), d_t \rangle + \frac{L_P \eta^2}{2} \|d_t\|^2$$

$$\leq P(\theta_t) - \eta \|d_t\|^2 - \eta \langle \nabla P(\theta_t) - \hat{\nabla} P(\theta_t), d_t \rangle + \frac{L_P \eta^2}{2} \|d_t\|^2$$

$$\leq P(\theta_t) - \eta \|d_t\|^2 + \frac{L_P \eta^2}{2} \|d_t\|^2 + \frac{\eta^2}{2} \|d_t\|^2 + \frac{1}{2} \|\nabla P(\theta_t) - \hat{\nabla} P(\theta_t)\|^2,$$

where the second inequality is due to the property of projection, and the last inequality is due to Cauchy–Schwarz inequality. Then, we have:

$$\eta \left( 1 - \frac{\eta}{2} - \frac{L_P \eta}{2} \right) \mathbb{E} \|d_t\|^2 \leq P(\theta_t) - P(\theta_{t+1}) + \frac{1}{2} \mathbb{E} \|\nabla P(\theta_t) - \hat{\nabla} P(\theta_t)\|^2.$$

**Step A.2: Stochastic Gradient Control.** Then, we control:

$$\mathbb{E} \|\nabla P(\theta_t) - \hat{\nabla} P(\theta_t)\|^2$$

$$= \underbrace{\mathbb{E} \|\nabla_\rho P(\theta_t) - \hat{\nabla}_\rho P(\theta_t)\|^2}_{A_t} + \underbrace{\mathbb{E} \|\nabla_z P(\theta_t) - \hat{\nabla}_z P(\theta_t)\|^2}_{B_t} + \underbrace{\mathbb{E} \|\nabla_\delta P(\theta_t) - \hat{\nabla}_\delta P(\theta_t)\|^2}_{C_t}.$$

According to Assumption 3, we have the following results: First,

$$A_t = \mathbb{E} \left( 1 - v \sum_{s=1}^{S} (\lambda_s f_s(z_t) + \delta_{t,s} - \rho_t) - (1 - v \sum_{s=1}^{S} (\lambda_s f_s(z_t; \mathcal{B}_t^s) + \delta_{t,s} - \rho_t)) \right)^2$$

$$= \mathbb{E} \left( v \sum_{s=1}^{S} (\lambda_s f_s(z_t; \mathcal{B}_t^s) - \lambda_s f_s(z_t)) \right)^2$$

$$\leq v^2 S \sum_{s=1}^{S} \lambda_s \mathbb{E} \left( f_s(z_t; \mathcal{B}_t^s) - f_s(z_t) \right)^2$$

$$\leq \frac{v^2 S \sigma_f^2}{\mathcal{B}(t)},$$

where the first inequality is due to the linearity of expectation, and the second inequality is due to $|\mathcal{B}_t^s| = \mathcal{B}(t), \forall s \in [S]$ and $\lambda \in \Delta_S^{++}$. Second, according to the similar argument, we have:

$$C_t = \mathbb{E} \left( \sum_{s=1}^{S} (v(\lambda_s f_s(z_t) + \delta_{t,s} - \rho_t) - v(\lambda_s f_s(z_t; \mathcal{B}_t^s) + \delta_{t,s} - \rho_t))^2 \right)$$

$$= v^2 \mathbb{E} \left( \sum_{s=1}^{S} (\lambda_s f_s(z_t) - \lambda_s f_s(z_t; \mathcal{B}_t^s))^2 \right)$$

$$= v^2 \sum_{s=1}^{S} \mathbb{E} \left( \lambda_s f_s(z_t) - \lambda_s f_s(z_t; \mathcal{B}_t^s) \right)^2$$

$$\leq \frac{v^2 \sigma_f^2}{\mathcal{B}(t)}.$$

Lastly, for $B_t$, we add and subtract a term to get:

$$B_t \leq \underbrace{2\mathbb{E}\|u\sum_{i=1}^{q} h_i(z_t)\nabla h_i(z_t) - u\sum_{i=1}^{q} h_i(z_t;\mathcal{T}_t^i)\nabla h_i(z_t;\mathcal{T}_t^i)\|^2}_{B_{t,1}}$$

$$+ \underbrace{2\mathbb{E}\|v\sum_{s=1}^{S}(\lambda_s f_s(z_t) + \delta_{t,s} - \rho_t)\lambda_s\nabla f_s(z_t) - v\sum_{s=1}^{S}(\lambda_s f_s(z_t;\mathcal{B}_t^s) + \delta_{t,s} - \rho_t)\lambda_s\nabla f_s(z_t;\mathcal{B}_t^s)\|^2}_{B_{t,2}}.$$

Therefore, due to the bounded variance and the smoothness assumption, we can get:

$$\begin{aligned}
B_{t,1} \leq & 4\mathbb{E}\|u\sum_{i=1}^{q} h_i(z_t)\nabla h_i(z_t) - u\sum_{i=1}^{q} h_i(z_t)\nabla h_i(z_t;\mathcal{T}_t^i)\|^2 \\
& + 4\mathbb{E}\|u\sum_{i=1}^{q} h_i(z_t)\nabla h_i(z_t;\mathcal{T}_t^i) - u\sum_{i=1}^{q} h_i(z_t;\mathcal{T}_t^i)\nabla h_i(z_t;\mathcal{T}_t^i)\|^2 \\
\leq & 4u^2 m\sum_{i=1}^{q}\mathbb{E}\|h_i(z_t)(\nabla h_i(z_t) - \nabla h_i(z_t;\mathcal{T}_t^i))\|^2 \\
& + 4u^2 m\sum_{i=1}^{q}\mathbb{E}\|(h_i(z_t) - h_i(z_t;\mathcal{T}_t^i))\nabla h_i(z_t;\mathcal{T}_t^i)\|^2 \\
\leq & \frac{4u^2 m M^2}{\mathcal{T}(t)}\sum_{i=1}^{q} h_i(z_t)^2 + \frac{4u^2 m^2 M^2 \sigma_h^2}{\mathcal{T}(t)},
\end{aligned}$$

and:

$$\begin{aligned}
B_{t,2} \leq & 4\mathbb{E}\|v\sum_{s=1}^{S}(\lambda_s f_s(z_t) + \delta_{t,s} - \rho_t)(\lambda_s\nabla f_s(z_t) - \lambda_s\nabla f_s(z_t;\mathcal{B}_t^s))\|^2 \\
& + 4\mathbb{E}\|v\sum_{s=1}^{S}(\lambda_s f_s(z_t) - \lambda_s f_s(z_t;\mathcal{B}_t^s))\lambda_s\nabla f_s(z_t;\mathcal{B}_t^s)\|^2 \\
\leq & 4v^2 S\sum_{s=1}^{S}(\lambda_s f_s(z_t) + \delta_{t,s} - \rho_t)^2\mathbb{E}\|\lambda_s\nabla f_s(z_t) - \lambda_s\nabla f_s(z_t;\mathcal{B}_t^s)\|^2 \\
& + 4v^2 S\sum_{s=1}^{S}\mathbb{E}\|(\lambda_s f_s(z_t) - \lambda_s f_s(z_t;\mathcal{B}_t^s))\lambda_s\nabla f_s(z_t;\mathcal{B}_t^s)\|^2 \\
\leq & \frac{4v^2 S M^2}{\mathcal{B}(t)}\sum_{s=1}^{S}(\lambda_s f_s(z_t) + \delta_{t,s} - \rho_t)^2 + \frac{4v^2 S^2 M^2 \sigma_f^2}{\mathcal{B}(t)}.
\end{aligned}$$

**Step A.3: Combination.** Hence, we can combine the results obtained in the last two sub-steps to get:

$$\begin{aligned}
& \eta\left(1 - \frac{\eta}{2} - \frac{L_P\eta}{2}\right)\frac{1}{T}\sum_{t=0}^{T-1}\mathbb{E}\|d_t\|^2 \\
\leq & \frac{1}{T}(P(\theta_0) - P(\theta_T)) + \frac{1}{T}\sum_{t=0}^{T-1}\left(\frac{v^2 S\sigma_f^2}{\mathcal{B}(t)} + \frac{2u^2 m^2 M^2 \sigma_h^2}{\mathcal{T}(t)} + \frac{2v^2 S^2 M^2 \sigma_f^2}{\mathcal{B}(t)}\right. \\
& \left. + \frac{2u^2 m M^2}{\mathcal{T}(t)}\sum_{i=1}^{q} h_i(z_t)^2 + \frac{2v^2 S M^2}{\mathcal{B}(t)}\sum_{s=1}^{S}(\lambda_s f_s(z_t) + \delta_{t,s} - \rho_t)^2\right).
\end{aligned}$$

Therefore, we almost bound $\sum_{t=0}^{T-1}\mathbb{E}\|d_t\|^2$. Later, we select proper $\mathcal{B}(t), \mathcal{T}(t)$, and $u, v$ to complete this process.

**Step B: KKT System.**

In this step, we consider the KKT system defined in Definition 5, and aim to control each term of $\|\mathcal{K}\|^2$. Before diving deep into the detailed analysis, we introduce some necessary notations here. We denote $\bar{d}_t, \forall t = 0, \ldots, T-1$, as the expected version of update direction. In other words, let $\bar{\theta}_{t+1} = \mathcal{P}_{\mathcal{C}}(\theta_t - \eta \nabla P(\theta_t))$, then we have $\bar{\theta}_{t+1} = \theta_t - \eta \bar{d}_t$. Since the KKT system is related to $\bar{d}_t$, we need to find the relationship between $d_t$, which has been controlled in Step A, and $\bar{d}_t$.

**Step B.1: Stationarity Terms.** We now consider the first two terms in the KKT system, i.e., the stationarity terms for $\rho$ and $z$, respectively. We consider $z$ first. On the one hand, we have:

$$\bar{d}_{t,z} = u \sum_{i=1}^{q} h_i(z_t) \nabla h_i(z_t) + v \sum_{s=1}^{S} (\lambda_s f_s(z_t) + \delta_{t,s} - \rho_t) \lambda_s \nabla f_s(z_t),$$

implying that in Definition 5, by setting $\omega_{t,s} = v(\lambda_s f_s(z_t) + \delta_{t,s} - \rho_t)$, $\nu_{t,i} = u h_i(z_t)$, and $\omega_t = (\omega_{t,1}, \ldots, \omega_{t,S})^\top$, $\nu_t = (\nu_{t,1}, \ldots, \nu_{t,q})^\top$, we can bound the corresponding stationarity term for $z$ as long as $\|\bar{d}_{t,z}\|^2$ can be controlled. On the other hand, we follow the argument in Step A.2 to obtain the following result:

$$
\begin{aligned}
\|\bar{d}_{t,z}\|^2 &= \mathbb{E}\|\bar{d}_{t,z}\|^2 \\
&\leq 2\mathbb{E}\|d_{t,z}\|^2 + 2\mathbb{E}\|\bar{d}_{t,z} - d_{t,z}\|^2 \\
&\leq 2\mathbb{E}\|d_{t,z}\|^2 + 8\left(\frac{u^2 m^2 M^2 \sigma_h^2}{\mathcal{T}(t)} + \frac{v^2 S^2 M^2 \sigma_f^2}{\mathcal{B}(t)}\right. \\
&\quad \left. + \frac{u^2 m M^2}{\mathcal{T}(t)} \sum_{i=1}^{q} h_i(z_t)^2 + \frac{v^2 S M^2}{\mathcal{B}(t)} \sum_{s=1}^{S} (\lambda_s f_s(z_t) + \delta_{t,s} - \rho_t)^2\right),
\end{aligned}
$$

where $\mathbb{E}\|d_{t,z}\|^2$ is already characterized previously. As for the stationarity term for $\rho$, we note that:

$$1 - \sum_{s=1}^{S} \omega_{t,s} = 1 - \sum_{s=1}^{S} v(\lambda_s f_s(z_t) + \delta_{t,s} - \rho_t) = \bar{d}_{t,\rho},$$

and:

$$\|\bar{d}_{t,\rho}\|^2 = \mathbb{E}\|\bar{d}_{t,\rho}\|^2 \leq 2\mathbb{E}\|d_{t,\rho}\|^2 + 2\mathbb{E}\|\bar{d}_{t,\rho} - d_{t,\rho}\|^2 \leq 2\mathbb{E}\|d_{t,\rho}\|^2 + \frac{2v^2 S \sigma_f^2}{\mathcal{B}(t)},$$

which indicates that the stationarity terms are well controlled.

**Step B.2: Primal Feasibility Term.** Then, we consider the primal feasibility, i.e., $\|h(z)\|^2$. Similar to the deterministic ECMO problem, we first define some notations as follows: Let $c_{t,s} = \lambda_s f_s(z_t; \mathcal{B}_t^s) + \delta_{t,s} - \rho_t$, and $\bar{c}_{t,s} = \lambda_s f_s(z_t) + \delta_{t,s} - \rho_t$. Besides, we also denote the following index sets:

$$\mathcal{I}_t = \{s \in [S] : vc_{t,s} \leq \frac{\delta_{t,s}}{\eta}\}, \qquad \mathcal{J}_t = \{s \in [S] : vc_{t,s} > \frac{\delta_{t,s}}{\eta}\}.$$

Therefore, we have:

$$u\sigma \|h(z_t)\| \leq \|u \nabla h(z_t) h(z_t)\| = \|\bar{d}_{t,z} - v \sum_{s=1}^{S} \bar{c}_{t,s} \lambda_s \nabla f_s(z_t)\| \leq \|\bar{d}_{t,z}\| + vM \sum_{s=1}^{S} |\bar{c}_{t,s}|,$$

which further implies:

$$\|h(z_t)\| \leq \frac{1}{u\sigma} \|\bar{d}_{t,z}\| + \frac{vM}{u\sigma} \sum_{s=1}^{S} |\bar{c}_{t,s}|,$$

$$\implies \|h(z_t)\|^2 \leq \frac{2}{u^2 \sigma^2} \|\bar{d}_{t,z}\|^2 + \frac{2v^2 M^2}{u^2 \sigma^2} \left(\sum_{s=1}^{S} |\bar{c}_{t,s}|\right)^2.$$

Hence, we need to control:

$$
\left( \sum_{s=1}^{S} |\bar{c}_{t,s}| \right)^2 \leq 2 \underbrace{\left( \sum_{s \in \mathcal{I}_t} |\bar{c}_{t,s}| \right)^2}_{C(\mathcal{I}_t)} + 2 \underbrace{\left( \sum_{s \in \mathcal{J}_t} |\bar{c}_{t,s}| \right)^2}_{C(\mathcal{J}_t)}.
$$

To this end, we can obtain the following two results:

$$
\begin{aligned}
C(\mathcal{I}_t) &= 2\mathbb{E} \left( \sum_{s \in \mathcal{I}_t} |\bar{c}_{t,s}| \right)^2 \\
&\leq 2\mathbb{E} \left[ \sum_{s \in \mathcal{I}_t} (|\bar{c}_{t,s} - c_{t,s}| + |c_{t,s}|) \right]^2 \\
&\leq 4\mathbb{E} \left[ \sum_{s \in \mathcal{I}_t} |\bar{c}_{t,s} - c_{t,s}| \right]^2 + 4\mathbb{E} \left[ \sum_{s \in \mathcal{I}_t} |c_{t,s}| \right]^2 \\
&\leq 4\mathbb{E} \left[ \sum_{s \in \mathcal{I}_t} |\lambda_s(f_s(z_t) - f_s(z_t; \mathcal{B}_t^s))| \right]^2 + 4\mathbb{E} \left[ \sqrt{|\mathcal{I}_t|} \frac{\|d_{t,\delta}\|}{v} \right]^2 \\
&\leq \frac{4|\mathcal{I}_t|\sigma_f^2}{\mathcal{B}(t)} + \frac{4|\mathcal{I}_t|}{v^2} \mathbb{E}\|d_{t,\delta}\|^2,
\end{aligned}
$$

and further:

$$
\begin{aligned}
C(\mathcal{J}_t) &= 2 \left( \sum_{s \in \mathcal{J}_t} \bar{c}_{t,s} \right)^2 \\
&\leq 4 \left( \sum_{s \in [S]} \bar{c}_{t,s} \right)^2 + 4 \left( \sum_{s \in \mathcal{I}_t} |\bar{c}_{t,s}| \right)^2 \\
&\leq \frac{8|\mathcal{I}_t|\sigma_f^2}{\mathcal{B}(t)} + \frac{8|\mathcal{I}_t|}{v^2} \mathbb{E}\|d_{t,\delta}\|^2 + 4 \left( \frac{1 - \bar{d}_{t,\rho}}{v} \right)^2 \\
&\leq \frac{8|\mathcal{I}_t|\sigma_f^2}{\mathcal{B}(t)} + \frac{8|\mathcal{I}_t|}{v^2} \mathbb{E}\|d_{t,\delta}\|^2 + \frac{8}{v^2} + \frac{16}{v^2} \left( \mathbb{E}\|d_{t,\rho}\|^2 + \frac{v^2 S \sigma_f^2}{\mathcal{B}(t)} \right).
\end{aligned}
$$

Therefore, we can obtain:

$$
\left( \sum_{s=1}^{S} |\bar{c}_{t,s}| \right)^2 \leq \frac{12|\mathcal{I}_t|\sigma_f^2}{\mathcal{B}(t)} + \frac{12|\mathcal{I}_t|}{v^2} \mathbb{E}\|d_{t,\delta}\|^2 + \frac{8}{v^2} + \frac{16}{v^2} \mathbb{E}\|d_{t,\rho}\|^2 + \frac{16 S \sigma_f^2}{\mathcal{B}(t)}.
$$

Finally, substituting it back, we can get:

$$\|h(z_t)\|^2 \leq \frac{2}{u^2\sigma^2}\|\bar{d}_{t,z}\|^2 + \frac{2v^2M^2}{u^2\sigma^2}\left(\sum_{s=1}^{S}|\bar{c}_{t,s}|\right)^2$$

$$\leq \frac{4}{u^2\sigma^2}\mathbb{E}\|d_{t,z}\|^2 + \frac{16}{u^2\sigma^2}\left(\frac{u^2m^2M^2\sigma_h^2}{\mathcal{T}(t)} + \frac{v^2S^2M^2\sigma_f^2}{\mathcal{B}(t)}\right.$$

$$+ \frac{u^2mM^2}{\mathcal{T}(t)}\sum_{i=1}^{q}h_i(z_t)^2 + \frac{v^2SM^2}{\mathcal{B}(t)}\sum_{s=1}^{S}(\lambda_sf_s(z_t)+\delta_{t,s}-\rho_t)^2\right)$$

$$+ \frac{2v^2M^2}{u^2\sigma^2}\left(\frac{12|\mathcal{I}_t|\sigma_f^2}{\mathcal{B}(t)} + \frac{12|\mathcal{I}_t|}{v^2}\mathbb{E}\|d_{t,\delta}\|^2 + \frac{8}{v^2} + \frac{16}{v^2}\mathbb{E}\|d_{t,\rho}\|^2 + \frac{16S\sigma_f^2}{\mathcal{B}(t)}\right)$$

$$\leq \frac{4}{u^2\sigma^2}\mathbb{E}\|d_{t,z}\|^2 + \frac{32M^2}{u^2\sigma^2}\mathbb{E}\|d_{t,\rho}\|^2 + \frac{24SM^2}{u^2\sigma^2}\mathbb{E}\|d_{t,\delta}\|^2 + \frac{16M^2}{u^2\sigma^2} + \frac{56Sv^2M^2\sigma_f^2}{u^2\sigma^2\mathcal{B}(t)}$$

$$+ \frac{16}{u^2\sigma^2}\left(\frac{u^2m^2M^2\sigma_h^2}{\mathcal{T}(t)} + \frac{v^2S^2M^2\sigma_f^2}{\mathcal{B}(t)}\right.$$

$$+ \frac{u^2mM^2}{\mathcal{T}(t)}\sum_{i=1}^{q}h_i(z_t)^2 + \frac{v^2SM^2}{\mathcal{B}(t)}\sum_{s=1}^{S}(\lambda_sf_s(z_t)+\delta_{t,s}-\rho_t)^2\right).$$

Thus, we can select $\mathcal{T}(t)$ to be sufficiently large such that:

$$\|h(z_t)\|^2 \leq \frac{8}{u^2\sigma^2}\mathbb{E}\|d_{t,z}\|^2 + \frac{64M^2}{u^2\sigma^2}\mathbb{E}\|d_{t,\rho}\|^2 + \frac{48SM^2}{u^2\sigma^2}\mathbb{E}\|d_{t,\delta}\|^2 + \frac{32M^2}{u^2\sigma^2} + \frac{112Sv^2M^2\sigma_f^2}{u^2\sigma^2\mathcal{B}(t)}$$

$$+ \frac{32}{u^2\sigma^2}\left(\frac{u^2m^2M^2\sigma_h^2}{\mathcal{T}(t)} + \frac{v^2S^2M^2\sigma_f^2}{\mathcal{B}(t)} + \frac{v^2SM^2}{\mathcal{B}(t)}\sum_{s=1}^{S}(\lambda_sf_s(z_t)+\delta_{t,s}-\rho_t)^2\right).$$

Then, we can select $u$ to be sufficiently large such that the sum of the first three terms in RHS is no larger than $\mathbb{E}\|d_t\|^2$. Besides, we let $\mathcal{B}(t)$ and $\mathcal{T}(t)$ be some $T$-dependent constant (but independent with $t$). Then, we have:

$$\frac{1}{T}\sum_{t=0}^{T-1}\|h(z_t)\|^2 \leq \frac{1}{T}\sum_{t=0}^{T-1}\mathbb{E}\|d_t\|^2 + \frac{32M^2}{u^2\sigma^2} + \frac{112Sv^2M^2\sigma_f^2}{u^2\sigma^2\mathcal{B}(t)} + \frac{32}{u^2\sigma^2T}\sum_{t=0}^{T-1}\left(\frac{u^2m^2M^2\sigma_h^2}{\mathcal{T}(t)}\right.$$

$$+ \frac{v^2S^2M^2\sigma_f^2}{\mathcal{B}(t)} + \frac{v^2SM^2}{\mathcal{B}(t)}\sum_{s=1}^{S}(\lambda_sf_s(z_t)+\delta_{t,s}-\rho_t)^2\right).$$

**Step B.3: Dual Feasibility and Complementary Slackness Term.** Now, we consider the last term in the KKT system: $\bar{r}_{t,s} = \min\{\omega_{t,s}, \rho_t - \lambda_sf_s(z_t)\}$.

To begin with, we first introduce some notations here: let $\bar{a}_{t,s} = \rho_t - \lambda_sf_s(z_t)$, $\bar{b}_{t,s} = \omega_{t,s} = v(\lambda_sf_s(z_t)+\delta_{t,s}-\rho_t)$, and $a_{t,s} = \rho_t - \lambda_sf_s(z_t;\mathcal{B}_t^s)$, $b_{t,s} = v(\lambda_sf_s(z_t;\mathcal{B}_t^s)+\delta_{t,s}-\rho_t)$. We also note the following fact:

$$(d_{t,\delta})_s = \frac{1}{\eta}(\delta_{t,s}-\delta_{t+1,s})$$

$$= \frac{1}{\eta}(\delta_{t,s}-\max\{\delta_{t,s}-\eta\nabla_\delta P(\theta_t),0\})$$

$$= \min\{b_{t,s}, \frac{\delta_{t,s}}{\eta}\},$$

and discuss $\bar{r}_{t,s}^2$ in two different cases.

In the first case, we suppose $s \in \mathcal{I}_t$, i.e., $b_{t,s} \leq \frac{\delta_{t,s}}{\eta}$. No matter what the orders among $b_{t,s}$, $\bar{b}_{t,s}$, and $\frac{\delta_{t,s}}{\eta}$ are, we can obtain:

$$\bar{r}_{t,s}^2 \leq \bar{b}_{t,s}^2 \leq 2\mathbb{E}(b_{t,s}^2) + 2\mathbb{E}(\bar{b}_{t,s}-b_{t,s})^2 \leq 2\mathbb{E}(d_{t,\delta})_s^2 + \frac{2v^2\lambda_s^2\sigma_f^2}{\mathcal{B}(t)}.$$

In the second case, we suppose $s \in \mathcal{J}_t$, i.e., $\frac{\delta_{t,s}}{\eta} < b_{t,s}$. If $\bar{b}_{t,s} > \frac{\delta_{t,s}}{\eta}$, following the same argument in the deterministic scenario, if $\bar{a}_{t,s} \geq 0$, we have: $\bar{r}_{t,s}^2 \leq \mathbb{E}(d_{t,\delta})_s^2$. If $\bar{a}_{t,s} < 0$, we have:

$$\bar{r}_{t,s}^2 = \bar{a}_{t,s}^2 \leq \frac{\bar{b}_{t,s}^2}{v^2} = \bar{c}_{t,s}^2.$$

Thus, we have:

$$\sum_{s \in \mathcal{J}_t, \bar{a}_{t,s} < 0} \bar{c}_{t,s}^2 \leq \sum_{s \in \mathcal{J}_t} \bar{c}_{t,s}^2$$

$$\leq \left( \sum_{s \in \mathcal{J}_t} \bar{c}_{t,s} \right)^2$$

$$\leq \frac{4|\mathcal{I}_t|\sigma_f^2}{\mathcal{B}(t)} + \frac{4|\mathcal{I}_t|}{v^2}\mathbb{E}\|d_{t,\delta}\|^2 + \frac{4}{v^2} + \frac{8}{v^2}\mathbb{E}\|d_{t,\rho}\|^2 + \frac{8S\sigma_f^2}{\mathcal{B}(t)}.$$

Otherwise, $\bar{b}_{t,s} \leq \frac{\delta_{t,s}}{\eta}$, we have: $\bar{r}_{t,s}^2 \leq \mathbb{E}(d_{t,\delta})_s^2$ if $\bar{b}_{t,s} \geq -\frac{\delta_{t,s}}{\eta}$, and $\bar{r}_{t,s}^2 = \bar{b}_{t,s}^2 \leq (\bar{b}_{t,s} - b_{t,s})^2 \leq \frac{v^2 \lambda_s^2 \sigma_f^2}{\kappa \mathcal{B}(t)}$ with probability at least $1 - \kappa$ if $\bar{b}_{t,s} < -\frac{\delta_{t,s}}{\eta}$ for any $\kappa \in (0,1)$ according to Chebyshev inequality.

Combining aforementioned results together, with probability at least $1 - \kappa$, we can get:

$$\sum_{s=1}^{S} \bar{r}_{t,s}^2 \leq 2\sum_{s=1}^{S} \mathbb{E}(d_{t,\delta})_s^2 + \frac{2v^2\sigma_f^2}{\mathcal{B}(t)} + \frac{v^2\sigma_f^2}{\kappa\mathcal{B}(t)}$$

$$+ \frac{4|\mathcal{I}_t|\sigma_f^2}{\mathcal{B}(t)} + \frac{4|\mathcal{I}_t|}{v^2}\mathbb{E}\|d_{t,\delta}\|^2 + \frac{4}{v^2} + \frac{8}{v^2}\mathbb{E}\|d_{t,\rho}\|^2 + \frac{8S\sigma_f^2}{\mathcal{B}(t)}$$

$$\leq 2\mathbb{E}\|d_{t,\delta}\|^2 + \frac{4S+8}{v^2}\mathbb{E}\|d_t\|^2 + \frac{4}{v^2} + \frac{12S\sigma_f^2}{\mathcal{B}(t)} + \frac{2v^2\sigma_f^2}{\mathcal{B}(t)} + \frac{v^2\sigma_f^2}{\kappa\mathcal{B}(t)},$$

which further implies:

$$\frac{1}{T}\sum_{t=0}^{T-1}\sum_{s=1}^{S} r_{t,s}^2 \leq \frac{2}{T}\sum_{t=0}^{T-1}\mathbb{E}\|d_{t,\delta}\|^2 + \frac{4S+8}{v^2}\cdot\frac{1}{T}\sum_{t=0}^{T-1}\mathbb{E}\|d_t\|^2 + \frac{4}{v^2} + \frac{12S\sigma_f^2}{\mathcal{B}(t)} + \frac{2v^2\sigma_f^2}{\mathcal{B}(t)} + \frac{v^2\sigma_f^2}{\kappa\mathcal{B}(t)},$$

holds with probability at least $1 - \kappa$ according to Chebyshev inequality. We can select $v$ to be sufficiently large such that the coefficient of the second terms in RHS is no larger than 1. Then, with probability at least $1 - \kappa$, we obtain:

$$\frac{1}{T}\sum_{t=0}^{T-1}\sum_{s=1}^{S} r_{t,s}^2 \leq \frac{2}{T}\sum_{t=0}^{T-1}\mathbb{E}\|d_{t,\delta}\|^2 + \frac{1}{T}\sum_{t=0}^{T-1}\mathbb{E}\|d_t\|^2 + \frac{4}{v^2} + \frac{12S\sigma_f^2}{\mathcal{B}(t)} + \frac{2v^2\sigma_f^2}{\mathcal{B}(t)} + \frac{v^2\sigma_f^2}{\kappa\mathcal{B}(t)}.$$

**Step C: Combination and Parameter Selection.**

Finally, we can combine the all the results we obtained from Steps A and B to get the following convergence performance guarantee:

$$\|\mathcal{K}(\rho_t, z_t, \omega_t, \nu_t, \lambda)\|^2$$

$$= (1 - \sum_{s=1}^{S}\omega_{t,s})^2 + \|\sum_{s=1}^{S}\omega_{t,s}\lambda_s\nabla f_s(z_t) + \sum_{i=1}^{q}\nu_{t,i}\nabla h_i(z_t)\|^2$$

$$+ \|h(z_t)\|^2 + \sum_{s=1}^{S}[\min\{\omega_{t,s}, \rho_t - \lambda_s f_s(z_t)\}]^2$$

$$= \|\bar{d}_{t,\rho}\|^2 + \|\bar{d}_{t,z}\|^2 + \|h(z_t)\|^2 + \sum_{s=1}^{S}\bar{r}_{t,s}^2,$$

which further implies:

$$\|\mathcal{K}(\rho_t, z_t, \omega_t, \nu_t, \lambda)\|^2$$

$$=\|\bar{d}_{t,\rho}\|^2 + \|\bar{d}_{t,z}\|^2 + \|h(z_t)\|^2 + \sum_{s=1}^{S} \bar{r}_{t,s}^2$$

$$\leq 2\mathbb{E}\|d_{t,\rho}\|^2 + 2\mathbb{E}\|d_{t,z}\|^2$$

$$+ \frac{2v^2 S \sigma_f^2}{\mathcal{B}(t)} + 8\left( \frac{u^2 m^2 M^2 \sigma_h^2}{\mathcal{T}(t)} + \frac{v^2 S^2 M^2 \sigma_f^2}{\mathcal{B}(t)} + \frac{v^2 S M^2}{\mathcal{B}(t)} \sum_{s=1}^{S} (\lambda_s f_s(z_t) + \delta_{t,s} - \rho_t)^2 \right)$$

$$+ 2\|h(z_t)\|^2 + \sum_{s=1}^{S} \bar{r}_{t,s}^2,$$

where we select large enough $\mathcal{T}(t) \geq 8u^2 m M^2$ in the last inequality. Hence, we can further obtain the following results with probability at least $1 - \kappa$:

$$\frac{1}{T} \sum_{t=0}^{T-1} \|\mathcal{K}(\rho_t, z_t, \omega_t, \nu_t, \lambda)\|^2$$

$$\leq \frac{6}{T} \sum_{t=0}^{T-1} \mathbb{E}\|d_t\|^2 + \frac{2v^2 S \sigma_f^2}{\mathcal{B}(t)} + \frac{8u^2 m^2 M^2 \sigma_h^2}{\mathcal{T}(t)} + \frac{8v^2 S^2 M^2 \sigma_f^2}{\mathcal{B}(t)}$$

$$+ \frac{64M^2}{u^2 \sigma^2} + \frac{224 S v^2 M^2 \sigma_f^2}{u^2 \sigma^2 \mathcal{B}(t)} + \frac{64}{u^2 \sigma^2} \left( \frac{u^2 m^2 M^2 \sigma_h^2}{\mathcal{T}(t)} + \frac{v^2 S^2 M^2 \sigma_f^2}{\mathcal{B}(t)} \right)$$

$$+ \frac{4}{v^2} + \frac{12 S \sigma_f^2}{\mathcal{B}(t)} + \frac{2v^2 \sigma_f^2}{\mathcal{B}(t)} + \frac{v^2 \sigma_f^2}{\kappa \mathcal{B}(t)}$$

$$+ \left( \frac{64 v^2 S M^2}{u^2 \sigma^2 \mathcal{B}(t)} + \frac{8v^2 S M^2}{\mathcal{B}(t)} \right) \cdot \left( \frac{12 S \sigma_f^2}{\mathcal{B}(t)} + \frac{8}{v^2} + \frac{16 S \sigma_f^2}{\mathcal{B}(t)} \right),$$

where the last line uses the result of $\left( \sum_{s=1}^{S} |\bar{c}_{t,s}| \right)^2$. Therefore, we can use the $\mathcal{O}(\cdot)$ notations to further organize this result as:

$$\frac{1}{T} \sum_{t=0}^{T-1} \|\mathcal{K}(\rho_t, z_t, \omega_t, \nu_t, \lambda)\|^2$$

$$= \mathcal{O}\left( \frac{S(u+v)}{\eta T} \right) + \mathcal{O}\left( \frac{Su^2}{\eta \mathcal{T}(t)} \right) + \mathcal{O}\left( \frac{Sv^2}{\eta \mathcal{B}(t)} \right) + \mathcal{O}\left( \frac{1}{u^2} \right) + \mathcal{O}\left( \frac{1}{v^2} \right), \tag{9}$$

with probability at least $1 - \kappa$. Thus, we can select parameters to ensure the convergence of Equation (9). Specifically, let $u = \Theta(T^\gamma)$, $v = \Theta(T^\gamma)$, $\eta = \Theta(T^{-\gamma})$, and $\mathcal{B}(t) = \mathcal{T}(t) = \Theta(T^\mu)$. Suppose the convergence order is $o$, then we maximize $o$, such that:

$$1 - 2\gamma \geq o,$$
$$\mu - 3\gamma \geq o,$$
$$2\gamma \geq o,$$

Hence, we can set $\gamma = \frac{1}{4}$ and $\mu = \frac{5}{4}$, to obtain $o = \frac{1}{2}$. In other words, The convergence rate is $\mathcal{O}(S/T^{\frac{1}{2}})$. $\qquad \square$

**Remark 6.** By comparing Algorithm 1 and Algorithm 3, along with their respective analyses, we identify that the key challenge in the stochastic scenario arises from the **stochastic gradients**. Specifically, due to the gap between the full gradient and its stochastic estimator, the analysis for Algorithm 3 becomes more complex, even though the overall structure of the analysis remains the same. Consequently, to deal with the stochasticity, we 1) add and subtract several intermediate terms, applying the triangle inequality to bridge the gap between the stochastic gradients and their full-gradient counterparts; 2) use the Chebyshev Inequality to *accurately* bound the dual feasibility and complementary slackness terms in the KKT system; and 3) carefully select the batch-sizes $\mathcal{B}$ and $\mathcal{T}$ to ensure finite-time convergence.

# E. Setups and Additional Results of Numerical Experiments

In this section, we begin by presenting the details of our experimental setups for two data weighting tasks introduced in Section 5, alongside additional numerical results, including evaluations on larger-scale LLMs. Following that, we provide the setup and numerical results of a multi-task meta-learning task to further validate our proposed algorithm.

## E.1. Data Weighting for Multi-Objective RLHF Reward Model Training

### 1) Detailed Setup.

**Overview.** The reward model scores LLM-generated responses to prompts based on human-aligned criteria in Reinforcement Learning from Human Feedback (RLHF). The multi-objective data weighting task aims to determine optimal weights over training datasets for training a reward model that maximize multiple validation metrics in Pareto sense. As shown in the literature, this data weighting task is often considered using a bilevel framework (Pan et al., 2024; Shen et al., 2024a). Moreover, potentially conflicting human preferences, such as *helpfulness*, *verbosity*, naturally motivates a multi-task formulation. Hence, we model this problem as an MTBL problem.

**Training.** Specifically, there are $N$ training sets $\mathcal{T}_1, \ldots, \mathcal{T}_N$. Each training set $\mathcal{T}_n, n \in [N]$ contains $|\mathcal{T}_n|$ prompt-response pairs $\{p_{n,i}, r_{n,i}\}, i = 1, \ldots, |\mathcal{T}_n|$, and the corresponding scores $s_{n,i}$. The derivation, quality, and tendency of these training sets may be unknown in practice, indicating that our data weighting task aims to assign larger weights to datasets that are of higher quality and better aligned with the target preference. To this end, we consider a weight vector $x = (x_1, \ldots, x_N)^\top$, where each element corresponds to a training set. These weights are normalized using a SoftMax function. We denote the parameter of the reward model as $y$, then it is a function of the weight $x$.

**Validation.** These trained weights are evaluated in the validation process, where $M$ validation sets $\mathcal{V}_1, \ldots, \mathcal{V}_M$ are considered. Each $\mathcal{V}_m, m \in [M]$ contains $|\mathcal{V}_m|$ prompt-response pairs $\{p_{m,j}, r_{m,j}\}, j = 1, \ldots, |\mathcal{V}_m|$, and the corresponding scores $s_{m,j}$, where the scores are labeled based on some specific and unique criteria such as *helpfulness*, *correctness*, and *verbosity*. In real-world scenarios, these metrics may not be aligned with training sets, and can be inaccessible. In other words, the $M$ validation sets verify the capability of the reward model in $M$ different directions.

**Formulation and Setup.** To sum up, the formulation of this task is stated as follows:

$$\min_{x,y} \left( \sum_{j=1}^{|\mathcal{V}_1|} \mathcal{L}(\tilde{s}_{1,j}(y(x)), s_{1,j}), \ldots, \sum_{j=1}^{|\mathcal{V}_M|} \mathcal{L}(\tilde{s}_{M,j}(y(x)), s_{M,j}) \right)$$

$$\text{s.t. } y(x) \in \arg\min_y \sum_{n=1}^N \frac{\exp(x_n)}{\sum_{n'} \exp(x_{n'})} \sum_{i=1}^{|\mathcal{T}_n|} \mathcal{L}(\tilde{s}_{n,i}(y), s_{n,i}),$$

where $\mathcal{L}$, set to mean squared error (MSE) here, denotes the loss evaluated by the true score label $s$ and the predicted score label $\tilde{s}$ generated by the reward model. We use the HelpSteer dataset (Wang et al., 2023) as the basic dataset. For training datasets, we select two sets with criteria *coherence* and *verbosity*, and also construct a set with *random generated scores*, indicating $N = 3$. It is worth noting that the prompt-response pairs in these 3 training sets are identical. For the validation sets, we consider all 5 validation sets, i.e., $M = 5$, each corresponding to a different evaluation criterion: *helpfulness*, *correctness*, *coherence*, *complexity*, and *verbosity*, respectively.

We utilize a multi-layer perceptron (MLP) with a depth of 500 and a width of 5 to represent the reward model. The input is encoded using (He et al., 2021) and has a dimension of 500. For the parameters, we set the batch size to 256, the learning rate to $\eta = 10^{-8}$, and the total number of iterations to $T = 3,000$. Moreover, we change the preference vector $\lambda \in \Delta_5^{++}$ to explore the Pareto front. We evaluate three MTBL algorithms, MOML (Ye et al., 2021), MoCo (Fernando et al., 2023), FORUM (Ye et al., 2024), as our baselines. Specifically, the inner loop of each algorithm is set to 50, with learning rates for the UL and LL variables ($x$ and $y$) set to $\alpha = 10^{-3}$ and $\beta = 10^{-8}$, respectively. Additionally, for MoCo, we set the extra parameters $\gamma = 10^{-3}$ and $\rho = 10^{-6}$; for FORUM, we set $\rho = 2$. Each experiment is repeated for 5 times. All numerical experiments for this reward model training task were conducted on a cluster of 4 NVIDIA H100 GPUs (94GB each) using PyTorch's DistributedDataParallel.

The expected results are as follows: 1) WC-Penalty Algorithm achieves a low validation loss for each metric, demonstrating the convergence behavior of our algorithm. 2) When different preferences are chosen during the validation process, our algorithm covers a much larger portion of Pareto front compared to other baselines. Moreover, when weights are assigned to

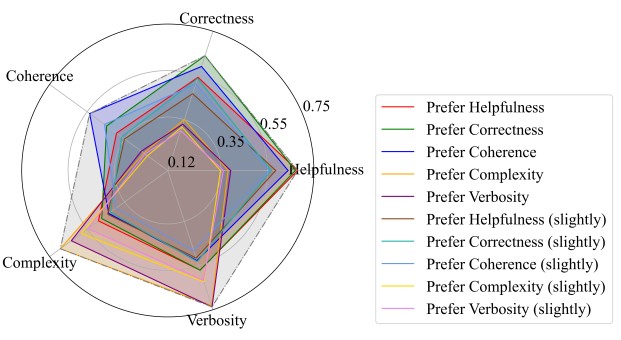

*(a)* Pareto exploration with more preferences.

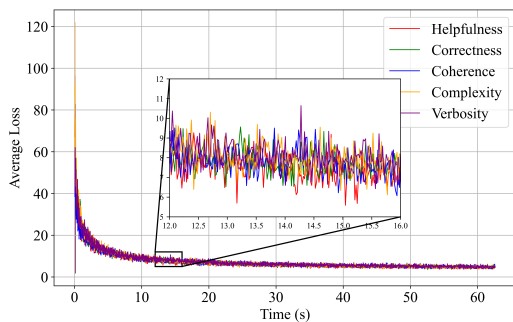

*(b)* Convergence for different objectives.

*Figure 11.* Additional results for Pareto exploration.

prioritize specific objectives, our algorithm yields a lower validation for those objectives compared to the case of using alternative preference vectors.

**2) Additional Numerical Results.**

We now provide more numerical results on this data weighting for reward model training task, accompanied by discussions to emphasize the advantages of Algorithm 1 in this subsection.

1. Pareto Exploration.

In addition to the results demonstrated in Section 5, we select 5 more additional preference vectors by setting $\lambda$ as $\lambda_s = 0.84$ for some $s \in [S]$ and $\lambda_{s'} = 0.04, \forall s' \neq s$, referring to this as "slightly prefer" some objective in Figure 11a. This further verifies the Pareto exploration capability of Algorithm 1. Furthermore, to provide a clearer intuition of how our algorithm converges for each objective, Figure 11b illustrates the convergence behavior of each objective based on their loss and standard error over the 5 trials when the preference vector is set to $\lambda = [0.01, 0.01, 0.01, 0.01, 0.96]^\top$ (i.e., under preference "prefer verbosity"). We also compute the area ratios $\frac{S_{\text{ours}}}{S_{\text{baseline}}}$ for each baseline shown in Figure 6b, yielding the following results: $\frac{S_{\text{ours}}}{S_{\text{MOML}}} = 1.67$, $\frac{S_{\text{ours}}}{S_{\text{FORUM}}} = 1.36$, and $\frac{S_{\text{ours}}}{S_{\text{MoCo}}} = 1.56$. Our approach demonstrates at least a $36\%$ improvement on this metric, quantifying the Pareto exploration capability of our algorithm.

Figure 6a and Figure 11 align with our expectations. Intuitively, the loss for each objective converges over time, as our algorithm takes all objectives into account and achieves a convergence rate of $\mathcal{O}(S/T^{\frac{1}{2}})$. What's more, we also find that the performances on *complexity* and *verbosity* are similar, but significantly different from those of the other three metrics. This outcome, while not entirely surprising, is interesting, as it aligns with our expectations as well. These two criteria focus on redundancy and response length, whereas the other metrics are more concerned with the content of the responses. Our algorithm captures this subtle distinction by selecting some specific preferences, while other baselines fail to consider this point. This strength becomes ***particularly valuable*** in practice when more objectives are introduced. Our approach enables a systematic exploration upfront, allowing the handling of these objectives, regardless of the complexity of their internal relationships.

2. Convergence Performance.

Except for the ability on Pareto exploration, we also highlight the good convergence behavior in Figure 12. Specifically, we compare the running time of our algorithm with that of all baselines over $T = 3,000$ steps in Figure 12a. We average the loss over 5 trials for each algorithm and include the standard error bars to ensure statistical significance. This result clearly illustrates that our algorithm converges to some weakly Pareto stationary solution in no more than than 70 seconds, while MoCo, MOML, and FORUM require over $5 \times 10^2$, $2 \times 10^3$, and $2 \times 10^4$ seconds, respectively, to complete this process. Similarly, Figure 12b shows how our algorithm and three baselines behave in $T = 3,000$ iterations, with the iteration axis shown on a logarithmic scale. It is evident that the slope of our method is the smallest (or the largest in absolute value sense).

The computational efficiency shown in Figure 12a can be attributed to the following two key factors. **First**, while other baselines follow a double-loop scheme to alternately update variables $x$ and $y$, investing significant effort in the inner loop to optimize the LL function $g(x, y)$, our approach uses a simple projected gradient descent, employing a single-loop paradigm to handle the variables as a unified entirety. **Second**, since the ECMO problem inherently treats $x$ and $y$ as a unified entity,

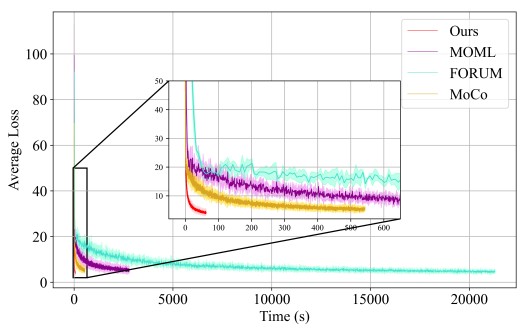
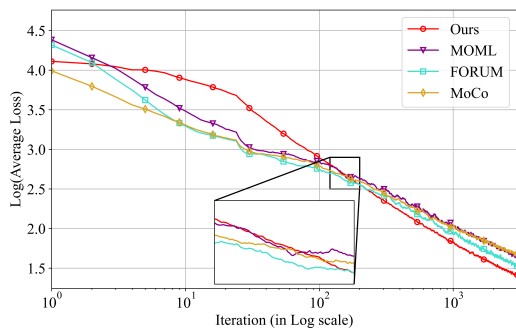

*(a)* Convergence performance vs running time.

*(b)* Convergence performance vs iterations.

*Figure 12.* Additional results for Convergence Performance.

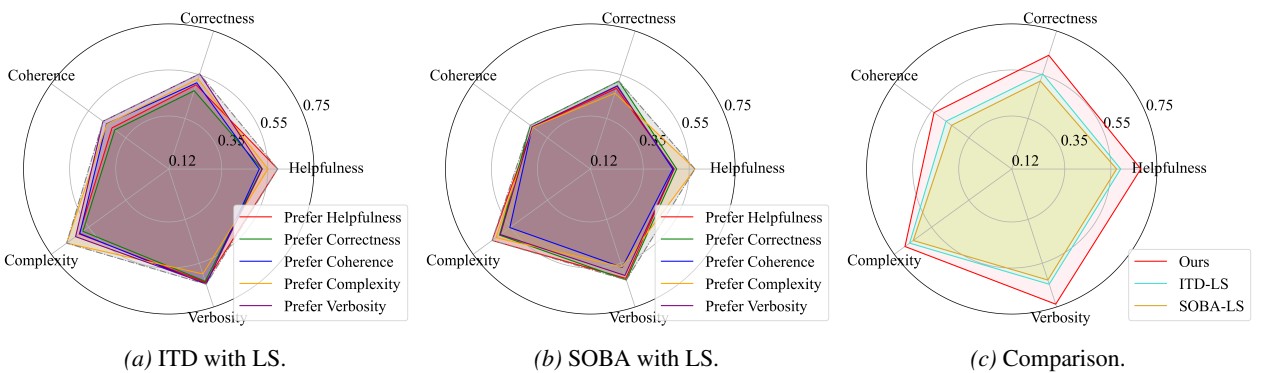

*(a)* ITD with LS.

*(b)* SOBA with LS.

*(c)* Comparison.

*Figure 13.* Additional results on bilevel algorithms.

we omit the use of implicit gradient methods (Ghadimi & Wang, 2018; Ji et al., 2021) to compute the Hessian inverse, significantly reducing computational costs.

The best slope of our approach in Figure 12b further validates its convergence performance. Specifically, as illustrated in Theorem 3, our WC-Penalty algorithm achieves a convergence rate of $\mathcal{O}(S/T^{\frac{1}{2}})$ for general ECMO problems, which also applies to this MTBL setup. This rate matches the one obtained for MoCo in the context of MTBL problems under their strongest assumption. In contrast, 1) MOML lacks finite-time convergence guarantees, and 2) FORUM provides a rate of $\mathcal{O}(S/T^{\frac{1}{4}})$ (where the parameter $S$ is omitted in the $\mathcal{O}(\cdot)$ notation in their work). In the end, we'd also like to point out that these convergence rates are based on different metrics. All of the baselines' setups require the strongly convex LL function $g(x, y)$ to ensure well-defined metrics. By contrast, our metric, $\|\mathcal{K}(\rho, z, \omega, \nu, \lambda)\|^2$, is more general, as it applies to general ECMO problems.

## 3. More Discussions.

Finally, we provide some additional discussion for this experiment, focusing on three main aspects as follows. **Dataset**: The dataset we use (HelpSteer, (Wang et al., 2023)) is almost the "optimal" to validate our algorithm, as it contains 5 objectives, whereas most other existing datasets have no more than 3. This allows a more realistic simulation of how algorithms perform with multiple potentially conflicting objectives. Furthermore, it is well-known and widely adopted within the community, reflecting its high quality. **Model**: We consider an MLP as our reward model. It not only performs well during the learning process (according to the loss values) but also requires relatively short running time. Therefore, we argue that this MLP model is well-suited for our simulation. **Baselines**: Finally, the baselines we select are state-of-the-art methods in MTBL, while other approaches lack theoretical convergence guarantees. Having said that, we also compare our algorithm with some bilevel algorithms for completeness. Specifically, we consider ITD (Ji et al., 2021) and SOBA (Dagréou et al., 2022) with linear scalarization as our baselines (note that it's nontrivial to extend their approaches with WC method). Again, we set $\lambda$ as $\lambda_s = 0.96$ for some $s \in [S]$ and $\lambda_{s'} = 0.01, \forall s' \neq s$ to evaluate their capabilities in exploring the Pareto front. Figure 13 compares our WC-Penalty Algorithm with bilevel algorithms. In particular, we highlight the following two points: 1) The LS method fails to guarantee a full exploration of the Pareto front in this highly nonconvex scenario, while our algorithm

excels at covering a larger portion of the Pareto front, further validating our theoretical analysis. 2) The explorations of the two bilevel algorithms are "irregular" and do not reveal the relationships between different objectives. In contrast, our algorithm provides rational guidance in exploring the Pareto front, as demonstrated in Figures 6a and 11a.

### E.2. Data Weighting in Multi-Objective LLM Alignment

**1) Detailed Setup.**

**Overview.** In the Large Language Model (LLM) Alignment task, our goal is to align a pretrained LLM with human preferences. Instead of relying on a reward model to guide the LLM, we directly utilize the prompt-response data to finetune the language model. In this section, we introduce our data weighting task for multi-objective LLM alignment. Similarly, given that 1) multiplex human preferences necessitate the multi-task formulation, and 2) the data weighting task is commonly framed as a bilevel problem, this problem can naturally be expressed as an MTBL problem.

**Training.** In the training process, there are $N$ training sets $\mathcal{T}_1, \ldots, \mathcal{T}_N$, where each $\mathcal{T}_n, n \in [N]$ contains $|\mathcal{T}_n|$ prompt-response pairs $(p_{n,i}, r_{p,i}), i = 1, \ldots, |\mathcal{T}_n|$. Each $\mathcal{T}_n, n \in [N]$ represents the conversation pairs aligned with one human metric, but may perform poorly in other directions. However, the focus of each dataset is typically unknown in practice. Hence, our goal is to assign an appropriate weight for each dataset, ensuring that the LLM performs well across all metrics. To this end, we consider a weight vector $x = (x_1, \ldots, x_N)^\top$, where each element corresponds to a training set. These weights are normalized using a SoftMax function. We denote the parameter of the base LLM as $y$, then it is a function of the weight $x$.

**Validation.** The trained weight $x$ is evaluated during the validation process, where $M$ validation sets $\mathcal{V}_1, \ldots, \mathcal{V}_M$ are considered. Each $\mathcal{V}_m, m \in [M]$ contains $|\mathcal{V}_m|$ prompt-response pairs $\{p_{m,j}, r_{m,j}\}, j = 1, \ldots, |\mathcal{V}_m|$. Similarly, each validation set represents high-quality conversation pairs based on a specific and unique criterion, such as *helpfulness*, *correctness*, or *verbosity*. In real-world scenarios, these metrics may not align with those used in the training sets ($M$ may be not equal to $N$) and can often be inaccessible. The overall goal is to finetune the pre-trained LLM, i.e., $y$, such that the validation loss for all metrics is minimized in Pareto sense.

**Formulation and Setup.** Based on the previous introduction, we can formally model the problem as follows:

$$
\min_{x,y} \left( \sum_{j=1}^{|\mathcal{V}_1|} \mathcal{L}(\tilde{r}_{1,j}(p_{1,j}; y(x)), r_{1,j}), \ldots, \sum_{j=1}^{|\mathcal{V}_M|} \mathcal{L}(\tilde{r}_{M,j}(p_{M,j}; y(x)), r_{M,j}) \right)
$$

$$
\text{s.t. } y(x) \in \arg\min_y \sum_{n=1}^{N} \frac{\exp(x_n)}{\sum_{n'} \exp(x_{n'})} \sum_{i=1}^{|\mathcal{T}_n|} \mathcal{L}(\tilde{r}_{n,i}(p_{n,i}; y), r_{n,i}),
$$

where $\mathcal{L}$, set to cross-entropy in this task, measures the difference between the generated response $\tilde{r}$ and the given response $r$. To incorporate more objectives, we continue to use HelpSteer (Wang et al., 2023) as our base dataset. However, HelpSteer does not provide separate datasets for each individual criterion. Hence, we construct the training and validation datasets as follows. For training sets, we calculate the average score $\bar{s}$ across the five metrics for each prompt-response pair, and consider it to construct $\mathcal{T}_1$ for $\bar{s} \geq 2.5$ and $\mathcal{T}_2$ for $\bar{s} \leq 2$, respectively. In other words, we set $N = 2$ to represent data with different quality levels. For validation sets, we assign a prompt-response pair to a criterion-specific dataset if its corresponding score for that criterion is at least 3 (with scores ranging from $\{0, 1, 2, 3, 4\}$). Besides, we consider all 5 validation sets, i.e., $M = 5$, each corresponding to a different evaluation criterion: *helpfulness*, *correctness*, *coherence*, *complexity*, and *verbosity*, respectively.

We use Llama-3.2-1B-Instruct (Meta, 2024) as our pretrained LLM, and apply the LoRA technique with a rank of 8. For the parameters, we set batch size to 32, learning rate to $\eta = 10^{-5}$ and run the algorithm for $T = 3,000$ iterations. We also set different preference vectors $\lambda \in \Delta_5^{++}$ to explore the Pareto front. For the baselines with a double-loop structure, the inner-loop iteration is set to 40. Each experiment is repeated for 5 times. All numerical experiments were conducted on a cluster of 4 NVIDIA H100 GPUs (94GB each) using PyTorch's DistributedDataParallel.

**2) Additional Numerical Results.**

Similarly, we provide more numerical results on this data weighting in LLM alignment task along with discussions in this subsection.

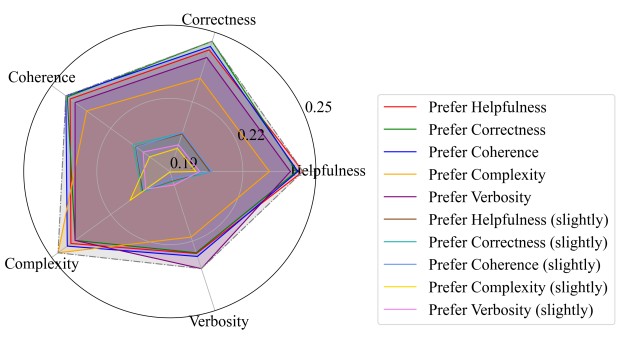

*(a)* Exploration with more preferences.

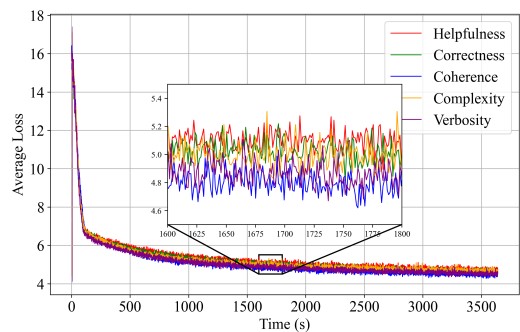

*(b)* Different objectives in Alg. 1.

*Figure 14.* Additional results for Pareto exploration.

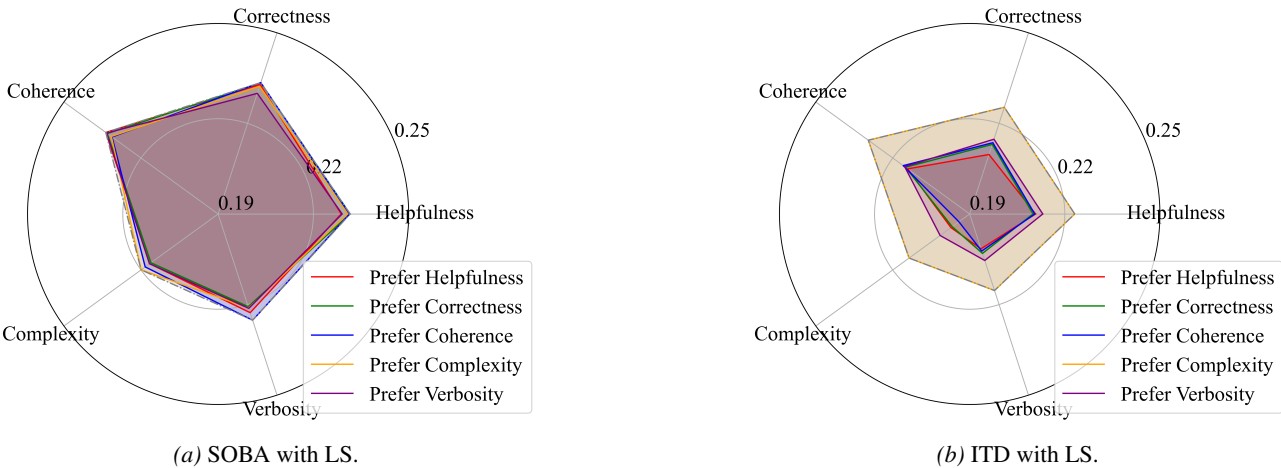

*(a)* SOBA with LS.

*(b)* ITD with LS.

*Figure 15.* Additional results on bilevel algorithms.

## 1. Pareto Exploration.

Figure 14 presents additional numerical results on Pareto exploration. In Figure 14a, "slightly prefer" refers to selecting $\lambda_s = 0.84$ for some $s$ and $\lambda_{s'} = 0.04$ for $s' \neq s$. While these preferences do not yield improved performance, they still exhibit regular Pareto exploration behavior, as the loss on the focused objective remains relatively small.

Figure 14b illustrates the convergence performances of different objectives in Algorithm 1 when selecting $\lambda = [0.01, 0.01, 0.01, 0.01, 0.96]^\top$ (i.e., under preference "prefer verbosity"). The loss is averaged over 5 trials, and the standard error bars are also included. Notably, the preferred objective, *verbosity*, achieves relatively better performance, which aligns with the results shown in Figure 14a.

We also provide how bilevel algorithms (Dagréou et al., 2022; Ji et al., 2021) explore the Pareto front using linear scalarization technique in Figure 15. The basic setup remains the same: we set $\lambda_s = 0.96$ for some $s$ and $\lambda_{s'} = 0.01$ for all other $s'$. Notably, while both algorithms still exhibit some exploration behaviors with different preference vectors, this exploration is highly irregular. In other words, when certain objectives are preferred, the relative performance may not be dominant. This irregularity stems from the highly nonconvex nature of the LLM alignment problem, where the neural networks, with billions of parameters and highly nonlinear operations, can take unpredictable forms, rendering the linear scalarization method ineffective.

## 2. MTBL Baselines and Discussions.

We also consider the aforementioned MTBL algorithms (Ye et al., 2021; Fernando et al., 2023; Ye et al., 2024) as our baselines in Figure 16. Specifically, our algorithm still outperforms in Pareto exploration when compared with MOML and MoCo algorithms, since a larger portion of Pareto front is covered by our approach, as demonstrated in Figure 16a. The reason we do not include the FORUM algorithm here lies in its impractical memory cost in large scale problems. As

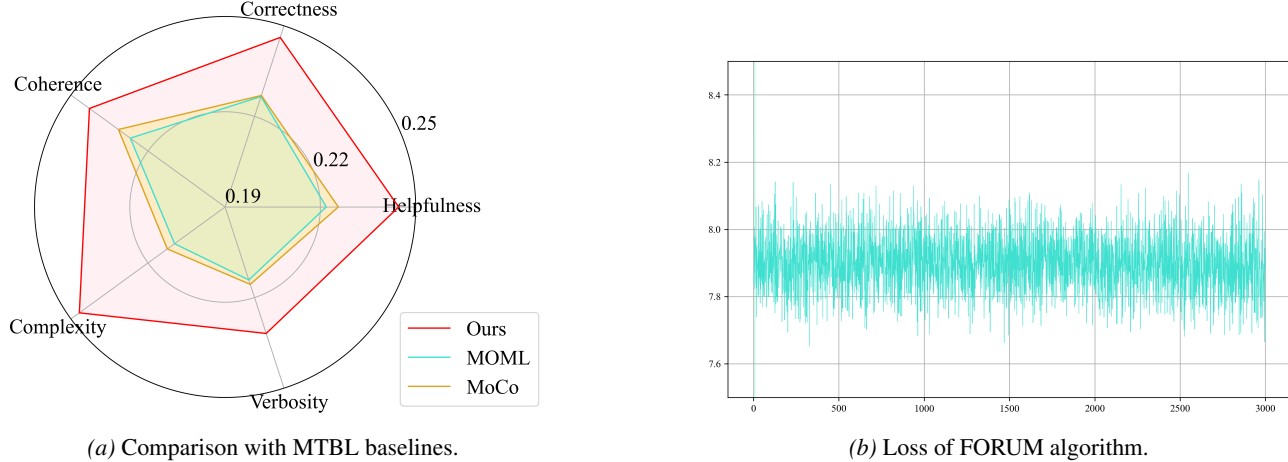

*(a)* Comparison with MTBL baselines.
*(b)* Loss of FORUM algorithm.

*Figure 16.* Additional results on MTBL algorithms.

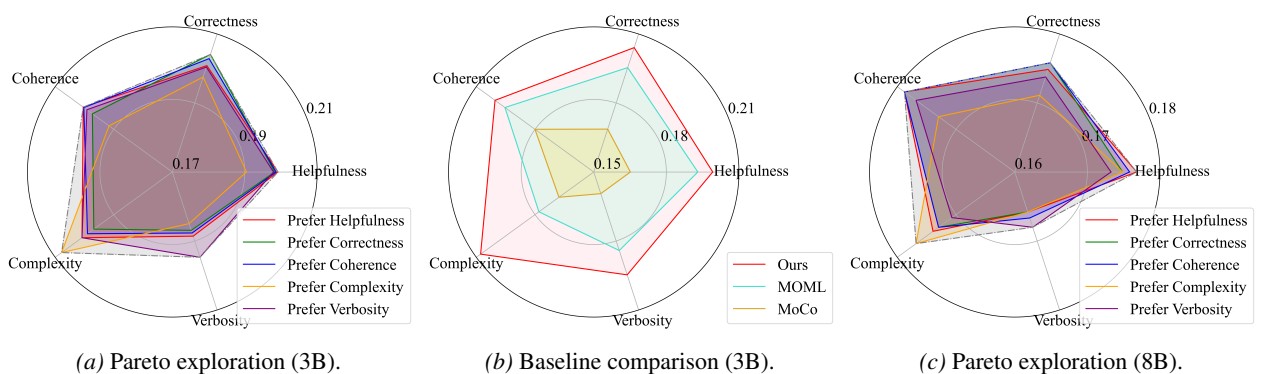

*(a)* Pareto exploration (3B).
*(b)* Baseline comparison (3B).
*(c)* Pareto exploration (8B).

*Figure 17.* Data weighting task in larger-scale (3B & 8B) LLM alignment.

mentioned in the setup, we set the inner-loop iterations (if applicable) as 40 for every algorithm. Nevertheless, this leads to "CUDA out of memory" error when implementing the FORUM algorithm, since 1) its workflows are overly complicated, and 2) its maintained values are extremely memory-consuming. In fact, in our GPUs with 94GB of memory each, the maximum number of inner-loop rounds for FORUM without causing an overflow is 2, which results in the performance shown in Figure 16b. Obviously, the validation loss does not decrease over time, thus, we exclude it from Figure 16a.

Finally, we also claim the rationale behind this experiment. **Dataset:** We still use HelpSteer as the basic dataset because it contains 5 potentially conflicting objectives, allowing us to intuitively demonstrate the performances on Pareto exploration. **Model:** The LLM model employed here is Llama-3.2-1B-Instruct, which has proven to generate reasonable responses and is relatively efficient to train. **Baselines:** For completeness, we consider both MTBL algorithms and bilevel algorithms with linear scalarization as baselines.

## 3. Larger-Scale Numerical Experiments and Results.

In order to further validate the capability of our Algorithm 1 in large-scale problems, we enlarge the pretrained LLM model from **Llama-3.2-1B-Instruct** to **Llama-3.2-3B-Instruct** and **Llama-3.1-8B-Instruct** in this subsection.

In Figure 17, we set the preference vector $\lambda$ as $\lambda_s = 0.96$ for some $s \in [S]$ and $\lambda_{s'} = 0.01$, $\forall s' \neq s$, using $1/\text{loss}$ as our metric for each objective. Specifically, as shown in Figure 17a, by varying the preference vectors, Algorithm 1 can efficiently explore a diverse set of Pareto stationary solutions, enabling our algorithm to recover a large portion of the Pareto front. Also, Figure 17b further demonstrates that our proposed algorithm outperforms existing methods in recovering the Pareto front, highlighting its effectiveness in Pareto front exploration. Furthermore, in the 8B model, our algorithm consistently demonstrates its ability to perform systematic Pareto exploration, as shown in Figure 17c. All of these numerical results further confirm the excellent scalability of our developed algorithm.

*Table 4.* Hypervolume results in larger-scale (3B) LLM alignment.

| Alg. | **Ours** | Helpfulness | Correctness | Coherence | Complexity | Verbosity | MOML | MoCo |
|------|----------|-------------|-------------|-----------|------------|-----------|------|------|
| HV ($\uparrow$) | **4.87e0** | 3.53e0 | 3.04e0 | 3.44e0 | 2.43e0 | 3.67e0 | 8.48e-1 | 8.49e-4 |

Moreover, we also compare our Algorithm 1 with MTBL baselines (Ye et al., 2021; Fernando et al., 2023) with two important metrics, hypervolume and $\epsilon$-metric. Table 4 demonstrates that our algorithm dramatically outperforms the baselines even before completing full Pareto exploration (labeled as Helpfulness, etc.) in terms of hypervolume, and the Pareto exploration still leads to better performances. Moreover, Figure 18 further confirms that, in terms of $\epsilon$-metric: 1) our method consistently outperforms the baselines, and 2) with varying preference vectors, our method converges to the desired solutions. This is consistent with our theoretical analysis and the previous numerical results.

*Figure 18.* $\epsilon$-metric.

### E.3. Multi-Task Meta-Learning Task

**1) Experimental Setup.**

**Overview.** We consider a multi-task meta-learning problem (Ye et al., 2021; Ji et al., 2021; Qin et al., 2025), where the goal is to train a single model capable of addressing multiple tasks within the MTBL framework. This task is particularly useful for handling heterogeneous datasets using a relatively small-scale model. Specifically, the training process corresponds to our lower-level problem, where the model is expected to develop a universal representation capability. Conversely, the validation process corresponds to the upper-level problem, which aims to balance the trade-offs among multiple potentially conflicting tasks. The overall objective is to enable the model to achieve superior performance in the Pareto sense.

**Detailed formulation.** We construct five heterogeneous MNIST tasks by assigning each task a distinct digit pair, resulting in different class distributions across tasks. In particular, for each task $s \in \{1, \ldots, 5\}$, we create a mixed training subset $\mathcal{T}_s$ containing $80\%$ samples from its specific digit pair $(0, 1), (2, 3), (4, 5), (6, 7)$, or $(8, 9)$ and $20\%$ from the remaining digits. The validation subsets $\mathcal{V}_s$ are constructed analogously from the MNIST test split using the same five digit-pair tasks.

Our model consists of a shared multi-layer perceptron (MLP) parameterized by $x$ and a final linear classifier parameterized by $y$. Each image is flattened and passed through two fully connected layers of width $512$ with ReLU activation functions, followed by a linear layer producing a $256$-dimensional representation. The the classifier maps this representation to $10$ logits.

The problem is formulated as follows:

$$\min_{x,y} F(x,y) = \left( \sum_{d \in \mathcal{V}_1} \mathcal{L}(\mathcal{NN}(d; x, y), l(d)), \ldots, \sum_{d \in \mathcal{V}_5} \mathcal{L}(\mathcal{NN}(d; x, y), l(d)) \right)^\top$$

$$\text{s.t. } y(x) \in \arg\min_y g(x,y) = \sum_{s=1}^{5} \sum_{d \in \mathcal{T}_s} \mathcal{L}(\mathcal{NN}(d; x, y), l(d)),$$

where $\mathcal{L}$ denotes the cross-entropy loss, $d$ denotes the digit in the dataset, $\mathcal{NN}(\cdot)$ denotes the output of our MLP model, and $l(d)$ denotes the label of $d$. Note that the cross-entropy loss is a convex function, and the parameter $y$ represents a linear layer. Therefore, the lower-level function $g(x, y)$ satisfies the LLGC condition with respective to $y$.

**2) Numerical Results.**

Figure 19 demonstrates the effectiveness of our Algorithm 1 in Pareto exploration and its superior performance compared to baselines. Specifically, in Figure 19a, in addition to the preference vectors used in the previous subsections, we also include the "Equally Prefer" preference, where $\lambda = [0.2, 0.2, 0.2, 0.2, 0.2]^\top$. The numerical results once again confirm the Pareto

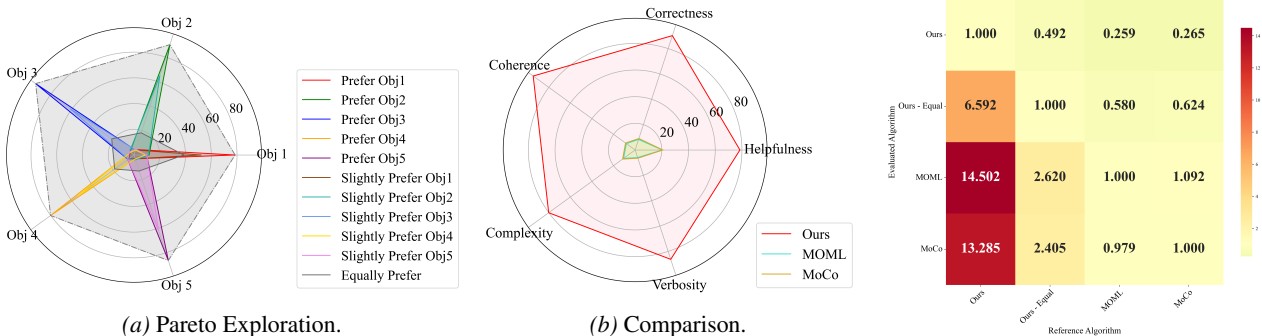

*(a)* Pareto Exploration.      *(b)* Comparison.

*Figure 19.* Results on multi-task meta-learning.      *Figure 20.* $\epsilon$-metric in meta-learning.

exploration capability of out approach, allowing it to effectively balance the trade-offs among multiple meta-learning tasks. Additionally, Figure 19b shows that our algorithm outperforms the baselines in this environment.

*Table 5.* Hypervolume results in multi-task meta-learning.

| Alg. | **Ours** | Ours - Equal | MOML | MoCo |
|---|---|---|---|---|
| HV ($\uparrow$) | **1.14e-3** | 5.10e-4 | 1.19e-4 | 1.51e-4 |

In addition, we compare our Algorithm 1 with MTBL baselines using two important metrics, hypervolume and $\epsilon$-metric as well. Table 5 demonstrates that our method outperforms the baselines even before completing full Pareto exploration (which is the result under the "Equally Prefer" preference vector selection, and is labeled as "Ours - Equal"), and as the preferences vary, the hypervolume (labeled as **Ours**) is significantly larger than that of the baselines. Moreover, Figure 20 further confirms that, in terms of $\epsilon$-metric: 1) our method consistently outperforms the baselines, and 2) with varying preference vectors, our method converges to the desired solutions.

## F. More Discussions on KKT System

**Part A:** To demonstrate that our KKT system defined in Definition 5 is rational, we first prove that: the KKT condition introduced in Appendix C.4 holds if and only if $\mathcal{K}(\rho, z, \omega, \nu, \lambda) = 0$.

*Proof.* We prove that both directions are correct.

$(\implies)$

We assume that KKT condition holds. Then, the stationary condition with respect to $\rho$ and $z$ exactly implies that the first two terms in $\mathcal{K}(\rho, z, \omega, \nu, \lambda)$ are $0 \in \mathbb{R}$ and $\mathbf{0} \in \mathbb{R}^{\mathbf{k}}$. Besides, $h(z) = 0$ directly leads to the third term in $\mathcal{K}(\rho, z, \omega, \nu, \lambda)$ is $\mathbf{0} \in \mathbb{R}^{\mathbf{q}}$. Finally, for any $s \in [S]$, according to $\lambda_s f_s(z) - \rho \leq 0$, $\omega_s \geq 0$, and $\omega_s(\lambda_s f_s(z) - \rho) = 0$, we have: $\min\{\omega_s, \rho - \lambda_s f_s(z)\} = 0$. Combining these arguments, we know $\mathcal{K}(\rho, z, \omega, \nu, \lambda) = 0$.

$(\impliedby)$

We assume $\mathcal{K}(\rho, z, \omega, \nu, \lambda) = 0$, then it's obvious that the stationary condition and the primal feasible condition for equality constraints are satisfied. We mainly focus on the last term in the KKT system.

For any $s \in [S]$, we have $\min\{\omega_s, \rho - \lambda_s f_s(z)\} = 0$. If $\omega_s \leq \rho - \lambda_s f_s(z)$, then we can get: $\omega_s = 0$ and $\rho \geq \lambda_s f_s(z)$. If $\omega_s > \rho - \lambda_s f_s(z)$, then we can get: $\rho - \lambda_s f_s(z) = 0$ and $\omega_s > 0$. Both scenario guarantee that (1) primal feasible condition for inequality constraints, (2) dual feasible condition, and (3) complementary slackness condition are satisfied. $\square$

**Part B:** Recall that, $\tilde{z}$ is an $\epsilon$-Pareto stationary solution of ECMO if and only if there exist some $\rho \in \mathbb{R}, \omega \in \mathbb{R}^S, \nu \in \mathbb{R}^q, \lambda \in \Delta_S^{++}$ such that $\|\mathcal{K}(\rho, z, \omega, \nu, \lambda)\|_2^2 \leq \epsilon$. According to this definition, when $\tilde{z}$ is $\epsilon$-Pareto stationary, we have:

$$|\min\{\omega_s, \rho - \lambda_s f_s(z)\}| \leq \|\mathcal{K}(\rho, z, \omega, \nu, \lambda)\|_2^2 \leq \epsilon,$$

for any $s \in [S]$.

From Part A, the primal difficulty of the understanding stems from the term $\min\{\omega_s, \rho - \lambda_s f_s(z)\}$, which is distinct from the counterparts of original KKT conditions, while the correspondences of other parts are trivial. This raises a natural but nontrivial question: *Can $|\min\{\omega_s, \rho - \lambda_s f_s(z)\}| \leq \epsilon$ really imply the complementary slackness condition $\omega_s(\lambda_s f_s(z) - \rho) \approx 0$?*

Even if the "accurate" scenario is already proved in Part A, the answer to this question still remains unclear. The primary issue is that: even though the minimum one is sufficiently close to 0, if the other one goes to infinity, then the multiplication of them cannot be directly concluded. To affirmatively answer this question, we prove the following proposition: In our Theorem 3, if $|\min\{\omega_{t,s}, \rho_t - \lambda_s f_s(z_t)\}| \leq \epsilon$ holds for any $s \in [S]$, then both $\omega_{t,s}$ and $\lambda_s f_s(z_t) - \rho_t$ are bounded for any $s \in [S]$.

*Proof.* The following arguments hold for any $s \in [S]$. If $\omega_{t,s} \leq \rho_t - \lambda_s f_s(z_t)$, then $|\omega_{t,s}| \leq \epsilon$. According to the selection of dual variables, we have:

$$\omega_{t,s} = v(\lambda_s f_s(z_t) + \delta_{t,s} - \rho_t)$$
$$\implies |\rho_t - \lambda_s f_s(z_t)| \leq |\lambda_s f_s(z_t) + \delta_{t,s} - \rho_t| + |\delta_{t,s}| \leq \frac{\epsilon}{v} + |\delta_{t,s}|.$$

If $\omega_{t,s} > \rho_t - \lambda_s f_s(z_t)$, then $|\rho_t - \lambda_s f_s(z_t)| \leq \epsilon$. According to the selection of dual variables, we have:

$$\omega_{t,s} = v(\lambda_s f_s(z_t) + \delta_{t,s} - \rho_t)$$
$$\implies |\omega_{t,s}| \leq v|\lambda_s f_s(z_t) + \delta_{t,s} - \rho_t| + v|\delta_{t,s}| \leq v\epsilon + v|\delta_{t,s}|.$$

Therefore, in order to show the desired boundedness, we only need to show that $\delta_{t,s}$ is bounded. Since $\delta_{0,s}$ is fixed, we prove the boundedness of $\delta_{t,s}$ by induction. First, we suppose $\delta_{t-1,s}$ is bounded.

According to our WC-Penalty Algorithm, we know that $\delta_{t,s} \geq 0$ for any $t = 0, \ldots, T - 1$. Besides, for any $t > 0$, we have:

$$\delta_{t,s} = \mathcal{P}_{\mathbb{R}_+}\left(\delta_{t-1,s} - \eta v(\lambda_s f_s(z_{t-1}) + \delta_{t-1,s} - \rho_{t-1})\right).$$

If $\lambda_s f_s(z_{t-1}) + \delta_{t-1,s} - \rho_{t-1} \geq 0$, then $0 \leq \delta_{t,s} \leq \delta_{t-1,s}$. Thus, the boundedness of $\delta_{t,s}$ can be derived from the boundedness of $\delta_{t-1,s}$.

If $\lambda_s f_s(z_{t-1}) + \delta_{t-1,s} - \rho_{t-1} < 0$, then we have:

$$0 \leq \delta_{t,s} \leq \delta_{t-1,s} + \eta v\left(\rho_{t-1} - \lambda_s f_s(z_{t-1}) - \delta_{t-1,s}\right) \leq \delta_{t-1,s} + \eta v\rho_0,$$

where the last inequality is due to the non-increasing property of the sequence $\{\rho_t\}_{t=0}^{T-1}$ demonstrated in **Step C** of the analysis of Theorem 3. This ends our proof. $\square$

Therefore, by the boundedness, we can argue that $|\min\{\omega_s, \rho - \lambda_s f_s(z)\}| \leq \epsilon$ indeed implies the complementary slackness condition $\omega_s(\lambda_s f_s(z) - \rho) \approx 0$.

**Part C:** From these analyses, we observe that (1) the KKT system defined in Definition 5 is not strictly equivalent to the conventional KKT condition, but (2) in our context, $\|\mathcal{K}(\rho, z, \omega, \nu, \lambda)\|_2^2 \leq \epsilon$ still ensures that all of the four kinds of original KKT conditions are only $\epsilon$-violated. The reason why we control this surrogate system primally lies in the difficulty of handling the inequality terms in the original KKT conditions, which are challenging to be quantified. Our newly defined KKT system not only forms the basis for the subsequent algorithmic design and theoretical analysis, but also constitutes a novel contribution in its own right.

## G. Synthetic Examples

To illustratively validate our theoretical results, we further provide three synthetic examples demonstrating that: (i) our Assumption 2 does *not* imply stronger conditions; (ii) our Algorithm 1 performs well in ECMO problems; and (iii) our Algorithm 1 performs well in LLGC-MTBL problems.

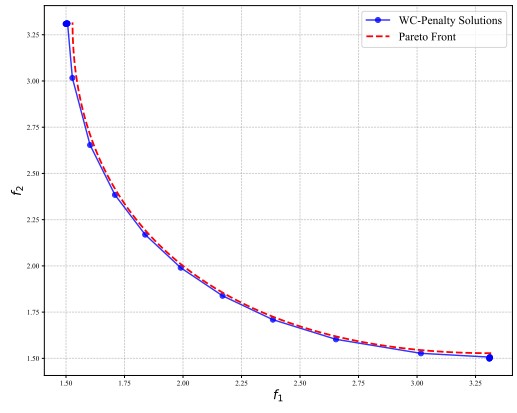

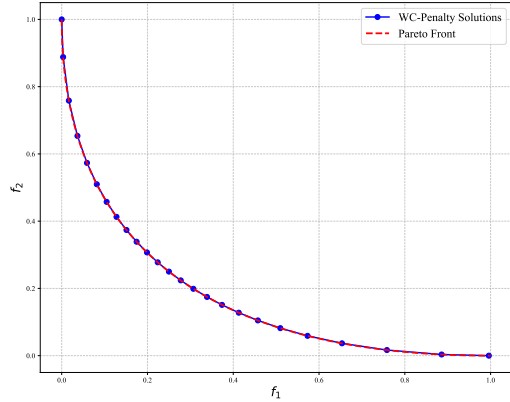

*(a)* Synthetic example for ECMO.         *(b)* Synthetic example for MTBL.

*Figure 21.* Additional synthetic examples.

**(i)** Consider an LLGC-MTBL problem with $x, y \in \mathbb{R}$:

$$\min_{x,y} \left(y, y+1\right)^{\top}, \quad \text{s.t. } y \in \mathcal{M}(x) := \arg\min_y g(x,y) = xy + \frac{1}{4}y^4.$$

Clearly, we have: $h(x,y) = \nabla_y g(x,y) = x + y^3$ and $\nabla_{yy}^2 g(x,y) = 3y^2 \geq 0$. This implies that $g(x,\cdot)$ is *convex but not strongly convex* for any $x$. To validate Assumption 2, we note that the associated gradient matrix is:

$$\begin{pmatrix} 1 & 0 \\ 3y^2 & 1 \end{pmatrix},$$

which always has a determinant of 1, meaning it is full column rank, thus perfectly satisfying Assumption 2 but not requiring strong convexity.

**(ii)** Following the synthetic example in (Gebken et al., 2017), we consider an ECMO problem as follows:

$$\min_{z \in \mathbb{R}^2} F(z) = \left((z_1 - 2)^2 + (z_2 - 1)^2, (z_1 - 2)^2 + (z_2 + 1)^2\right)^{\top} \quad \text{s.t. } h(z) = -z_1^2 - z_2^2 + 1 = 0.$$

As shown in Figure 21a, we uniformly select 11 preference vectors across $\Delta_2^+$, and our WC-Penalty algorithm recovers the entire Pareto front.

**(iii)** Following the synthetic example in (Ye et al., 2024), we consider an LLGC-MTBL problem as follows:

$$\min_{x,y} \left((y_1-1)^2 + (y_2-x)^2 + y_3^2, (y_1-2)^2 + (y_2-x)^2 + y_3^2\right)^{\top} \quad \text{s.t. } y \in \arg\min_y g(x,y) := \frac{1}{2}(y_1-x)^2 + \frac{1}{2}(y_2-x)^2 + \frac{1}{4}y_3^4.$$

It is easy to verify that $\nabla_{yy}^2 g(x,y) = \mathrm{Diag}(1, 1, 3y_3^2)$, which implies that $g(x,\cdot)$ is convex but not strongly convex for any given $x$. As shown in Figure 21b, we uniformly select 26 preference vectors across $\Delta_2^+$, and our WC-Penalty algorithm, again, explores the entire Pareto front.

