# OpenReview forum: "A Tale of Two Problems: Multi-Task Bilevel Learning Meets Equality Constrained Multi-Objective Optimization"
_ICML.cc/2026/Conference — ICML 2026 regular_

### Official Review · Reviewer_Bjfn · 2026-03-11

**Soundness:** 4
**Presentation:** 4
**Significance:** 4
**Originality:** 4
**Overall Recommendation:** 5
**Confidence:** 5

**Summary:**

In this paper, the author makes the first attempt to extend BLO to the multi-task setting under a relaxed lower-level general convexity (LLGC) assumption. The theoretical contribution is clear.

**Compliance With Llm Reviewing Policy:**

Affirmed.

**Final Justification:**

I have checked other reviewers' comments, and some concerns are very insightful. Given that my problems have been addressed, I increased my score to 5.

**Key Questions For Authors:**

1. What is the difference between the KKT condition in this work and in FORUM?

2. The "lower-level general convexity (LLGC) assumption" is called the LLS condition, which was first formally proposed in the work  (A generic first-order algorithmic framework for bi-level programming beyond lower-level singleton. ICML)

3. Table 3. MOML has a journal version, which involves non-asymptotic results (Multi-Objective Meta-Learning, AIJ).

4. I suggest the author add a numerical synthetic MOBLO problem to verify their theoretical results. For example, in Appendix C of FORUM. Add addition dim to construct a convex but not strongly convex LL subproblem. Then verify the results.

5. I'm wondering if the LLGC assumption is really that difficult to handle? How about adding an L2 regularization term to the LL subproblem to make it strongly convex and then decreasing its weights, or just using methods in (A generic first-order algorithmic framework for bi-level programming beyond lower-level singleton. ICML)?

6. You mentioned lots of MTBL methods. Why do you compare three of them? I think Auto-Lambda is also easy to implement. You should add a discussion on the baseline selection.

I look forward to your reply and will adjust the score accordingly.

**Limitations:**

See questions.

**Strengths And Weaknesses:**

Strengths:
1. Well-motivated problem. Previous work on MOBLO mainly focuses on strong convex assumpution on LL subproblem.

2. Impressive theoretical results.

Weaknesses:
The major problem of this work is the lack of numerical adjustment.

---

> ### Author Rebuttal · Authors · 2026-03-31
>
> Thanks for your constructive comments. Please see our point-to-point responses below:
>
> **Response to Q1:**
> * The KKT condition considered in FORUM is:
> $$\sum_{i=1}^S\lambda_i\nabla f_i(z)+\nu\nabla q(z)=0,\\,\\,q(z)\le0,\\,\\,\nu\ge0,\\,\\,\nu q(z)=0,$$
> where $q(z)=g(z)-g^*(z)$ is the value function, and $\nu$ is the dual variable.
> * The KKT condition in our work is:
> $$\sum_{s=1}^S\omega_s-1=0,\\,\\,\sum_{s=1}^S\omega_s\lambda_s\nabla f_s(z)+\sum_{i=1}^q\nu_i\nabla h_i(z)=0,\\,\\,h(z)=0,\\,\\,\lambda_sf_s(z)-\rho\le0,\\,\\,\omega_s\ge0,$$
> where $h(z)=\nabla_yg(z)$ is the gradient, $\rho$ is the WC-variable, and $\nu,\omega$ are the dual variables.
>
> While both works use KKT conditions, our approach differs from FORUM in two key aspects:
> * **Different Reformulation:** FORUM considered the LLSC setting and uses a value-function-based reformulation to handle the bilevel structure. In contrast, our approach considers the LLGC setting and uses a lower-level stationarity reformulation. This approach leads to our distinct KKT system above.
> * **Different Algorithmic Design:** Our WC-scalarization technique also leads to the different KKT system. Specifically, our WC-scalarization converts the upper-level objectives to constraints, which introduces a unique term in our KKT system (cf. Line 262, right column, the last term) absent in FORUM.
>
> **Response to Q2:**  We remark that the "LLS" condition in the mentioned reference (referred to as [Liu'20] hereafter) stands for "Lower-Level Singleton". Both [Liu'20] and this work actually focus on **relaxing** it by considering the LLGC setting. However, our work further considers multiple objectives at the upper level, which fundamentally differs from [Liu'20].
>
> **Response to Q3:** Thanks for pointing this out. We will update Table 3 in our revision accordingly.
>
> **Response to Q4:** Similar to FORUM, we consider a synthetic LLGC-MTBL problem with $x\in\mathbb{R},y\in\mathbb{R}^3$ as follows:
> $$\min_{x,y}\Big((y_1-1)^2+(y_2-x)^2+y_3^2,(y_1-2)^2+(y_2-x)^2+y_3^2\Big)^\top\text{ s.t. }y\in\arg\min_yg(x,y):=\frac{1}{2}(y_1-x)^2+\frac{1}{2}(y_2-x)^2+\frac{1}{4}y_3^4.$$
> Note that $\nabla_{yy}^2g(x,y)=\mathrm{Diag}(1,1,3y_3^2)$ implies only convexity but not strong convexity, and the Pareto optimal solutions are $\\{(x,y):x\in[1,2],y=(x,x,0)^\top\\}$.
>
> We use our proposed WC-Penalty algorithm to solve this problem with parameters $u=v=5,\eta=0.05,T=1000$. By uniformly selecting the preference vectors $\lambda\in\Delta_2^+$, our algorithm explores the Pareto front (see the figures at https://anonymous.4open.science/r/Rebuttal_Experiments-F75C).
>
> This synthetic example shows that our method effectively handles the LLGC setting in MTBL. We will include this experiment in the revised appendix.
>
> **Response to Q5:** While adding $\ell_2$-regularization has been used in LLGC-BLO literature, our proposed method offers distinct and significant advantages:
> * $\ell_2$-regularization-based methods that "restore" LLSC require intricate tuning in shrinking the regularizer to achieve finite-time convergence [1,2], which often doesn't work well in practice. In contrast, our method fundamentally bypasses this tuning process through the ECMO reformulation, which leads to a new theoretical framework to efficiently solve the LLGC-MTBL problem.
> * Our approach can be generalized to even more challenging scenarios, such as MTBL problems **with lower-level constraints**, which can be absorbed into the $h(z)=0$. In contrast, how to extend regularization-based methods to constrained lower-level settings remains unclear.
>
> [1] A generic first-order algorithmic framework for bi-level programming beyond lower-level singleton\
> [2] Averaged method of multipliers for bi-level optimization without lower-level strong convexity
>
> **Response to Q6:** Upon carefully reading Auto-Lambda [1], we note that it considered a specialized weighting problem for multi-task learning, which is different from ours:
> * Our method and the selected baselines address the general MTBL formulation. In contrast, the weight $\lambda$ in [1] *only* appears in the lower-level training loss, not the upper-level validation loss.
> * The notion of "primary tasks" in [1] is neither applicable to our problem setting, nor to other baselines. As a result, their specific update rules (Eqs. 4-6) do not fit our data curation or meta-learning tasks (cf. Sec.5 and App.E.3).
>
> Because of these fundamental formulation differences, the approach in [1] and our work are not directly comparable. Regarding baseline selection, we chose baselines that represent state-of-the-art MTBL methods that provide rigorous theoretical convergence guarantees (cf. Line 1862). This selection is also consistent with the closely related work [2] for fair comparisons. As suggested, we will add this discussion in Sec. 5.
>
> [1] Auto-Lambda: Disentangling Dynamic Task Relationships\
> [2] A first-order multi-gradient algorithm for multi-objective bi-level optimization

---

> > ### Author Rebuttal · Reviewer_Bjfn · 2026-04-02
> >
> > Thank you for your reply, I will keep my score.
> >
> >
> > ---
> > Updated (03 Apr):
> >
> > I have checked other reviewers' comments, and some concerns are still very insightful.
> >
> > Thanks for your suggestions. I have increased my score to 5.

---

> > > ### Author Response · Authors · 2026-04-03
> > >
> > > **Response:** We sincerely appreciate your careful reading and constructive feedback, which have directly helped us to highlight the theoretical intuitions through a synthetic experiment and to better contextualize our work in the literature.
> > >
> > > Given that all your concerns have been adequately addressed as you indicated, we would be very grateful if you would consider re-evaluating the paper for a potential score increase. Thank you again for your time and constructive guidance.

---

### Official Review · Reviewer_9q8q · 2026-03-12

**Soundness:** 2
**Presentation:** 2
**Significance:** 2
**Originality:** 2
**Overall Recommendation:** 3
**Confidence:** 4

**Summary:**

This paper considers a bilevel optimization (BLO) problem with a multi-objective upper-level function and an unconstrained, smooth, and convex lower-level problem. The authors approach this by reformulating the lower-level problem via its first-order stationarity conditions, reducing the problemto a single-level constrained multi-objective optimization (MOO) problem. They then employ Weighted-Chebyshev (WC) scalarization to convert this into a conventional scalar-valued optimization problem, which is subsequently solved using a penalty-based algorithm. Theoretical convergence analysis of the proposed algorithm and accompanying numerical experiments are provided.

**Compliance With Llm Reviewing Policy:**

Affirmed.

**Final Justification:**

The primary concern remains that the method appears to be a general-purpose algorithm for constrained multi-objective optimization, framed within a bilevel learning narrative without exploiting any structural properties specific to bilevel problems.

Because the authors did not successfully demonstrate how their approach addresses the unique challenges of bilevel optimization or how it differs from standard multi-objective solvers, the novelty and relevance to the bilevel optimization remain unconvincing. Consequently, I maintain my negative assessment.

**Key Questions For Authors:**

what specific structural properties or challenges unique to bilevel optimization does your algorithm actually exploit?

**Limitations:**

yes

**Strengths And Weaknesses:**

Strength:

The paper presents a theoretical convergence analysis for the proposed penalty-based algorithm and provides numerical experiments to validate the approach.

Weaknesses:

- While the abstract and introduction emphasize the "bilevel" nature of the problem, the core methodology largely bypasses the complexities of bilevel optimization. By simply replacing the unconstrained, convex lower-level problem with its standard stationarity conditions, the formulation is reduced to a standard nonlinear equality-constrained problem. The paper does not exploit any special structural properties of bilevel optimization. Furthermore, the treatment of the resulting constrained MOO problem is heavily reliant on standard techniques: applying WC scalarization to yield a conventional scalar problem, followed by a classical quadratic penalty method combined with proximal gradient descent. Consequently, the algorithmic novelty is highly limited.

- The extensive discussion regarding the stationarity of MOO problems in Section 3 is largely redundant, as this topic is thoroughly established in the optimization literature (see, for example, Cristofari, A., De Santis, M. & Lucidi, S. On Necessary Optimality Conditions for Sets of Points in Multiobjective Optimization. J Optim Theory Appl 203, 126–145 (2024)).
Dedicating significant space (e.g., the left column of Page 4 and Figure 2) to illustrate that unconstrained MOO stationarity conditions cannot be applied to constrained cases is unnecessary; this is a widely understood concept.
Definition 4 relies on a notion of a "feasible direction" that is problematic for general nonlinear constraints. For instance, if a constraint has curvature, such as $h(z)=z^2
 =0$, a non-zero feasible direction does not exist for all feasible z, rendering the subsequent definition of stationarity meaningless in such contexts.


- The numerical experiments do not properly isolate the source of the proposed method's performance. Comparing the proposed algorithm (which utilizes WC scalarization) against baseline bilevel algorithms that rely on linear scalarization constitutes an unfair comparison. To demonstrate the actual efficacy of the proposed optimization algorithm, the baselines should be evaluated using the same WC scalarization framework. In its current state, the empirical section fails to provide meaningful insight into why the proposed algorithmic machinery (beyond the mere choice of scalarization) is effective or advantageous.

---

> ### Author Rebuttal · Authors · 2026-03-31
>
> Thanks for your constructive comments. Please see our point-to-point responses below:
>
> **Response to W1,Q1:** We remark that the "bypass" approach is precisely the **key idea and contribution** in solving the challenging MTBL problem. Note that translating MTBL to ECMO is *far from* a trivial reduction. To do so, one has to **exploit the special structural properties of the LLGC-MTBL problem** and fully leverage the equivalence relationship under stationarity conditions. Consequently, we build an elegant path (cf. Fig. 1) bridging these two foundational problems. Then, by developing the theory and algorithm for the ECMO problem, the MTBL problem is **equivalently** solved.
>
> Moreover, while the WC-scalarization is an existing technique in multi-objective optimization, our work has the following novelties:
> * **Theory:** The use of WC-scalarization in the constrained non-convex setting is **new** and non-trivial. We rigorously prove a **bijection** between the solutions of the ECMO and WC-scalarized problems in the constrained non-convex setting.
> * **Algorithm:** Our convergence analysis overcomes unique challenges in controlling the KKT system with complex constraints, which necessitates **new** proof techniques unseen in the standard WC-scalarization (cf. App.D, Line 1144).
>
> **Response to W2:** Upon checking [Cristofari et al. 24] (an independent work *unknown* to us at submission time, thanks for the pointer! referred to as [C'24] hereafter for convenience), we agree that our discussion of Pareto stationarity in Sec.3 is similar to that in [C'24]. However, there remain several subtle and yet important differences:
> * Our Def. 4 is based on *feasible descent directions* from a *geometric* perspective, which is similar to the weakly Pareto optimal characterization in Lemma 3.1 in [C'24]. However, since our Def. 4 is on Pareto stationarity, which is weaker than weakly Pareto optimality, our Def. 4 does **not** require equality constraints to be linearly independent.
> * In contrast, [C'24] defines the Pareto stationarity based on KKT-type conditions in their Def. 3.4 from an *algorithmic* design perspective, which is similar to the converse part of our Thm. 1. However, the 2nd condition $\sum_{i=1}^q\mu_{s,i}\nabla h_i(\tilde{z})d\neq0$ of Thm. 1 is **not** in their Def. 3.4 for the reason described next.
> * Our Thm. 4 characterizes both the **necessary and sufficient conditions** for "Local Pareto Optimality $\iff$ Pareto Stationarity", which is **different** from [C'24], where only the *necessary* conditions of weak Pareto optimality is established (cf. their Prop. 3.3).
>
> We would condense some similar parts and highlight the above key differences in our revision.
>
> Regarding the reviewer's example $h(z)=z^2=0$, we respectfully argue that this does not invalidate our definition for the following reasons:
> * From our Def. 4 and $h(z):=z^2=0$, $\tilde{z}=0$ is the unique feasible solution. Thus, the set of feasible directions is the singleton $\\{0\\}$. Consequently, no feasible improving direction $d$ exists that can improve all objective functions. This is consistent with Def. 4 that $\tilde{z}$ is a Pareto stationary point.
> * We agree with the reviewer regarding equality constraints with curvature, where the set of feasible directions may be empty. Fortunately, this issue can be addressed by generalizing the notion of **feasible direction set** to the **Tangent Cone** [1], which is a standard treatment in the optimization literature. We will add the above discussion in our revision.
>
> [1] Constraint qualifications, Solodov, M. V., 2010
>
> **Response to W3:** To our knowledge, there are no existing WC-scalarized algorithms in the MTBL literature except for a recent work [1], which assumed LLSC *rather than* LLGC. Other baselines in prior works [2] have been included in our experiments. However, none of them is based on WC-scalarization. To compare with the WC-based algorithm in [1] as suggested by the reviewer, in this rebuttal period, we have tested [1] on the Data Weighting task in Sec. 5.2. We can see from the table below that our method outperforms [1].
>
> |1/Loss|Helpfulness|Correctness|Coherence|Complexity|Verbosity|
> |-|-|-|-|-|-|
> |WC-MHGD|0.200|0.208|0.208|0.188|0.189|
> |**WC-Penalty (Ours)**|0.245|0.246|0.243|0.247|0.232|
>
> In addition to MTBL baselines, we also heuristically extended well-known BLO methods [3,4] to the MTBL setup, although none of them considered the LLGC setting. Note that extending them via Linear Scalarization (LS) is mathematically natural. However, directly extending [3,4] via WC-scalarization is infeasible due to the non-smooth "min-max" nature of WC even in the LLSC setting.
>
> [1] Multi-objective bilevel learning\
> [2] A first-order multi-gradient algorithm for multi-objective bi-level optimization\
> [3] Bilevel optimization: Convergence analysis and enhanced design\
> [4] A framework for bilevel optimization that enables stochastic and global variance reduction algorithms

---

> > ### Author Rebuttal · Reviewer_9q8q · 2026-04-01
> >
> > Thank the authors for their rebuttal. However, I remain unconvinced by their claim that "the 'bypass' approach is precisely the key idea and contribution in solving the challenging MTBL problem. Note that translating MTBL to ECMO is far from a trivial reduction. To do so, one has to exploit the special structural properties of the LLGC-MTBL problem and fully leverage the equivalence relationship under stationarity conditions." Replacing the lower-level problem with stationarity conditions is a standard and classical  technique in bilevel optimization. And the rebuttal does not demonstrate how special structural properties of the bilevel optimization problem were exploited. I therefore maintain my original score.

---

> > > ### Author Response · Authors · 2026-04-01
> > >
> > > **Response:** Thanks for your follow-up comments. We would like to further clarify how we have exploited the special structural properties of the LLGC-MTBL problem:
> > >
> > > * **The special structure of LLGC-MTBL is exploited throughout the entire algorithmic design process, not just the initial reformulation step:** The initial reformulation from the LLGC-MTBL problem to the ECMO problem via the equivalent LL stationarity should *not* be viewed in isolation and the only step where we exploit the special structure of LLGC-MTBL. Rather, it is just an initial step for our entire algorithmic design (Steps 1-5 as shown in Fig. 1). Our subsequent algorithmic steps also heavily rely on the special structure of the original MTBL problem. Notably, the special-structured KKT system proposed in **Definition 5** is a consequence of the ECMO problem, which largely inherits the structural properties of LLGC-MTBL. Note that our KKT system differs significantly from standard KKT systems in the literature. For example, the **special-structured terms** $\sum_{s=1}^S\omega_s\lambda_s\nabla f_s(z)+\sum_{i=1}^q\nu_i\nabla h_i(z)$ and $[\min\\{ \omega_s, \rho-\lambda_s f_s(z) \\}]\_{s\in[S]}$ in $\mathcal{K}(\rho,z,\omega,\nu,\lambda)$ in Definition 5 stem from the underlying multi-task bilevel structure under our proposed Weighted-Chebyshev scalarization approach. Furthermore, the $[\min\\{ \omega_s, \rho-\lambda_s f_s(z) \\}]\_{s\in[S]}$ in $\mathcal{K}(\rho,z,\omega,\nu,\lambda)$ term implies but is different from the conventional complementary slackness condition in standard KKT. Clearly, without considering and leveraging the special structure of the LLGC-MTBL problem, it is *impossible* for us to to derive such a special-structured KKT system, which is **unseen** in the literature and served as an important metric to measure the convergence to *Pareto-stationarity* in solving the LLGC-MTBL problem.
> > >
> > >
> > > * **The argument that "...replacing the lower-level problem with stationarity conditions is simply a `standard and classical technique'..." is debatable:** The *"lower-level stationarity reformulation"* approach (referred to as "**LLStRe**" hereafter for convenience) only appeared quite recently in the bilevel optimization literature out of the necessity of dealing with the highly challenging LLGC [1] and LLNC (lower-level nonconvex, [2-5]) settings. To date, most of the bilevel optimization algorithms in the machine learning field are still based on the LLSC setting (implying well-defined hypergradients).
> > >
> > >   To our knowledge, among all the existing works that studied bilevel optimization in the LLGC setting, only [1] had leveraged the LLStRe approach (note: LLStRe is *only a heuristic* for the LLNC setting as it doesn't guarantee any local optimality in the LL problem). Moreover, [1] does not consider multi-objective as in our work. In this sense, one may claim that this work is the first to use the LLStRe approach for solving LLGC-MTBL problems. We would highly appreciate it if the reviewer could provide more pointers to the LLStRe approach in the literature in case we have missed anything.
> > >
> > >   In our humble opinion, instead of being immersed in a philosophical debate over whether the LLStRe approach is "standard and classical", we argue that it is more important to focus on the whole body of our algorithmic design rather than dwelling on a specific step, when assessing the novelty of this paper. Note that, even after the first LLStRe step, the reformulated problem (ECMO) remains an open problem and necessitates a significant amount of efforts and **new ideas**. Indeed, all subsequent algorithmic designs and proof techniques used in convergence performance analysis in this paper differ fundamentally from those in prior works.
> > >
> > > In summary, in order to facilitate all algorithmic designs and theoretical analysis for solving LLGC-MTBL, we have to leverage the special structural structure of the LLGC-MTBL problem, which are highly non-trivial and goes far beyond simply applying the LLStRe approach in the first step.
> > >
> > > [1] Averaged Method of Multipliers for Bi-Level Optimization without Lower-Level Strong Convexity\
> > > [2] A Generalized Alternating Method for Bilevel Optimization under the Polyak-Łojasiewicz Condition\
> > > [3] A method for bilevel optimization with convex lower-level problem\
> > > [4] Moreau envelope for nonconvex bi-level optimization: A single-loop and hessian-free solution strategy\
> > > [5] Sun-dsbo: A structured unified framework for nonconvex decentralized stochastic bilevel optimization

---

### Official Review · Reviewer_PPpL · 2026-03-13

**Soundness:** 3
**Presentation:** 3
**Significance:** 2
**Originality:** 3
**Overall Recommendation:** 4
**Confidence:** 4

**Summary:**

This paper studied multi-objective bilevel optimization problem and solved it by reformulating it to a gradient-based equality constrained multi-objective optimization problem. The authors first established a new KKT Pareto stationarity and proposed a weighted Chebyshev penalty algorithm which converged to the stationary point in ${\cal O}(ST^{-1/2})$ iterations in both deterministic and stochastic settings. Numerical experiments on data weighting for multi-objective RLHF reward model training and LLM alignment validate the effectiveness of the proposed method.

**Compliance With Llm Reviewing Policy:**

Affirmed.

**Final Justification:**

Overall, this paper developed an efficient and principled penalized algorithm to solve multi-task bilevel learning problem with sufficient theoretical guarantee and superior empirical performance on LLM fine-tuning tasks. The rebuttal solves most of my concerns, including the relations of Assumption 2 and strong convexity, and the stationary equivalence of the penalized problem with the original problem. Therefore, I decided to raise the score to 4. I cannot further increase my score because there remains a minor gap: establishing the equivalence between the KKT condition and local optimality requires the algorithm to converge not only to a first-order stationary point, but also to a second-order stationary point. But except for this small gap, I appreciate the overall contribution of this work.

**Key Questions For Authors:**

Key questions are listed in weakness.

If the stationary relations are hard to establish, what is the key reason and the challenges?

Assumption 2 needs more justification to differentiate with the strongly convex setting, or the paper needs to be tone down for this point.

I'm willing to increase the score if these two main concerns are solved.

**Limitations:**

Yes

**Strengths And Weaknesses:**

Strengths:

1. Multi-objective bilevel optimization problem has many applications in LLM fine-tuning and this paper proposed an efficient and easy-to-implement algorithm with strong theoretical guarantee.

2. The local and global solutions equivalence of multi-objective bilevel problem and the WC-scalarized problem are well-justified.

3. Experiments on LLM fine-tuning validate the effectiveness of the proposed algorithm.

Weakness:

1.	As the algorithm is proved to converge to the stationary point of WC-scalarized problem, it is not clear that whether the stationary point of WC-scalarized problem is closed to the original multi-objective bilevel problem or not. It would be helpful to establish the stationary relations of WC-scalarized problem and the original problem, as in, e.g. [A1, proposition 3].

2.	Another concern is on the Assumption 2. LICQ itself is a standard assumption in constrained optimization. However, as you focus on the convex objective function case where $g(x,\cdot)$ is convex, combined with the linear independent conditions of $\{\nabla h_i(z),\nabla f_s(z)\}$, I’m wondering whether they imply the strongly convexity. The discussion in this paper is not fully convinced on this point, especially linear independency of $\{\nabla h_i(z),\nabla f_s(z)\}$ appears stronger than that of $\{\nabla h_i(z)\}$. It would be better if the authors can provide an example that is not strongly convex but satisfies Assumption 2 and convexity. As a side note, although convexity is mentioned in the beginning, it is good to state it explicitly as a formal assumption to improve the readability.

[A1] Shen, Han, Quan Xiao, and Tianyi Chen. "On penalty-based bilevel gradient descent method: On penalty-based bilevel gradient descent method." Mathematical Programming 214.1-2 (2025): 539-589.

---

> ### Author Rebuttal · Authors · 2026-03-31
>
> Thanks for your constructive comments. Please see our point-to-point responses below:
>
> **Response to W1:** The WC-scalarized stationary solutions are indeed **equivalent** to those of the original multi-objective bilevel problem. In fact, this is a key contribution stated in **Theorem 2**, which rigorously established the **bijection** between the *locally weakly Pareto stationary solutions to the ECMO problem* and the *locally optimal solutions of the WC-scalarized problem*. Thus, this **1-to-1** mapping ensures that, by solving the WC-scalarized problem and varying the preference vector across the standard simplex, one can explore the entire Pareto front. We'll highlight this point in the revision.
>
> **Response to W2:** We would like to clarify that Assumption 2 does **not** implicitly require strong convexity. Algebraically, Assumption 2 only requires that the gradient matrix (consisting of the gradients of constraint $h_i(z),i\in[q]$) and the gradient of the active WC-constraint $f_s(z),\lambda_sf_s(z)=\rho$, has full column rank. Notably, this matrix takes the shape of $\mathbb{R}^{k\times(q+s(z))}$, where $k$ denotes the dimension of the variable $z$ and $s(z)\le S$ denotes the number of active WC-constraints at point $z$. In practice, $k\gg q+S \ge q+s(z)$ holds in general, since the parameter space is typically high-dimensional. Consequently, Assumption 2 merely requires a "**tall and skinny**" matrix to have full column rank, which is a mild condition rather than a restrictive strong convexity requirement.
>
> To further illustrate that convexity and Assumption 2 do not imply strong convexity, consider the following simple example with $x,y\in\mathbb{R}$:
> $$f_1(x,y)=y,\\,\\,f_2(x,y)=y+1,\\,\\,g(x,y)=xy+\frac{1}{4}y^4.$$
> It then follows that:
> $$h(x,y)=\nabla_yg(x,y)=x+y^3,\\,\\,\nabla_{yy}^2g(x,y)=3y^2\ge0,$$
> which implies that $g(x,\cdot)$ is **only convex but not strongly convex**. On the other hand, the aforementioned gradient matrix:
> $$\left[\begin{align*}&1&0\\\\&3y^2&1\end{align*}\right],$$
> always has a determinant of 1, meaning it is **full column rank**, thus perfectly satisfying Assumption 2. We thank the reviewer again and will add these discussions into the revision.

---

> > ### Author Rebuttal · Reviewer_PPpL · 2026-04-03
> >
> > Thank you for the response, and I appreciate the authors’ response to the second point, which fully resolved that concern. For the stationary equivalence, it seems that Theorem 2 in the main paper only established the local solution relations rather than the stationarity, but please correct me if I miss something.

---

> > > ### Author Response · Authors · 2026-04-03
> > >
> > > **Response:** Thanks for your follow-up comment. In what follows, we would like to further clarify **why the local optimality of the WC-solution in Theorem 2 (guaranteed by KKT-solution in Theorem 3) implies the Pareto-stationarity of the corresponding solution to the ECMO problem** (i.e., there exists a 1-to-1 mapping between them):
> > > 1. Note first that **Theorem 3** shows that our WC-Penalty algorithm converges to an **$\epsilon$-KKT solution**, i.e., the solution that satisfies $\\|\mathcal{K}(\rho,z,\omega,\nu,\lambda)\\|^2 \le \epsilon$ for any $\epsilon$, with a finite-time convergence rate.
> > > 2. **Lemma 1** shows that under the mentioned constraint qualification conditions, a point is an **$\epsilon$-KKT solution iff it is a locally optimal WC-solution**.
> > > 3. **Theorem 2**, as the reviewer pointed out, further establishes the equivalence between the **locally optimal WC-solution** and the **locally weakly Pareto optimal solutions** to the ECMO problem.
> > > 4. Lastly, **Theorem 1** says that the **locally weakly Pareto optimal solutions**  to the ECMO problem is also the **Pareto stationary solution** to it.
> > >
> > > In summary, this **combined logic chain** builds the following bridge between **local optimality of WC-solution** and **Pareto stationarity of the ECMO solution**:
> > >     $$ \begin{aligned} \text{WC-Penalty} \stackrel{\text{Thm. 3}}{\implies} &\epsilon\text{-KKT solution} \\\ \stackrel{\text{Lemma 1}}{\iff}  &\text{locally optimal WC-solution} \\\ \stackrel{\text{Thm. 2}} {\iff} & \text{locally weakly Pareto optimal solutions to ECMO} \\\ \stackrel{\text{Thm. 1}}{\implies} & \text{Pareto stationary solutions to ECMO} \end{aligned}$$
> > >
> > > Therefore, our theoretical results indeed prove that our proposed WC-Penalty algorithm achieves a Pareto stationary solution of the ECMO problem, which also implies Pareto stationary solution to the original MTBL problem. Also, due to the non-convexity of the MTBL and the associated ECMO problems, the **Pareto stationary solution** is the necessary condition of Pareto optimality of MTBL/ECMO.
> > >
> > > We sincerely thank the reviewer for the thoughtful and insightful questions, which have greatly contributed to improving the quality of this work. We hope that the detailed clarifications provided above adequately address the reviewer's follow-up concerns. Should these clarifications meet the reviewer's expectations, we would be most grateful if the reviewer would kindly consider re-evaluating the manuscript, with the possibility of a raised score. We deeply appreciate the reviewer's time, expertise, and constructive guidance throughout this process.

---

### Official Review · Reviewer_iUYp · 2026-03-13

**Soundness:** 3
**Presentation:** 2
**Significance:** 2
**Originality:** 3
**Overall Recommendation:** 4
**Confidence:** 4

**Summary:**

This paper studies multi-task bilevel learning (MTBL) under the lower-level general convexity (LLGC) assumption by reformulating MTBL as an equality-constrained multi-objective optimization (ECMO) problem. To solve ECMO, the paper links ECMO to weighted-Chebyshev scalarization, uses the KKT system of the scalarized problem to define Pareto stationarity, and then designs a WC-Penalty method with slack variables, penalty terms, and projected gradient updates to optimize this characterization. The paper proves a finite-time convergence guarantee to KKT-based Pareto stationarity with rate $\mathcal{O}(ST^{-\frac{1}{2}})$. Experiments on multi-objective RLHF reward-model training and LLM alignment show that the method recovers a larger portion of the Pareto front than baselines, and scales to larger LLM settings.

**Compliance With Llm Reviewing Policy:**

Affirmed.

**Final Justification:**

All of my concerns and questions have been fully addressed in the authors' detailed rebuttal and follow-up responses. I hereby update my score to 4.

**Key Questions For Authors:**

1. Since the paper’s main theoretical guarantee is convergence to KKT-based Pareto stationarity rather than global Pareto optimality, could the authors provide more empirical validation, for example by comparing against a simpler pipeline that directly trains on the training set and selects the checkpoint with the best validation performance?
2. The related-work discussion seems to focus more on earlier literature. Are there also more recent works closely related to LLGC bilevel optimization, constrained multi-objective optimization, or multi-task bilevel learning that should be discussed here?

**Limitations:**

No, the paper includes an Impact Statement, but it is too minimal to count as an adequate discussion of limitations or potential negative societal impact.

Suggestions:
1. Add a explicit limitations paragraph stating that the main theory only guarantees convergence to KKT-based Pareto stationarity under assumptions such as smoothness and LICQ, rather than global Pareto optimality.
2. Discuss possible practical failure modes in the RLHF use cases, such as sensitivity to the choice of preference weights, the risk of over-optimizing certain validation criteria at the expense of others, and the possibility that the selected Pareto solution may encode unintended trade-offs.

**Strengths And Weaknesses:**

Strengths:
1. The paper identifies a underexplored setting, which is multi-task bilevel learning under lower-level general convexity.
2. The authors show a strong understanding of the area, and the paper is clear about what is solved, as well as what remains unresolved.
3. The technical development is well organized: the paper motivates a new Pareto stationarity notion for ECMO, connects it to weighted-Chebyshev scalarization, and then derives the WC-Penalty method in a logically coherent way.
4. The proofs and assumptions are presented in a complete and transparent manner.

Weaknesses:

If the authors could properly address them during the rebuttal phase, I am willing to raise my score.
1. The main theoretical guarantee is only convergence to KKT-based Pareto stationarity, rather than global Pareto optimality in the general nonconvex setting. Stronger global-type guarantees only appear in the convex special case in the appendix, so the practical meaning of the stationary solutions in the main setting remains somewhat limited.
2. Although the paper emphasizes that prior work mainly focuses on the single-task setting while this work addresses the multi-task case, it does not sufficiently explain why the multi-task setting is important. A clearer discussion of the motivation and significance would strengthen the paper.
3. I am curious about the relationship between this paper and several prior works on multi-domain training for large language models, such as [1][2][3][4]. If the authors could add some discussion on this point, it would greatly strengthen the paper's context and positioning.
4. The experimental modeling for the reward-model training task is not in the main text. The main paper gives only a brief high-level description of mixing datasets and optimizing multiple validation criteria, which makes it harder to understand exactly how the bilevel formulation is instantiated and what role the proposed method plays in that experiment.
5. The experimental evaluation is somewhat limited, as it mainly focuses on MTBL-style tasks; given that ECMO is positioned as a broader foundational optimization contribution, the paper would benefit from additional standalone ECMO benchmarks.

[1] How Abilities in Large Language Models are Affected by Supervised Fine-Tuning Data Composition. ACL 2024.

[2] Mixture-of-Skills: Learning to Optimize Data Usage for Fine-Tuning Large Language Models. EMNLP 2024.

[3] Boosting Multi-Domain Fine-Tuning of Large Language Models through Evolving Interactions between Samples. ICML 2025.

[4] Boosting Multi-Domain Reasoning of LLMs via Curvature-Guided Policy Optimization. ICLR 2026.

---

> ### Author Rebuttal · Authors · 2026-03-30
>
> Thanks for your constructive comments. Please see our point-to-point responses below:
>
> **Response to W1, Q1, L1:** It is true that our theory guarantees the convergence to Pareto stationarity rather than global optimality. However, this is due to the fundamental NP-hardness of the **non-convex** upper-level problem of MTBL in this paper, **not** a limitation of our algorithm.
>
> To see this, note that even in non-convex single-objective optimization, seeking a global optimal solution is NP-hard. Thus, it is typical to pursue to a *stationary solution*. In our MTBL/ECMO problem, the multi-objective structure further compounds this NP-hardness. Hence, pursuing a Pareto-stationary solution is the best one can do for non-convex MTBL problems.
>
> The NP-hardness of solving non-convex MTBL problems can also be seen in the literature, where **all** prior works only established stationarity-based guarantees instead of global optimality (see Tab. 3). We will add the above discussions on the NP-hardness and seeking KKT-based Pareto-stationarity in the revision as suggested by the reviewer.
>
> **Response to W2:** Many emerging fields tackle complex-structured problems that *motivate* multi-task and bilevel formulations to capture conflicting or nested objectives. The practical relevance of multi-task setting is evidenced by a surge of recent works on meta-learning, LLM alignment, and contrastive learning [1-5], etc. We'll incorporate a discussion of these motivating applications in our revision.
>
> [1] Multi-objective meta learning\
> [2] A first-order multi-gradient algorithm for multi-objective bi-level optimization\
> [3] EMORL: Ensemble Multi-Objective Reinforcement Learning for Efficient and Flexible LLM Fine-Tuning\
> [4] Iterative Foundation Model Fine-Tuning on Multiple Rewards\
> [5] Objective Soups: Multilingual Multi-Task Modeling for Speech Processing
>
> **Response to W3:** Multi-domain training (MDT) in [1-4] is related to MTBL in this paper but has several key differences:
> * **Connections**: MDT considers balancing multiple conflicting objectives, which can be inherently formulated as a multi-objective optimization (MOO) problem. Thus, our WC-scalarization technique and the associated MOO insights could be leveraged for solving MDT.
> * **Differences:** *1) Bilevel Structure:* The bilevel structure in MTBL addresses nested objectives, which is *not* seen in MDT. *2) Theoretical Guarantees:* Unlike [1-4] that only provide heuristic solutions to MDT, we develop algorithms with rigorous theoretical guarantees, which are also verified by our experiments.
>
> **Response to W4:** Due to space limitation, we relegated detailed experiment settings on the bilevel formulation instantiation, dataset construction, and hyperparameter selection in Appendix E in the initial submission. With an extra page permitted in the revision, we will incorporate detailed experimental settings in Sec. 5 as suggested by the reviewer.
>
> **Response to W5:** Although there are a few related works on ECMO in the literature (e.g., [1,2]), there is *no* ECMO benchmark in the machine learning community. In addition, the numerical experiments in [1,2] are restrictive: they only considered synthetic "toy" examples without using datasets for large-scale tasks.
>
> Following your suggestion, we have tested our WC-Penalty algorithm on Example 4.1 of [1] stated as follows:
> $$\min_{z\in\mathbb{R}^2}F(z)=\Big((z_1-2)^2+(z_2-1)^2,(z_1-2)^2+(z_2+1)^2\Big)\text{ s.t. }h(z)=-z_1^2-z_2^2+1=0.$$
> Our new experiment results (https://anonymous.4open.science/r/Rebuttal_Experiments-F75C) show that our method **explores the entire Pareto front** by varying $\lambda$ across $\Delta_2^+$. We'll include these result in our revision.
>
> [1] A descent method for equality and inequality constrained multiobjective optimization problems\
> [2] A benchmark for equality constrained multi-objective optimization
>
> **Response to Q2:** To our knowledge, there is **no** related work using the LLGC assumption. While there are several recent works in the BLO and MOO literature, **none** of them is directly applicable to MTBL due to the mixed BLO-MOO structure. The only related MTBL work is [1]. However, [1] requires the LLSC assumption rather than the much weaker LLGC assumption in this paper. We will incorporate the above discussions in our revision.
>
> [1] Multi-objective bilevel learning, AAAI'26
>
> **Response to L2:** We would like to clarify that the "choice of preference weights" problem is *irrelevant* in this work due to the "Pareto front exploration" (i.e., all Pareto-solutions) solution philosophy. Specifically, the beauty of our proposed method is that, the WC-scalarization enables the **Pareto front exploration** by varying the preference weights in the standard simplex. Thus, with our method, one does **not** pick a specific preference weight at the beginning. Rather, the user first explores the Pareto front by enumerating preference weights and then pick a favorable solution from the Pareto front.

---

> > ### Author Rebuttal · Reviewer_iUYp · 2026-04-02
> >
> > Thank you for the rebuttal. Most of my concerns have been addressed.
> >
> > Regarding W3, I appreciate the authors' clarification. I would be glad to see a **more detailed and in-depth discussion** of the connections and differences between MTBL and these studies on MDT, as well as **how the authors plan to incorporate this discussion into the revised paper**. If the authors' further response is satisfactory, I will raise my score to 4 by directly editing my review, even though I will no longer be able to reply at that stage.

---

> > > ### Author Response · Authors · 2026-04-03
> > >
> > > **Response:** Thanks for your follow-up! We have carefully reviewed Refs. [1-4] on MDT. First, we note that our work is focused on a general MTBL problem, with a wide range of applications including MDT [1-4] for LLMs (as evidenced by our experiments on dataset curation for LLMs in Sec. 5). Next, we will provide in-depth discussions on the relationship between MDT [1-4] and our work:
> > > * **Ref. [1]** empirically studied how **dataset composition ratios** affect the performance of multi-domain LLM fine-tuning. Our MTBL method can be used in the data composition task to **quantitatively** and **theoretically** answer *Research Questions 3-4* and improve the performance of the dual-stage mixed fine-tuning (DMT) in [1]. Specifically, DMT in [1] can be formulated as an MTBL problem, where i) the lower level corresponds to the composition ratio optimization of the SFT datasets in the *1st-stage* similar to our data curation in Sec. 5; and ii) the upper level corresponds to the *2nd-stage* multi-domain model training with the mixed-general-SFT dataset. Note that our MTBL method offers **Pareto front exploration** guarantee for the DMT approach in Ref. [1].
> > > * **Ref. [2]** formulated the data sampling/usage problem for MDT as a *bilevel* optimization problem. In [2], the lower level learns the parameters $\psi$ to characterize the sampling distribution over multi-domain datasets. Upon solving the lower-level problem, the upper level trains the model parameter $\theta$ for multi-domain learning based on the multi-domain dataset sampling. Ref. [2] is closely related to our work in the sense that the lower-level multi-domain dataset sampling distribution is mathematically equivalent to the dataset weighting curation in our Sec. 5. The only difference is that the lower-level sampling problem in [2] is solved via a reinforcement learning approach. Due to this close relation, Mixture-of-Skills (MoS) in [2] can be formulated as MTBL and our method is applicable to MoS. Moreover, our MTBL approach can enhance the MoS results in [2] as follows: Our **weighted-Chebyshev-penalty** approach (referred to as WC-Penalty hereafter, cf. Eq. (1) in our work) can replace the linear scalarization approach (LS) in [2] to achieve a **finite-time Pareto-stationarity convergence rate guarantee** (cf. Theorem 3) and systematically **explore the entire Pareto front** (cf. Figure 7).
> > > * **Ref. [3]** investigated the multi-domain fine-tuning problem in LLMs, where an evolving interaction-guided curriculum (EVIC) method is proposed to dynamically sample from a multi-domain dataset and adapt the training curriculum to prioritize samples that have the most contributions. Throughout the entire training process, EVIC evaluates the inter-sample interactions for SFT sample selection. Compared to [1,2], Ref. [3] is less related to our work in the sense that the composition of multi-domain dataset has been provided (cf. Step 1 of Alg. 1 in [3]). That said, our MTBL approach can still be used to enhance the performance of EVIC in a **complementary** fashion. Specifically, the lower level of MTBL can be used to optimize the composition of the multi-domain dataset $D$, which is then used in the upper-level for EVIC-based multi-task learning. We note that our WC-Penalty multi-objective optimization technique can also be integrated in EVIC and replace the LS-based average performance metric (cf. Page 5, right column in [3]) to offer Pareto-type theoretical performance guarantees.
> > > * **Ref. [4]** studied a multi-domain reinforcement learning problem for optimizing multi-domain reasoning performance of LLMs. The key idea of the CGPO method proposed in [4] is to leverage Hessian-based curvature information to capture cross-domain interactions, which enhances the performance of GRPO-type policy-gradient methods. Similar to [3], our work could be used to potentially enhance the performance of CGPO in a **complementary** fashion. Specifically, the lower level of our MTBL approach could replace the step of *"repeatedly drawing mini-batches and performing a randomized sequential update"* (Steps 10-11 of Alg. 1 in [4]) by iteratively optimizing the "composition" of multi-task dataset. Moreover, our WC-Penalty-based multi-objective optimization technique can be leveraged in Step 14 of Alg. 1 to replace the LS-based aggregation and achieve Pareto-stationary convergence and facilitate Pareto front exploration.
> > >
> > > To formally incorporate the above discussions into our paper, we plan to:
> > > * Cite [1,2] as motivating applications of MTBL in Section 1 (Introduction) and Appendix B.
> > > * Cite [3,4] and compare them with our work in Section 2 (Related Work) to discuss ways of combining them to further enhance multi-domain learning performances.
> > > * Cite [1-4] as future directions in Section 6 (Conclusion) and as real-world applications in "Impact Statement."
> > >
> > > Again, we thank the reviewer for pointers to these related works on MDT, which have helped broaden the scope of MTBL.

---

### Decision · Program_Chairs · 2026-04-30

**Decision:**

Accept (regular)

**Comment:**

This paper studies multi-task bilevel learning under lower-level general convexity, a challenging and relatively underexplored setting. It reformulates the original problem as an equality-constrained multi-objective optimization problem, introduces a KKT-based notion of Pareto stationarity for the new formulation, and develops a weighted-Chebyshev penalty method with finite-time convergence guarantees. The experimental results on RLHF reward model training and LLL alignment are generally strong and suggest that the proposed approach is effective in exploring the Pareto front and outperforming relevant baselines.

Overall, most reviewers were positive about the paper's technical depth, clarity, and novelty, and found the connection between bilevel learning and constrained multi-objective optimization to be insightful. Many of the concerns raised in the initial reviews were adequately addressed during the rebuttal and discussion phase, including clarification of the theoretical target, the motivation for the multi-task setting, the role of the assumptions, and the relation to recent work on multi-domain LLM training.

There remains one reviewer with a score of 3 who was not fully convinced about the degree of algorithmic novelty, mainly arguing that the method can be viewed more as a general constrained multi-objective optimization approach than as one that deeply exploits bilevel structure. The authors responded to this concern in the discussion and later reiterated to the AC that this remaining disagreement should be considered alongside the broader post-rebuttal reviewer feedback. While this concern is worth noting, the authors engaged with it thoughtfully during the rebuttal and follow-up discussion, and I do not believe that this remaining difference in perspective outweighs the paper’s overall technical contribution. The paper identifies an important problem, develops a technically meaningful framework, and received supportive assessments from the majority of reviewers after rebuttal.
Therefore, I recommend accepting this paper.